# Asymptotically matched quasi-circular inspiral and transition-to-plunge in the small mass ratio expansion

**Geoffrey Compère[1\*] and Lorenzo Küchler[1,2†]**

**1** Université Libre de Bruxelles and International Solvay Institutes,
C.P. 231, B-1050 Bruxelles, Belgium
**2** Institute for Theoretical Physics, KU Leuven, Celestijnenlaan 200D,
B-3001 Leuven, Belgium

\* geoffrey.compere@ulb.be, † lorenzo.kuchler@ulb.be

## Abstract

In the small mass ratio expansion and on the equatorial plane, the two-body problem for point particles in general relativity admits a quasi-circular inspiral motion followed by a transition-to-plunge motion. We first derive the equations governing the quasi-circular inspiral in the Kerr background at adiabatic, post-adiabatic and post-post-adiabatic orders in the slow-timescale expansion in terms of the self-force and we highlight the structure of the equations of motion at higher subleading orders. We derive in parallel the equations governing the transition-to-plunge motion to any subleading order, and demonstrate that they are governed by sourced linearized Painlevé transcendental equations of the first kind. The first ten perturbative orders do not require any further developments in self-force theory, as they are determined by the second-order self-force. We propose a scheme that matches the slow-timescale expansion of the inspiral with the transition-to-plunge motion to all perturbative orders in the overlapping region exterior to the last stable orbit where both expansions are valid. We explicitly verify the validity of the matching conditions for a large set of coefficients involved, on the one hand, in the adiabatic or post-adiabatic inspiral and, on the other hand, in the leading, subleading or higher subleading transition-to-plunge motion. This result is instrumental for deriving gravitational waveforms within the self-force formalism beyond the innermost stable circular orbit.

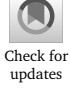 Check for updates

# 1 Introduction

Solving the binary problem in General Relativity has been an outstanding problem for a century. For a mass ratio of order unity, numerical relativity [1], the post-Newtonian/post-Minkowskian formalism [2], and effective one-body (EOB) methods [3] (for reviews, see [4–6]) have been successful at deriving accurate waveform models currently under confrontation with observations of gravitational waves produced from compact binary mergers [7,8]. For large mass ratios, waveforms can be generated from (non-linear) black hole perturbation methods using the self-force formalism [9, 10]. The inspiral motion can be efficiently solved using the slow-timescale expansion, which leads to the adiabatic approximation with post-adiabatic corrections [11–13]. One of the main motivation for developing methods in the small mass ratio limit is the prospect of observations of EMRIs (extreme mass ratio inspirals) by LISA [14,15]. Another motivation is the recent realization that, even when restricting to the

first post-adiabatic approximation, the small mass ratio expansion is applicable to intermediate mass binaries and, up to a certain extent, to equal mass binaries [16–18].

The slow-timescale expansion of a quasi-circular inspiral breaks down around the innermost stable circular orbit where the transition-to-plunge takes place [19, 20]. A transition-timescale expansion can be formulated and matched to the quasi-circular inspiral in the small mass ratio limit [19, 21–26], or in the small velocity limit using resummed post-Newtonian expansions [20, 27–34]. The main aim of this paper is to provide a systematic, accurate and complete treatment of the matching between the quasi-circular inspiral and the transition-to-plunge motion in the small mass ratio regime $\eta \ll 1$, neglecting finite size effects but taking into account self-force effects, following the leading-order asymptotic matching derived in [35, 36]. As shown in Section 6, while the leading-order transition matching admits an error of the order of $\eta^{3/5}$, the matching scheme that we propose when using the first and second order self-force only admits an error of the order of $\eta^2$.

After a summary of the equatorial forced geodesics equations governing the motion of point particles in the Kerr geometry including self-force corrections in Section 2, we derive in Section 3 the quasi-circular inspiral motion in the slow-timescale expansion and its asymptotic solution in the limit of reaching the last stable orbit. Our expansions complement the ones obtained at adiabatic and post-adiabatic order [10, 12] to post-post-adiabatic order and higher orders. We then turn to the transition motion around the last stable orbit in the transition-timescale expansion in Section 4 and compute the asymptotic solution to the equations of motion at early times. In Section 5, the matching of these two motions is performed in the overlapping region where both the slow-timescale and the transition-timescale expansions are valid. We derive in Section 6 the sketch of an algorithm to extend the post-adiabatic inspiral motion to the transition-to-plunge motion using the results obtained in previous sections. We conclude in Section 7. Several appendices collect lengthy formulae used in the main text.

**Ancillary files:** Four ancillary Mathematica notebooks are publicly available at this URL. They contain data associated with, respectively, Sections III.B,C,D,F.1,F.2, Sections III.E,F.3, Sections IV.A,B, and Sections IV.C,D.

## 2 Equatorial forced geodesics

We use the same conventions and notations as [35]. We consider a Kerr background with mass $M$ and angular momentum $J$. The Kerr metric in Boyer-Lindquist coordinates is used to raise and lower spacetime indices. We use geometrical units $G = c = 1$. Quantities are made dimensionless using the black hole mass $M$, including the angular momentum $a = J/M^2$ and the binary mass ratio $\eta = m/M$ where $m$ is the point-particle mass. The event horizon is located at the largest root of $\Delta = r^2 - 2r + a^2$. We introduce the dimensionless proper time $\tau$ and define the redshift as $U = dt/d\tau$. We consider orbits in the equatorial plane of a Kerr black hole parametrized by $z^\mu = (t, r, \frac{\pi}{2}, \int \Omega dt)$ where $\Omega = d\phi/dt$ is the orbital frequency. We denote $\sigma = \text{sign}(\Omega)$, i.e. $\sigma = +1$ for prograde orbits and $\sigma = -1$ for retrograde orbits. Finally, the four-velocity is $v^\mu = dz^\mu/d\tau = (U, dr/d\tau, 0, U\Omega)$.

In terms of the specific particle energy $e = -p_t/m = v_t$ and angular momentum $\ell = p_\phi/(mM) = v_\phi$ one has

$$U = -g^{tt}e + g^{t\phi}\ell = \frac{r(r^2 + a^2)e + 2a(ae - \ell)}{r\Delta}, \tag{1}$$

$$\Omega = \frac{-g^{t\phi}e + g^{\phi\phi}\ell}{-g^{tt}e + g^{t\phi}\ell} = \frac{2(ae - \ell) + r\ell}{r(r^2 + a^2)e + 2a(ae - \ell)}. \tag{2}$$

The forced geodesic equation is given by

$$\frac{dv^\mu}{d\tau} + \Gamma^\mu_{\nu\sigma} v^\nu v^\sigma = f^\mu, \tag{3}$$

where $f^\mu$ is the gravitational self-force that obeys $f^\theta = 0$ on equatorial orbits. Note that we rescaled the geodesic equation with the point particle mass $m$. We define the slowly evolving helicoidal vector $\xi \equiv \partial_t + \Omega \partial_\phi$.

The normalization of the four-velocity for massive particles implies $g_{\mu\nu} v^\mu v^\nu = -1$ which gives the radial first order equation of motion

$$\left(\frac{dr}{d\tau}\right)^2 = g^{rr}\left(-1 - U^2 g_{\mu\nu} \xi^\mu \xi^\nu\right), \tag{4}$$

or, equivalently after using Eqs. (1) and (2),

$$\left(\frac{dr}{d\tau}\right)^2 = e^2 - V^{\text{geo}}, \qquad V^{\text{geo}}(r, e, \ell, a) \equiv e^2 - \frac{\left[e\left(r^2 + a^2\right) - a\ell\right]^2 - \Delta\left[r^2 + (ae - \ell)^2\right]}{r^4}. \tag{5}$$

The derivative along proper time of the normalization condition gives the orthogonality condition $f_\mu v^\mu = 0$ which can be written in the two equivalent ways:

$$f^r = -U g^{rr}(f_t + \Omega f_\phi)\left(\frac{dr}{d\tau}\right)^{-1}, \tag{6}$$

$$f^r = g^{rr}(ef^t - \ell f^\phi)\left(\frac{dr}{d\tau}\right)^{-1}. \tag{7}$$

The angular momentum $\ell$ and energy $e$ evolve according to the forced geodesic equation as

$$\frac{d\ell}{d\tau} = f_\phi, \qquad \frac{de}{d\tau} = -f_t, \tag{8}$$

allowing to rewrite the orthogonality condition (6) as

$$f^r = U g^{rr}\left(\frac{de}{d\tau} - \Omega\frac{d\ell}{d\tau}\right)\left(\frac{dr}{d\tau}\right)^{-1}. \tag{9}$$

Taking the $\tau$-derivative of Eq. (5) we obtain

$$\frac{d^2r}{d\tau^2} + \frac{1}{2}\frac{\partial V^{\text{geo}}}{\partial r} = f^r, \tag{10}$$

where

$$f^r = \left[\left(e - \frac{1}{2}\frac{\partial V^{\text{geo}}}{\partial e}\right)\frac{de}{d\tau} - \frac{1}{2}\frac{\partial V^{\text{geo}}}{\partial \ell}\frac{d\ell}{d\tau} - \frac{1}{2}\frac{\partial V^{\text{geo}}}{\partial a}\frac{da}{d\tau}\right]\left(\frac{dr}{d\tau}\right)^{-1}. \tag{11}$$

Equations (10) and (11) agree with the radial component of the forced geodesic equation (3),

$$\frac{d^2r}{d\tau^2} + \Gamma^r_{rr}\left(\frac{dr}{d\tau}\right)^2 + U^2 \Gamma^r_{\mu\nu} \xi^\mu \xi^\nu = f^r, \tag{12}$$

given the identity

$$\frac{1}{2}\frac{\partial V^{\text{geo}}}{\partial r} = \Gamma^r_{rr}(e^2 - V^{\text{geo}}) + U^2 \Gamma^r_{\mu\nu} \xi^\mu \xi^\nu. \tag{13}$$

Comparing the writings of the radial self-force in Eqs. (9) and (11) and using the identities $e - \frac{1}{2}\frac{\partial V^{\text{geo}}}{\partial e} = U g^{rr}$ and $\frac{1}{2}\frac{\partial V^{\text{geo}}}{\partial \ell} = \Omega U g^{rr}$, we deduce

$$\frac{da}{d\tau} = 0. \tag{14}$$

At early times, the inspiralling body has no causal contact with the central body and therefore it cannot change its angular momentum. From this causality argument we set $a(\tau) = a$ constant. In summary, the conservation of the normalization of the four-velocity along the forced geodesic motion fixes the evolution of the background $a$ to be constant along the motion at the location of the point particle. The spin of the central black hole is still evolving due to gravitational wave emission, which is encoded in perturbations of the background metric $h^{(1)}_{\mu\nu}$, $h^{(2)}_{\mu\nu}$, ... [37,38] and these effects appear in the self-force term $f^{\mu}$ of the forced geodesic equation.

It will be convenient to introduce $\delta$ as the deviation from the geodesic angular velocity for circular orbits $\Omega_{\text{geo}} = \sigma/(r^{3/2} + \sigma a)$ as

$$\delta \equiv \sigma\left(\Omega^{-1} - \Omega_{\text{geo}}^{-1}\right), \qquad \Omega = \frac{\sigma}{r^{3/2} + \sigma a + \delta}\,. \tag{15}$$

We collectively denote all variables of interest as

$$X \equiv (r, \delta, \Omega, U, e, \ell)\,. \tag{16}$$

## 3 Quasi-circular inspiral in the $n$th post-adiabatic expansion

We now restrict our analysis to the small mass ratio limit $\eta \ll 1$ and to orbits without eccentricity. Such orbits can be described in the inspiral phase using the slow proper time

$$\tilde{\tau} \equiv \eta\,\tau\,, \tag{17}$$

and the slow-timescale expansion

$$X = X_{(0)}(\tilde{\tau}) + \eta X_{(1)}(\tilde{\tau}) + \eta^2 X_{(2)}(\tilde{\tau}) + O_{\tilde{\tau}}(\eta^3)\,, \tag{18}$$

where the collective variable $X$ was defined in Eq. (16). Here and below, the indices in parentheses $(i)$ label the terms appearing at order $\eta^i$ in the expansion. The symbol $O_{\tilde{\tau}}(\eta)$ refers to the limit $\eta \to 0$ at fixed slow proper time $\tilde{\tau}$. We name the leading $X_{(0)}$ terms the adiabatic order or 0PA order. The $n$th subleading terms are denoted as $n$th post-adiabatic order or $n$PA order. We define similarly the slow Boyer-Lindquist time $\tilde{t} \equiv \eta\,t$. Equations (6) and (8) are consistent with the expansions

$$f^{\mu} = \eta f^{\mu}_{(1)}(\tilde{\tau}) + \eta^2 f^{\mu}_{(2)}(\tilde{\tau}) + O_{\tilde{\tau}}(\eta^3)\,. \tag{19}$$

Note that although we are taking into account the radial self-force $f^r$ in Eq. (9) the motion is quasi-circular in the sense that the radial and angular motions evolve on different timescales due to the expansion (18): indeed $d\phi/d\tau = O_{\tilde{\tau}}(1)$ while $dr/d\tau = O_{\tilde{\tau}}(\eta)$, so that

$$\frac{d\log r}{d\phi} = \frac{dr/d\tau}{r\,d\phi/d\tau} = O_{\tilde{\tau}}(\eta)\,. \tag{20}$$

We will algebraically solve the equations and write the remaining differential equations in the $n$th post-adiabatic expansion. Algebraic equations are obtained at order $\eta^n$ while first-order differential equations come from order $\eta^{n+1}$ due to the slow-time dependence of the quantitites $X = X(\tilde{\tau})$. As we will see in Sections 3.2, 3.3 and 3.4 the motion is governed by one differential equation controlling the evolution of the radial coordinate $r_{(n)}$. All other quantities are algebraically determined.

## 3.1 Structure of the self-force

Quasi-circular orbits are entirely parametrized by the proper time along the orbit. Since the Boyer-Linquist radius is monotonically decreasing in proper time, one can parametrize as well any quantity along the orbit in terms of the radius. In particular, we can write the self-force as

$$f^\mu = \sum_{n=1}^\infty \eta^n F_n^\mu(r), \tag{21}$$

where $F_n^\mu$ are functions of $r$ only. Using the slow-timescale expansion (18) we can expand the self-force around the adiabatic order $r = r_{(0)}$,

$$f^\mu = \eta\, F_1^\mu(r_{(0)}) + \eta^2 \left( F_2^\mu(r_{(0)}) + \frac{dF_1^\mu}{dr}\bigg|_{(0)} r_{(1)} \right)$$
$$+ \eta^3 \left( F_3^\mu(r_{(0)}) + \frac{dF_1^\mu}{dr}\bigg|_{(0)} r_{(2)} + \frac{1}{2}\frac{d^2 F_1^\mu}{dr^2}\bigg|_{(0)} r_{(1)}^2 \right) + O(\eta^4). \tag{22}$$

Comparing with Eq. (19) we obtain

$$f_{(1)}^\mu = F_1^\mu(r_{(0)}), \tag{23}$$

$$f_{(2)}^\mu = F_2^\mu(r_{(0)}) + \frac{dF_1^\mu}{dr}\bigg|_{(0)} r_{(1)}, \tag{24}$$

$$f_{(3)}^\mu = F_3^\mu(r_{(0)}) + \frac{dF_1^\mu}{dr}\bigg|_{(0)} r_{(2)} + \frac{1}{2}\frac{d^2 F_1^\mu}{dr^2}\bigg|_{(0)} r_{(1)}^2, \qquad \cdots \tag{25}$$

At order $n$ the self-force will be composed of a term linear in $r_{(n-1)}$ and non-linear terms $NL_{(n-1)}$ homogeneous of degree $n-1$ in the mass ratio,

$$f_{(n)}^\mu = f_{(1)}^{\mu,\text{lin}}(r_{(0)}) r_{(n-1)} + f_{(n)}^{\mu,\text{non-lin}}\left( \{r_{(k)}\}_{k=0,1,2,\dots,n-2} \right), \tag{26a}$$

$$f_{(1)}^{\mu,\text{lin}} \equiv \frac{dF_1^\mu}{dr}\bigg|_{(0)}, \qquad f_{(n)}^{\mu,\text{non-lin}} \equiv NL_{(n-1)}\left( \{r_{(k)}\}_{k=0,1,2,\dots,n-2} \right). \tag{26b}$$

## 3.2 Adiabatic inspiral

The adiabatic solution without eccentricity to Eqs. (1), (2), (4), (5), (8), (10), (12) and (15) can be found straightforwardly. Equation (9) implies an exact adiabatic quasi-circular inspiral:

$$\frac{de_{(0)}}{d\tilde{\tau}} - \Omega_{(0)}\frac{d\ell_{(0)}}{d\tilde{\tau}} = 0. \tag{27}$$

From Eq. (7) we obtain

$$e_{(0)}f_{(1)}^t - \ell_{(0)}f_{(1)}^\phi = 0. \tag{28}$$

In order to write compact expressions, it is convenient to define the following functions of $r_{(0)}$:

$$A = r_{(0)}^3 - 3r_{(0)}^2 + 2\sigma a r_{(0)}^{3/2}, \qquad B = r_{(0)}^2 - 2\sigma a r_{(0)}^{1/2} + a^2, \tag{29a}$$

$$C = r_{(0)}^{3/2} - 2r_{(0)}^{1/2} + \sigma a, \qquad D = 4Ar_{(0)}^{-1} - 3\Delta_{(0)}, \tag{29b}$$

where $\Delta_{(0)} \equiv \Delta|_{(0)} = r_{(0)}^2 - 2r_{(0)} + a^2$. We note that the function $D$ of $r_{(0)}$ admits a single root outside the horizon, which occurs at the location of the geodesic innermost stable circular orbit (ISCO) $r_{(0)} = r_{(0)*}$

$$D_* \equiv D|_* = r_{(0)*}^2 - 6r_{(0)*} + 8\sigma a\sqrt{r_{(0)*}} - 3a^2 = 0. \tag{30}$$

The root of this function is of particular interest as it appears in the denominator of the equations of motion (32), (38) and (44) causing the slow-timescale expansion to break down at the location of the ISCO. The roots of the function $B$, which also appears in the same denominators that $D$ does, lie behind the event horizon.

The algebraic part of the resolution is standard and we shall therefore be concise. Equation (12) at order $\eta^0$ gives $\Gamma_{\mu\nu}^r \xi_{(0)}^\mu \xi_{(0)}^\nu = 0$ with $\xi_{(0)} = \partial_t + \Omega_{(0)}\partial_\phi$. This yields $\Omega_{(0)} = \sigma(r_{(0)}^{3/2} + \sigma a)^{-1}$. The redshift $U_{(0)} = \sigma A^{-1/2}\Omega_{(0)}^{-1}$ is computed from Eq. (4). We can then solve Eqs. (1) and (2) to obtain $e_{(0)} = e_{(0)}(r_{(0)})$ and $\ell_{(0)} = \ell_{(0)}(r_{(0)})$. The unique solution to Eqs. (1), (2), (4) and (12) at order $\eta^0$ can then be written as

$$\Omega_{(0)}(\tilde\tau) = \Omega_{(0)}[r_{(0)}(\tilde\tau)] \equiv \sigma(r_{(0)}^{3/2} + \sigma a)^{-1}, \tag{31a}$$

$$U_{(0)}(\tilde\tau) = U_{(0)}[r_{(0)}(\tilde\tau)] \equiv \sigma A^{-1/2}\Omega_{(0)}^{-1} = A^{-1/2}|\Omega_{(0)}|^{-1}, \tag{31b}$$

$$\ell_{(0)}(\tilde\tau) = \ell_{(0)}[r_{(0)}(\tilde\tau)] \equiv \sigma B A^{-1/2}, \tag{31c}$$

$$e_{(0)}(\tilde\tau) = e_{(0)}[r_{(0)}(\tilde\tau)] \equiv C A^{-1/2}. \tag{31d}$$

Then, from Eq. (15) at adiabatic order we obtain $\delta_{(0)} = 0$. Note that a $\mathbb{Z}_2$ "parity" symmetry exists for these equations when the signs of both $\sigma$ and $a$ are flipped. Each variable is either even or odd under parity, e.g. $\sigma, a, \ell_{(0)}, \Omega_{(0)}$ are odd while $U_{(0)}, e_{(0)}$ are even.

The slow time evolution of $r_{(0)}$ is then deduced from either the $t$ or $\phi$ components of the geodesic equation at linear order in $\eta$ as

$$\frac{dr_{(0)}}{d\tilde\tau} = \frac{2\Delta_{(0)}A^{3/2}}{r_{(0)}^{1/2}BD}f_{(1)}^t. \tag{32}$$

Since $D$ appears in the denominator, the solution breaks down at the ISCO. In the Schwarzschild case ($a = 0, \sigma = 1$), this equation reduces to Eq. (103) of [12] after using the conversion from slow proper time to slow coordinate time, $d\tilde\tau = U_{(0)}^{-1}d\tilde t$.

### 3.3 1-post-adiabatic inspiral

In this section we obtain the equations for the quasi-circular inspiral at first post-adiabatic (1PA) order. The redshift and the orbital frequency are computed from Eqs. (4) and (12), respectively,

$$U_{(1)} = \frac{\sigma B}{\Omega_{(0)}^2 A^{3/2}}\Omega_{(1)}, \qquad \Omega_{(1)} = -\sigma\frac{r_{(0)}^{5/2}\Omega_{(0)}^2 A}{2\Delta_{(0)}}f_{(1)}^r - \frac{3\sigma r_{(0)}^{1/2}\Omega_{(0)}^2}{2}r_{(1)}. \tag{33}$$

In the Schwarzschild case, we recover Eq. (104) of [12].

One can use Eq. (5) to relate the 1PA corrections of the energy and the angular momentum, and, use Eq. (2) to obtain $\ell_{(1)}$,

$$e_{(1)} = \Omega_{(0)}\ell_{(1)}, \qquad \ell_{(1)} = \frac{r_{(0)}^{1/2}}{2\Omega_{(0)}A^{1/2}}\left(\frac{D}{A}r_{(1)} - r_{(0)}^2 f_{(1)}^r\right). \tag{34}$$

Equation (1) is then obeyed. It is straightforward to compute the first-order deviation $\delta_{(1)}$ from the radial self-force $f_{(1)}^r$ using Eqs. (15) and (33),

$$\delta_{(1)} = \frac{r_{(0)}^{5/2} A}{2\Delta_{(0)}} f_{(1)}^r .\tag{35}$$

From Eq. (7) we obtain

$$f_{(2)}^\phi = \frac{e_{(0)}}{\ell_{(0)}} f_{(2)}^t - \sigma \left( \frac{3\Delta_{(0)}^2}{B^2 D} \delta_{(1)} + \frac{r_{(0)}^{1/2} D}{2B^2} r_{(1)} \right) f_{(1)}^t .\tag{36}$$

Following Eq. (26), the second-order self-force can be decomposed as

$$f_{(2)}^\mu = r_{(1)} f_{(1)}^{\mu,\text{lin}}(r_{(0)}) + f_{(2)}^{\mu,\text{non-lin}}(r_{(0)}),\tag{37}$$

where $f_{(1)}^{\mu,\text{lin}}(r_{(0)})$ and $f_{(2)}^{\mu,\text{non-lin}}(r_{(0)})$ are retarded functionals of $r_{(0)}$. This form is compatible with Eq. (209) of [12].

The $t$-component of the geodesic equation finally gives the slow time evolution of $r_{(1)}$,

$$\begin{aligned}
\frac{dr_{(1)}}{d\tilde{\tau}} - \frac{A^{1/2} T_1}{B^2 D^2} f_{(1)}^t r_{(1)} - \frac{2A^{3/2} \Delta_{(0)}}{r_{(0)}^{1/2} B D} f_{(1)}^{t,\text{lin}} r_{(1)} &= S_{(1)}^r \\
&\equiv \frac{2A^{3/2} \Delta_{(0)}}{r_{(0)}^{1/2} B D} f_{(2)}^{t,\text{non-lin}} + \frac{r_{(0)}^2 A}{D} \frac{df_{(1)}^r}{d\tilde{\tau}} + \frac{r_{(0)}^2 A^{3/2} \Delta_{(0)} T_2}{B^2 D^2} f_{(1)}^t f_{(1)}^r .
\end{aligned}\tag{38}$$

The functions $T_1$ and $T_2$ are given in Appendix A. In the Schwarzschild case, this equation is compatible with Eqs. (103) and (106) of [12] after using the conversion from slow proper time to slow coordinate time, $\frac{dr_{(1)}}{d\tilde{\tau}} = U_{(0)} \frac{dr_{(1)}}{d\tilde{t}} + U_{(1)} \frac{dr_{(0)}}{d\tilde{t}}$.

## 3.4 2-post-adiabatic inspiral

In this section we derive the equations for the quasi-circular inspiral at second post-adiabatic (2PA) order keeping in mind the 0PA and 1PA results.

The redshift and the orbital frequency are computed from Eqs. (4) and (12), respectively,

$$\begin{aligned}
U_{(2)} =& \frac{\sigma B}{\Omega_{(0)}^2 A^{3/2}} \Omega_{(2)} - \frac{3\sigma r_{(0)} \Omega_{(0)} T_3}{8A^{5/2}} r_{(1)}^2 + \frac{2\sigma r_{(0)} A^{5/2} \Delta_{(0)}}{\Omega_{(0)} B^2 D^2} \left( f_{(1)}^t \right)^2 \\
& + \frac{\sigma r_{(0)}^5 \Omega_{(0)} T_4}{8A^{1/2} \Delta_{(0)}^2} \left( f_{(1)}^r \right)^2 - \frac{\sigma r_{(0)}^3 \Omega_{(0)} T_3}{4A^{3/2} \Delta_{(0)}} r_{(1)} f_{(1)}^r ,
\end{aligned}\tag{39}$$

$$\begin{aligned}
\Omega_{(2)} =& -\sigma \frac{r_{(0)}^{5/2} \Omega_{(0)}^2 A}{2\Delta_{(0)}} f_{(2)}^r - \frac{3\sigma r_{(0)}^{1/2} \Omega_{(0)}^2}{2} r_{(2)} - \frac{3\Omega_{(0)}^3 \left( \sigma a - 5r_{(0)}^{3/2} \right)}{8r_{(0)}^{1/2}} r_{(1)}^2 + \sigma \frac{r_{(0)}^2 \Omega_{(0)}^2 A^3 T_5}{B^3 D^3} \left( f_{(1)}^t \right)^2 \\
& - \frac{r_{(0)}^{13/2} \Omega_{(0)}^3 T_6}{8\Delta_{(0)}^2} \left( f_{(1)}^r \right)^2 - \frac{r_{(0)}^3 \Omega_{(0)}^3 T_7}{4\Delta_{(0)}^2} r_{(1)} f_{(1)}^r + \sigma \frac{r_{(0)}^2 \Omega_{(0)}^2 A^{5/2}}{B D} \frac{df_{(1)}^t}{d\tilde{\tau}} .
\end{aligned}\tag{40}$$

All auxiliary functions $T_i$, $i = 3, 4, \ldots, 20$ necessary to write the inspiral equations are given in Appendix A. One can use Eqs. (5) and (2) to write the 2PA corrections to the energy and the

angular momentum, respectively,

$$
\begin{aligned}
e_{(2)} = {} & \Omega_{(0)}\ell_{(2)} - \frac{3\sigma r_{(0)}\Omega_{(0)}D}{8A^{3/2}}r_{(1)}^2 + \frac{2\sigma r_{(0)}\Omega_{(0)}A^{7/2}\Delta_{(0)}}{B^2 D^2}\left(f_{(1)}^t\right)^2 \\
& + \frac{\sigma r_{(0)}^5 \Omega_{(0)}A^{1/2}}{8\Delta_{(0)}}\left(f_{(1)}^r\right)^2 + \frac{3\sigma r_{(0)}^3 \Omega_{(0)}}{4A^{1/2}}r_{(1)}f_{(1)}^r,
\end{aligned}
\tag{41}
$$

$$
\begin{aligned}
\ell_{(2)} = {} & \frac{r_{(0)}^{1/2}}{2\Omega_{(0)}A^{1/2}}\left(\frac{D}{A}r_{(2)} - r_{(0)}^2 f_{(2)}^r\right) + \frac{\sigma r_{(0)}T_8}{8A^{5/2}}r_{(1)}^2 + \frac{\sigma r_{(0)}^2 A^{3/2}\Delta_{(0)}T_9}{B^3 D^3}\left(f_{(1)}^t\right)^2 \\
& - \frac{\sigma r_{(0)}^5 T_{10}}{8A^{1/2}\Delta_{(0)}}\left(f_{(1)}^r\right)^2 - \frac{\sigma r_{(0)}^3 T_{11}}{4A^{3/2}}r_{(1)}f_{(1)}^r + \frac{r_{(0)}^2 A\Delta_{(0)}}{\Omega_{(0)}B D}\frac{df_{(1)}^t}{d\tilde{\tau}}.
\end{aligned}
\tag{42}
$$

Equation (1) is then obeyed. We compute the second-order deviation $\delta_{(2)}$ using the definition (15) and Eqs. (33) and (40),

$$
\delta_{(2)} = \frac{r_{(0)}^{5/2}A}{2\Delta_{(0)}}f_{(2)}^r - \frac{r_{(0)}^2 A^3 T_5}{B^3 D^3}\left(f_{(1)}^t\right)^2 + \frac{r_{(0)}^5 A T_{12}}{8\Delta_{(0)}^2}\left(f_{(1)}^r\right)^2 + \frac{r_{(0)}^3 T_{13}}{4\Delta_{(0)}^2}r_{(1)}f_{(1)}^r - \frac{r_{(0)}^2 A^{5/2}}{B D}\frac{df_{(1)}^t}{d\tilde{\tau}}.
\tag{43}
$$

Equation (7) can be used to deduce $f_{(3)}^{\phi}$ in terms of $f_{(3)}^t$ and zeroth and first-order quantities. The $t$-component of geodesic equation finally gives the slow time evolution of $r_{(2)}$,

$$
\frac{dr_{(2)}}{d\tilde{\tau}} - \frac{A^{1/2}T_1}{B^2 D^2}f_{(1)}^t r_{(2)} - \frac{2A^{3/2}\Delta_{(0)}}{r_{(0)}^{1/2}B D}f_{(1)}^{t,\mathrm{lin}}r_{(2)} = S_{(2)}^r,
\tag{44}
$$

$$
\begin{aligned}
S_{(2)}^r \equiv {} & \frac{2A^{3/2}\Delta_{(0)}}{r_{(0)}^{1/2}B D}f_{(3)}^{t,\mathrm{non\text{-}lin}} + \frac{r_{(0)}^2 A}{D}\frac{df_{(2)}^r}{d\tilde{\tau}} - \frac{2r_{(0)}^{3/2}A^{5/2}\Delta_{(0)}}{B D^2}\frac{d^2 f_{(1)}^t}{d\tilde{\tau}^2} + \frac{A^{1/2}T_1}{B^2 D^2}r_{(1)}f_{(2)}^t \\
& + \frac{r_{(0)}^2 A^{3/2}\Delta_{(0)}T_2}{B^2 D^2}\left(f_{(1)}^t f_{(2)}^r + f_{(2)}^t f_{(1)}^r\right) - \frac{4r_{(0)}^{3/2}A^3\Delta_{(0)}T_{14}}{B^3 D^4}f_{(1)}^t\frac{df_{(1)}^t}{d\tilde{\tau}} \\
& + \frac{r_{(0)}^{9/2}A T_{15}}{\Delta_{(0)}B D^2}f_{(1)}^r\frac{df_{(1)}^r}{d\tilde{\tau}} - \frac{3T_{16}}{D^2}r_{(1)}\frac{df_{(1)}^r}{d\tilde{\tau}} - \frac{2r_{(0)}^{3/2}A^{7/2}\Delta_{(0)}T_{17}}{B^5 D^6}\left(f_{(1)}^t\right)^3 \\
& + \frac{r_{(0)}^{9/2}A^{3/2}T_{18}}{4\Delta_{(0)}B^3 D^3}f_{(1)}^t\left(f_{(1)}^r\right)^2 + \frac{r_{(0)}^{1/2}T_{19}}{4A^{1/2}B^3 D^3}r_{(1)}^2 f_{(1)}^t + \frac{r_{(0)}^{5/2}A^{1/2}T_{20}}{2B^3 D^3}r_{(1)}f_{(1)}^t f_{(1)}^r,
\end{aligned}
\tag{45}
$$

where we have used the writing (26) of the third-order self-force.

## 3.5 n-post-adiabatic inspiral

We follow the same procedure as the one described in the previous sections. Equation (4) is first solved for $U_{(n)}$ while $\Omega_{(n)}$ is computed from Eq. (12). One then solves Eqs. (5) and (2) to obtain the $n$PA corrections to the energy and angular momentum. Using the angular velocity and the definition (15) one can compute $\delta_{(n)}$. Equation (7) can be used to deduce $f_{(n+1)}^{\phi}$ in terms of $f_{(n+1)}^t$ and lower order quantities. The self-force at order $(n+1)$ can be written using the decomposition (26). The slow time evolution of $r_{(n)}$ is then be obtained from the $t$-component of the geodesic equation at order $\eta^{n+1}$. It has the following structure

$$
\frac{dr_{(n)}}{d\tilde{\tau}} - \frac{A^{1/2}T_1}{B^2 D^2}f_{(1)}^t r_{(n)} - \frac{2A^{3/2}\Delta_{(0)}}{r_{(0)}^{1/2}B D}f_{(1)}^{t,\mathrm{lin}}r_{(n)} = S_{(n)}^r,
\tag{46}
$$

where $S_{(n)}^r$ is the source term.

### 3.6 Inspiral towards the last stable orbit

We will now consider the inspiral motion close to the last stable orbit (LSO) first in the adiabatic approximation (0PA), then including subleading corrections (1PA), and we will comment upon arbitrary high subleading orders ($n$PA). In this limit the quasi-circular inspiral is expected to smoothly match with the transition-to-plunge motion. We will denote the slow proper time at the LSO as $\tilde{\tau}_*$, which is either prograde or retrograde depending upon the sign of $\sigma$. The last stable orbit is defined from the radius $r$ where

$$\frac{\partial^2 V^{\text{geo}}(e, \ell, r, a)}{\partial r^2} = \frac{2D}{r_{(0)}^2 A} + O_{\tilde{\tau}}(\eta) = 0, \tag{47}$$

after using $\ell = \ell_{(0)}$ and $e = e_{(0)}$ as given in Eqs. (31c) and (31d). At leading order in the small mass ratio expansion, the LSO therefore coincides with the ISCO since $D|_* = 0$. Hence, we can either use the terminology of LSO or ISCO at this order though both concepts differ once subleading corrections in the mass ratio are considered.

In the absence of radial self-force, as assumed for simplicity in the original Ori-Thorne analysis [19], Eq. (9) implies the exact quasi-circularity condition,

$$f^r = 0 \quad \Rightarrow \quad \frac{de}{d\tau} = \Omega \frac{d\ell}{d\tau}. \tag{48}$$

In reality, the normalization of the four-velocity gives the orthogonality condition $f_\mu v^\mu = 0$, precisely Eq. (9), which leads to a deviation from quasi-circularity, as originally observed by Kesden [21], see also [25]. The motion still remains quasi-circular in the sense of Eqs. (20) and (107) below. We will refer to a deviation from Eq. (48) as a deviation from exact quasi-circularity. In order to account for deviation from exact quasi-circularity close to the LSO, we define the quantity

$$W = \Omega_{(0)*}^{-1}(e - e_{(0)*}) - (\ell - \ell_{(0)*}), \tag{49}$$

which naturally arises in the transition motion and will be therefore helpful in the asymptotic matching. It admits the same slow-timescale expansion as (18),

$$W = \sum_{i=0}^{\infty} \eta^i W_{(i)}, \qquad W_{(i)} = \Omega_{(0)*}^{-1}(e_{(i)} - e_{(0)*}\delta_{i,0}) - (\ell_{(i)} - \ell_{(0)*}\delta_{i,0}), \tag{50}$$

where $\delta_{i,j}$ is the Kronecker delta.

#### 3.6.1 Adiabatic order

As demonstrated in [19, 21], the slow-timescale expansion breaks down at the ISCO in the absence of radial self-force corrections. Now, radial self-force corrections only occur starting from 1PA order, see Eq. (35). At 0PA order, the radial evolution (32) breaks down at the radial geodesic ISCO value $r_{(0)} = r_{(0)*}$ since $D_* = 0$ and other quantities remain finite and non-vanishing, which implies that $dr_{(0)}/d\tilde{\tau}$ blows up.

We consider the limit in which $\tilde{\tau} \to \tilde{\tau}_*$ and $r_{(0)} \to r_{(0)*}$ and expand the equation of motion (32) around $r_{(0)*}$. In particular

$$D = \frac{2A_*}{r_{(0)*}^2}\left(r_{(0)} - r_{(0)*}\right) + O\left(r_{(0)} - r_{(0)*}\right)^2. \tag{51}$$

From Eq. (23), the first-order self-force admits the Taylor expansion

$$f_{(1)}^\mu = \sum_{i=0} \left.\frac{d^i f_{(1)}^\mu}{dr_{(0)}^i}\right|_* \left(r_{(0)} - r_{(0)*}\right)^i = \sum_{i=0} \left.\frac{d^i F_1^\mu}{dr^i}\right|_{(0)*} \left(r_{(0)} - r_{(0)*}\right)^i. \tag{52}$$

The functional dependence of $F_1^\mu$ on $r$ is the same as the one on $r_{(0)}$. The subscript $(0)*$ indicates that the self-force is evaluated both at zeroth-order in its arguments and at the ISCO. The expansion of Eq. (32) around the ISCO then takes the form

$$\frac{dr_{(0)}}{d\tilde{\tau}} = \sum_{i=-1}^{\infty} h_{(0),i}^* \left(r_{(0)} - r_{(0)*}\right)^i , \tag{53}$$

where $h_{(0),-1}^* = \left.\frac{4r_{(0)*}^{1/2}A_*^{3/2}}{3B_*} f_{(1)}^t\right|_*$ after using the property

$$4A_* = 3r_{(0)*}\Delta_{(0)*} . \tag{54}$$

Note that $h_{(0),-1}^* < 0$ because angular momentum is lost by the radiation, which implies $\left.f_{(1)}^t\right|_* < 0$. The solution to the equation of motion is consistent with an expansion in half-integer powers of $(\tilde{\tau}_* - \tilde{\tau})$,

$$r_{(0)} = r_{(0)*} + \sum_{i=1}^{\infty} r_{(0),i}^* (\tilde{\tau}_* - \tilde{\tau})^{i/2} , \tag{55}$$

where

$$r_{(0),1}^* = \sqrt{-2h_{(0),-1}^*} = \sqrt{-\left.\frac{8r_{(0)*}^{1/2}A_*^{3/2}}{3B_*} f_{(1)}^t\right|_*} = \sqrt{-\left.\frac{8r_{(0)*}^{1/2}A_*^{3/2}}{3B_*} F_1^t\right|_{(0)*}} . \tag{56}$$

All $r_{(0),i}^*$ can be algebraically computed for any $i \geq 2$ in terms of $h_{(0),j}^*$, $j = -1, 0, 1, 2, \ldots$ The first such subleading terms are $r_{(0),2}^* = -\frac{2}{3}h_{(0),0}^*$ and $r_{(0),3}^* = \frac{(h_{(0),0}^*)^2 + 9h_{(0),-1}^* h_{(0),1}^*}{9\sqrt{2}(-h_{(0),-1}^*)^{1/2}}$.

We can now obtain the entire expansion close to the LSO of all adiabatic quantities after substituting the solution (55) into the constraints (31). We denote the adiabatic energy, angular momentum and angular velocity at the ISCO as $e_{(0)*} = C_*/\sqrt{A_*}$, $\ell_{(0)*} = \sigma B_*/\sqrt{A_*}$, $\Omega_{(0)*} = \sigma/(r_{(0)*}^{3/2} + \sigma a)$. We obtain

$$\ell_{(0)} = \ell_{(0)*} + \ell_{(0),2}^*(\tilde{\tau}_* - \tilde{\tau}) + \ell_{(0),3}^*(\tilde{\tau}_* - \tilde{\tau})^{3/2} + O(\tilde{\tau}_* - \tilde{\tau})^2 , \tag{57a}$$

$$e_{(0)} = e_{(0)*} + \Omega_{(0)*}\ell_{(0),2}^*(\tilde{\tau}_* - \tilde{\tau}) + e_{(0),3}^*(\tilde{\tau}_* - \tilde{\tau})^{3/2} + O(\tilde{\tau}_* - \tilde{\tau})^2 , \tag{57b}$$

$$\Omega_{(0)} = \Omega_{(0)*} + \Omega_{(0),1}^*(\tilde{\tau}_* - \tilde{\tau})^{1/2} + O(\tilde{\tau}_* - \tilde{\tau}), \tag{57c}$$

with

$$\ell_{(0),2}^* = \frac{(r_{(0),1}^*)^2}{2r_{(0)*}^{3/2}\Omega_{(0)*}A_*^{1/2}} , \tag{58}$$

$$\ell_{(0),3}^* = \frac{\sigma r_{(0)*}}{4A_*^{5/2}} \left(\frac{r_{(0)*}^{1/2}E_*}{4A_*}(r_{(0),1}^*)^2 + T_{15}^* r_{(0),2}^*\right) r_{(0),1}^* , \tag{59}$$

$$e_{(0),3}^* = \Omega_{(0)*}\left(\ell_{(0),3}^* - \sigma r_{(0)*}^{1/2}\Omega_{(0)*}r_{(0),1}^*\ell_{(0),2}^*\right) , \tag{60}$$

$$\Omega_{(0),1}^* = -\frac{3}{2}\sigma r_{(0)*}^{1/2}\Omega_{(0)*}^2 r_{(0),1}^* . \tag{61}$$

The coefficients $E_*$ and $T_8^* \equiv T_8|_*$ are given in Appendix A. Using the above expressions for $\ell_{(0)}$ and $e_{(0)}$ in the flux-balance equations (8), we note that the first-order self-forces are obtained

as

$$f_{(1)}^t = \sum_{i=0}^{\infty} f_{(1),i}^{t*}(\tilde{\tau}_* - \tilde{\tau})^{i/2}, \qquad f_{(1)}^{\phi} = \sum_{i=0}^{\infty} f_{(1),i}^{\phi*}(\tilde{\tau}_* - \tilde{\tau})^{i/2}. \tag{62}$$

which are equivalent to Eq. (52). The lowest order terms are $f_{(1),0}^{t*} = -\frac{\Omega_{(0)*}B_*}{\Delta_{(0)*}}\ell_{(0),2}^*$ and $f_{(1),0}^{\phi*} = -\frac{\sigma\Omega_{(0)*}C_*}{\Delta_{(0)*}}\ell_{(0),2}^*$.

We deduce from the solution (57) that the deviation from exact quasi-circularity (49) is

$$W_{(0)} = \Omega_{(0)*}^{-1}(e_{(0)} - e_{(0)*}) - (\ell_{(0)} - \ell_{(0)*}) = -\sigma r_{(0)*}^{1/2}\Omega_{(0)*}r_{(0),1}^*\ell_{(0),2}^*(\tilde{\tau}_* - \tilde{\tau})^{3/2} + O(\tilde{\tau}_* - \tilde{\tau})^2. \tag{63}$$

Let us now obtain expressions for $r_{(0),1}^*$, $r_{(0),2}^*$, $\ell_{(0),2}^*$ and $\ell_{(0),3}^*$ which will allow to easily match the transition motion. Inverting Eqs. (58) and (59) we can rewrite $r_{(0),1}^*$ and $r_{(0),2}^*$ as

$$r_{(0),1}^* = \sqrt{2r_{(0)*}^{3/2}\Omega_{(0)*}A_*^{1/2}\ell_{(0),2}^*}, \tag{64}$$

$$r_{(0),2}^* = -\frac{1}{T_{15}^*}\left(\frac{r_{(0)*}^{1/2}E_*}{4A_*}\left(r_{(0),1}^*\right)^2 - \frac{4A_*^{5/2}}{r_{(0)*}r_{(0),1}^*}\ell_{(0),3}^*\right). \tag{65}$$

One can compute $\ell_{(0)}$ also by using the flux-balance equations (8) and a Taylor expansion as in Eq. (52) for $f_\phi^{(1)}$. Then, comparing with Eq. (57a), the coefficients $\ell_{(0),2}^*$ and $\ell_{(0),3}^*$ are obtained as

$$\ell_{(0),2}^* = -f_\phi^{(1)}\Big|_* = -F_{\phi\,1}\Big|_{(0)*}, \tag{66}$$

$$\ell_{(0),3}^* = -\frac{2}{3}\frac{df_\phi^{(1)}}{dr_{(0)}}\Big|_* r_{(0),1}^* = -\frac{2}{3}\frac{dF_{\phi\,1}}{dr}\Big|_{(0)*} r_{(0),1}^*. \tag{67}$$

### 3.6.2 1-post-adiabatic order

The variables at 1PA order are $(r_{(1)}, \Omega_{(1)}, e_{(1)}, \ell_{(1)}, \delta_{(1)}, W_{(1)})$ and the self-force components appearing at 1PA order are $f_{(2)}^{t,\phi}$ and $f_{(1)}^r$. We will now deduce the behavior of these fields close to the LSO. Given the expansion (52) and the solution (55), the first-order radial self-force $f_{(1)}^r$ can be expanded as

$$f_{(1)}^r = \sum_{i=0}^{\infty} f_{(1),i}^{r*}(\tilde{\tau}_* - \tilde{\tau})^{i/2}, \tag{68}$$

where the coefficients $f_{(1),i}^{r*}$ can be written in terms of the Taylor coefficients of Eq. (52), that is,

$$f_{(1),0}^{r*} = F_1^r\Big|_{(0)*}, \qquad f_{(1),1}^{r*} = \frac{dF_1^r}{dr}\Big|_{(0)*} r_{(0),1}^*, \qquad \cdots \tag{69}$$

The second-order self-forces, on the other hand, depend non-linearly on $r_{(0)}$ and at most linearly on $r_{(1)}$. They can be written as in Eq. (37). Given that they only depend on $r_{(0)}$, the

functions $f_{(1)}^{\mu,\text{lin}}$ and $f_{(2)}^{\mu,\text{non-lin}}$ admit the expansion

$$f_{(1)}^{\mu,\text{lin}} = \sum_{i=0}^{\infty} f_{(1),i}^{\mu,\text{lin}*}(\tilde{\tau}_* - \tilde{\tau})^{i/2} \quad \text{where} \quad f_{(1),0}^{\mu,\text{lin}*} = \left.\frac{dF_1^{\mu}}{dr}\right|_{(0)*}, \tag{70a}$$

$$f_{(1),1}^{\mu,\text{lin}*} = \left.\frac{d^2 F_1^{\mu}}{dr^2}\right|_{(0)*} r_{(0),1}^*, \qquad \cdots \tag{70b}$$

$$f_{(2)}^{\mu,\text{non-lin}} = \sum_{i=0}^{\infty} f_{(2),i}^{\mu,\text{non-lin}*}(\tilde{\tau}_* - \tilde{\tau})^{i/2} \quad \text{where} \quad f_{(2),0}^{\mu,\text{non-lin}*} = F_2^{\mu}\big|_{(0)*}, \tag{70c}$$

$$f_{(2),1}^{\mu,\text{non-lin}*} = \left.\frac{dF_2^{\mu}}{dr}\right|_{(0)*} r_{(0),1}^*, \qquad \cdots \tag{70d}$$

Using these expansions together with the adiabatic results the differential equation for $r_{(1)}$ (38) takes the schematic form

$$\frac{dr_{(1)}}{d\tilde{\tau}} + r_{(1)} \sum_{i=-2}^{\infty} h_{(1),i}^*(\tilde{\tau}_* - \tilde{\tau})^{i/2} = \sum_{j=-2}^{\infty} S_{(1),j}^{r*}(\tilde{\tau}_* - \tilde{\tau})^{i/2}, \tag{71}$$

where $S_{(1),j}^{r*}$ are the LSO-coefficients of the source $S_{(1)}^r(\tilde{\tau})$ and $h_{(1),i}^*$ are coefficients at the LSO. In particular, $h_{(1),-2}^* = \frac{3r_{(0)*}^{7/2} T_1^*}{32 A_*^3 B_*} = -\frac{1}{2}$. The homogeneous solution is then

$$r_{(1)}^{\text{h}}(\tilde{\tau}) = \sum_{i=-1}^{\infty} q_{(1),i}^*(\tilde{\tau}_* - \tilde{\tau})^{i/2}. \tag{72}$$

The first coefficient $q_{(1),-1}^*$ is currently left arbitrary. It will be fixed from Eqs. (78) and (90) below. The subleading coefficients are easily found in terms of the coefficients $h_{(1),i}^*$ by plugging the solution (72) back into the homogeneous equation (71). We find for $i \geq 0$,

$$q_{(1),i}^* = \frac{2}{i+1} \sum_{j=0}^{i} q_{(1),j-1}^* h_{(1),i-j-1}^*. \tag{73}$$

We compute the particular solution using the method of variation of constants. We seek a solution of the form

$$r_{(1)}^{\text{p}}(\tilde{\tau}) = k_{(1)}(\tilde{\tau}) r_{(1)}^{\text{h}}(\tilde{\tau}). \tag{74}$$

Substituting it into Eq. (71) we obtain that $k_{(1)}(\tilde{\tau})$ is given by

$$k_{(1)}(\tilde{\tau}) = \int^{\tilde{\tau}} \frac{S_{(1)}^r(\tilde{\tau}')}{r_{(1)}^{\text{h}}(\tilde{\tau}')} d\tilde{\tau}' = \int^{\tilde{\tau}} \sum_{i=0}^{\infty} S_{(1),i-2}^{r*}(\tilde{\tau}_* - \tilde{\tau}')^{i/2-1} \sum_{j=0}^{\infty} q_{(1),j+1}^{*(-)}(\tilde{\tau}_* - \tilde{\tau}')^{(j+1)/2} d\tilde{\tau}'$$

$$= \int^{\tilde{\tau}} \sum_{n=0}^{\infty} \sum_{i=0}^{n} S_{(1),i-2}^{r*} q_{(1),n-i+1}^{*(-)}(\tilde{\tau}_* - \tilde{\tau})^{(n-1)/2} d\tilde{\tau}' \tag{75}$$

$$= \sum_{n=1}^{\infty} \left( -\frac{2}{n} \sum_{i=0}^{n-1} S_{(1),i-2}^{r*} q_{(1),n-i}^{*(-)} \right)(\tilde{\tau}_* - \tilde{\tau})^{n/2} \equiv \sum_{n=1}^{\infty} k_{(1),n}^*(\tilde{\tau}_* - \tilde{\tau})^{n/2}.$$

Here $q_{(1),i}^{*(-)}$ are the coefficients of the reciprocal of the homogeneous solution, where $q_{(1),1}^{*(-)} = 1/q_{(1),-1}^{*}$ and

$$q_{(1),i+1}^{*(-)} = -\sum_{j=1}^{i} \frac{q_{(1),j-1}^{*} q_{(1),i-j+1}^{*(-)}}{q_{(1),-1}^{*}}, \qquad \forall i \geq 1. \tag{76}$$

Finally, we obtain

$$
\begin{aligned}
r_{(1)}^{\mathrm{p}}(\tilde{\tau}) &= \sum_{n=1}^{\infty} k_{(1),n}^{*} (\tilde{\tau}_{*} - \tilde{\tau})^{n/2} \sum_{j=-1}^{\infty} q_{(1),j}^{*} (\tilde{\tau}_{*} - \tilde{\tau})^{j/2} \\
&= \sum_{i=0}^{\infty} \sum_{n=0}^{i} k_{(1),n+1}^{*} q_{(1),i-n-1}^{*} (\tilde{\tau}_{*} - \tilde{\tau})^{i/2} \\
&= \sum_{i=0}^{\infty} \left( \sum_{n=0}^{i} q_{(1),i-n-1}^{*} \frac{-2}{n+1} \sum_{\ell=0}^{n} S_{(1),\ell-2}^{r*} q_{(1),n-\ell+1}^{*(-)} \right) (\tilde{\tau}_{*} - \tilde{\tau})^{i/2} \\
&\equiv \sum_{i=0}^{\infty} p_{(1),i}^{*} (\tilde{\tau}_{*} - \tilde{\tau})^{i/2}.
\end{aligned}
\tag{77}
$$

We have $p_{(1),0}^{*} = -2 S_{(1),-2}^{r*}$. We can finally write the asymptotic solution for $r_{(1)}$ as $\tilde{\tau} \to \tilde{\tau}_{*}$ as

$$r_{(1)} = \sum_{i=-1}^{\infty} r_{(1),i}^{*} (\tilde{\tau}_{*} - \tilde{\tau})^{i/2}, \qquad r_{(1),-1}^{*} = q_{(1),-1}^{*}, \qquad r_{(1),i}^{*} = q_{(1),i}^{*} + p_{(1),i}^{*}, \quad \forall i \geq 0. \tag{78}$$

The energy $e_{(1)}$, angular momentum $\ell_{(1)}$, angular velocity $\Omega_{(1)}$ and deviations $\delta_{(1)}$ and $W_{(1)}$ are obtained from the solutions to the constraints at 1PA order. Given Eq. (78), they can be expanded as

$$\ell_{(1)} = \sum_{i=0}^{\infty} \ell_{(1),i}^{*} (\tilde{\tau}_{*} - \tilde{\tau})^{i/2}, \tag{79}$$

$$e_{(1)} = \sum_{i=0}^{\infty} e_{(1),i}^{*} (\tilde{\tau}_{*} - \tilde{\tau})^{i/2}, \tag{80}$$

$$\Omega_{(1)} = \sum_{i=-1}^{\infty} \Omega_{(1),i}^{*} (\tilde{\tau}_{*} - \tilde{\tau})^{i/2}, \tag{81}$$

$$\delta_{(1)} = \sum_{i=0}^{\infty} \delta_{(1),i}^{*} (\tilde{\tau}_{*} - \tilde{\tau})^{i/2}, \tag{82}$$

$$W_{(1)} = \sum_{i=1}^{\infty} w_{(1),i}^{*} (\tilde{\tau}_{*} - \tilde{\tau})^{i/2}, \tag{83}$$

where the lowest order coefficients are

$$\ell^*_{(1),0} = -\frac{r^{5/2}_{(0)*}}{2\Omega_{(0)*}A^{1/2}_*}f^{r*}_{(1),0} + \frac{r_{(0)*}T^*_{15}}{4A^{5/2}_*}r^*_{(0),1}r^*_{(1),-1}, \tag{84}$$

$$e^*_{(1),0} = \Omega_{(0)*}\ell^*_{(1),0}, \tag{85}$$

$$\Omega^*_{(1),-1} = -\frac{3r^{1/2}_{(0)*}\Omega^2_{(0)*}}{2}r^*_{(1),-1}, \tag{86}$$

$$\delta^*_{(1),0} = \frac{r^{5/2}_{(0)*}A_*}{2\Delta_{(0)*}}f^{r*}_{(1),0}, \tag{87}$$

$$w^*_{(1),1} = \frac{3r^{3/2}_{(0)*}}{4A^{1/2}_*}\left(\frac{F_*}{4A^2_*}\left(r^*_{(0),1}\right)^2 r^*_{(1),-1} + r^{3/2}_{(0)*}r^*_{(0),1}f^{r*}_{(1),0}\right), \tag{88}$$

$$w^*_{(1),2} = \frac{3r^3_{(0)*}}{4A^{1/2}_*}\left(r^*_{(0),1}f^{r*}_{(1),1} + r^*_{(0),2}f^{r*}_{(1),0}\right) + \frac{3r^{3/2}_{(0)*}F_*}{16A^{5/2}_*}\left(\left(r^*_{(0),1}\right)^2 r^*_{(1),0} + 2r^*_{(0),1}r^*_{(0),2}r^*_{(1),-1}\right) +$$
$$+ G_*\left(r^*_{(1),0}\right)^2 f^{r*}_{(1),0} + H_*\left(r^*_{(0),1}\right)^3 r^*_{(1),-1}. \tag{89}$$

The functions $F_*$, $G_*$ and $H_*$ are given in Appendix A.

The above expansion is compatible with imposing the condition $\ell^*_{(1),0} = 0$, which implies from Eqs. (84) and (85) that $e^*_{(1),0} = 0$ and

$$r^*_{(1),-1} = \frac{2r^{3/2}_{(0)*}A^2_*}{r^*_{(0),1}\Omega_{(0)*}T^*_{15}}f^{r*}_{(1),0}. \tag{90}$$

We will show that this further restriction is consistent with the transition motion in the matching performed in Section 5.3.

Using the above expressions for $\ell_{(1)}$ and $e_{(1)}$ in the flux-balance equations (8), one can rewrite the second-order self-forces as

$$f^t_{(2)} = \sum_{i=-1}^{\infty}f^{t*}_{(2),i}(\tilde\tau_* - \tilde\tau)^{i/2}, \qquad f^\phi_{(2)} = \sum_{i=-1}^{\infty}f^{\phi*}_{(2),i}(\tilde\tau_* - \tilde\tau)^{i/2}. \tag{91}$$

We now obtain expressions which will become useful when asymptotically matching the transition motion. We can compute the second-order azimuthal self-force in two equivalent ways: either using the flux-balance equation (8) and the expansion of $\ell_{(1)}$ (79), or using the solutions (55) and (78) in the Taylor-expansion as (24). We get

$$f_{\phi(2)} = \sum_{i=-1}^{\infty}f^*_{\phi(2),i}(\tilde\tau_* - \tilde\tau)^{i/2}, \tag{92}$$

where the lowest order coefficient is given equivalently by

$$f^*_{\phi(2),-1} = \frac{r^{5/2}_{(0)*}}{4\Omega_{(0)*}A^{1/2}_*} f^{r*}_{(1),1} + \frac{r^3_{(0)*}T^*_{11}}{8A^{3/2}_*} r^*_{(0),1} f^{r*}_{(1),0} \tag{93a}$$

$$- \frac{r_{(0)*}T^*_{15}}{8A^{5/2}_*}\left(r^*_{(0),1}r^*_{(1),0} + r^*_{(0),2}r^*_{(1),-1}\right) - \frac{3r^{3/2}_{(0)*}}{32A^{7/2}_*}E_*(r^*_{(0),1})^2 r^*_{(1),-1}, \tag{93b}$$

$$f^*_{\phi(2),-1} = \left.\frac{dF^t_1}{dr}\right|_{(0)*} r^*_{(1),-1}. \tag{93c}$$

We will use these two expressions to compute $r^*_{(1),0}$ in Section 5.3.

### 3.6.3 n-post-adiabatic order

We further investigate the behaviour of the $n$th post-adiabatic ($n$PA) motion for $n = 2, 3, \ldots$ close to the LSO. At each $n$PA order the homogeneous solution to Eq. (46) is given by the same solution as the 1PA expansion (72). We then consider a specific configuration $(\sigma a, r_{(0)*})$. We compute the source term $S^r_{(n)}$ and its $(\tilde{\tau}_* - \tilde{\tau})$-behaviour close to the LSO, which is determined from lower order quantities that at this point are known. We assume that the order of the divergences at the LSO of the source terms originating from the non-linear terms $f^{t,\text{non-lin}}_{(n+1)}(r_{(0)}, \ldots, r_{(n-1)})$ are lower or equal to the order of the divergences originating from all the other known source terms. Given the behaviour of the source one can straightforwardly compute the one of the particular solution using the method of variation of constants. We find that Eq. (46) is consistent with the following solution as $\tilde{\tau} \to \tilde{\tau}_*$

$$r_{(n)}(\tilde{\tau}) - r_{(0)*}\delta_{n,0} = \sum_{i=-q_{(n)}}^{\infty} r^*_{(n),i}(\tilde{\tau}_* - \tilde{\tau})^{i/2}, \tag{94}$$

where

$$q_{(n)} = \begin{cases} \frac{5n-2}{2} & \text{for } n \text{ even} \\ \frac{5n-3}{2} & \text{for } n \text{ odd} \end{cases}, \tag{95}$$

for all $n = 0, 1, 2, \ldots$

In the same manner as in the previous sections, we also computed the quantities $\delta_{(n)}$ and $W_{(n)}$ that encode the deviation from a circular orbit, and which will be useful to match the transition motion. We find the behaviors

$$\delta_{(n)}(\tilde{\tau}) = \sum_{i=-q'_{(n)}}^{\infty} \delta^*_{(n),i}(\tilde{\tau}_* - \tilde{\tau})^{i/2}, \tag{96}$$

$$W_{(n)}(\tilde{\tau}) = \sum_{i=-q''_{(n)}}^{\infty} w^*_{(n),i}(\tilde{\tau}_* - \tilde{\tau})^{i/2}, \tag{97}$$

where

$$q'_{(n)} = \begin{cases} \frac{5n-4}{2} & \text{for } n \text{ even and } n \neq 0 \\ \frac{5n-5}{2} & \text{for } n \text{ odd} \end{cases}, \qquad q''_{(n)} = \begin{cases} \frac{5n-6}{2} & \text{for } n \text{ even} \\ \frac{5n-7}{2} & \text{for } n \text{ odd} \end{cases}, \tag{98}$$

for all $n = 0, 1, 2, \ldots$. We recall that for $n = 0$ we found $\delta_{(0)} = 0$, see below Eq. (31). We did extensive checks up to $n = 5$. We conjecture that these expansions remain valid for larger $n$.

# 4 The transition-to-plunge motion in the $n$th post-leading transition expansion

We consider an expansion in the rescaled transition proper time

$$s \equiv \eta^{1/5}(\tau - \tau_*), \tag{99}$$

around the LSO crossing time $\tau_*$ or $s_* = 0$. Even in the presence of self-force, the radial potential is given by $V^{\text{geo}}$ as described in Eq. (5) and at leading order in $\eta$ the LSO matches with the ISCO. The transition motion will be defined from $s = -\infty$ (where it will asymptotically match the inspiral) up to the time $s_{\text{break}} > 0$ where the transition-timescale expansion breaks down, see Eq. (118) below. The validity of the transition equations with respect to the full equations of motion will be a smaller interval of proper time which will be assessed in Eq. (121) below.

We denote with $r_{[0]*}$, $\ell_{[0]*}$ and $e_{[0]*}$ the radius, angular momentum and energy evaluated at the ISCO. We define the variables $R$, $\xi$ and $Y$ as [19,21]

$$r - r_{[0]*} = \eta^{2/5} R(\eta, s), \tag{100a}$$

$$\ell - \ell_{[0]*} = \eta^{4/5} \xi(\eta, s), \tag{100b}$$

$$e - e_{[0]*} = \Omega_{[0]*}\left[\eta^{6/5} Y(\eta, s) + \eta^{4/5}\xi(\eta, s)\right]. \tag{100c}$$

Integrating the angular momentum flux-balance law (8) and comparing with Eq. (100b) we obtain

$$\xi(\eta, s) = \eta^{-1}\int^s f_\phi(\eta, s') ds'. \tag{101}$$

In the absence of radial self-force all variables $R$, $\xi$, and $Y$ scale as $\eta^0$ in the transition region for standard spins [19,21,25,26]. In the presence of radial self-force corrections we expand

$$R(\eta, s) = \sum_{i=0}^{\infty} \eta^{i/5} R_{[i]}(s), \qquad \xi(\eta, s) = \sum_{i=0}^{\infty} \eta^{i/5}\xi_{[i]}(s), \qquad Y(\eta, s) = \sum_{i=0}^{\infty} \eta^{i/5} Y_{[i]}(s). \tag{102}$$

The indices in square brackets $[i]$ label the terms appearing at relative order $\eta^{i/5}$ with respect to the first nonvanishing leading terms in the transition-timescale expansion around $r_{[0]*}$, $\ell_{[0]*}$ and $e_{[0]*}$.

We consider the self-force along the particle worldline in the transition region. Using the transition-timescale expansion (100) and (102) in Eq. (21) we obtain

$$
\begin{aligned}
f^\mu =\ & \eta\ F_1^\mu\big|_{[0]*} + \eta^{7/5}\frac{dF_1^\mu}{dr}\bigg|_{[0]*} R_{[0]} + \eta^{8/5}\frac{dF_1^\mu}{dr}\bigg|_{[0]*} R_{[1]} \\
& + \eta^{9/5}\left(\frac{dF_1^\mu}{dr}\bigg|_{[0]*} R_{[2]} + \frac{1}{2}\frac{d^2F_1^\mu}{dr^2}\bigg|_{[0]*} R_{[0]}^2\right) \\
& + \eta^2\left(\frac{dF_1^\mu}{dr}\bigg|_{[0]*} R_{[3]} + \frac{d^2F_1^\mu}{dr^2}\bigg|_{[0]*} R_{[0]}R_{[1]} + F_2^r\big|_{[0]*}\right) \\
& + \eta^{11/5}\left(\frac{dF_1^\mu}{dr}\bigg|_{[0]*} R_{[4]} + \frac{1}{2}\frac{d^2F_1^\mu}{dr^2}\bigg|_{[0]*}\left(R_{[1]}^2 + R_{[0]}R_{[2]}\right) + \frac{1}{6}\frac{d^3F_1^\mu}{dr^3}\bigg|_{[0]*} R_{[0]}^3\right)
\end{aligned}
$$

$$+\eta^{12/5}\left(\frac{dF_1^\mu}{dr}\bigg|_{[0]*}R_{[5]}+\frac{d^2F_1^\mu}{dr^2}\bigg|_{[0]*}\left(R_{[1]}R_{[2]}+R_{[0]}R_{[3]}\right)\right.$$
$$+\frac{1}{2}\frac{d^3F_1^\mu}{dr^3}\bigg|_{[0]*}R_{[0]}^2R_{[1]}+\frac{dF_2^\mu}{dr}\bigg|_{[0]*}R_{[0]}\right)$$
$$+\eta^{13/5}\left(\frac{dF_1^\mu}{dr}\bigg|_{[0]*}R_{[6]}+\frac{1}{2}\frac{d^2F_1^\mu}{dr^2}\bigg|_{[0]*}\left(R_{[2]}^2+2R_{[1]}R_{[3]}+2R_{[0]}R_{[4]}\right)\right.$$
$$+\frac{1}{2}\frac{d^3F_1^\mu}{dr^3}\bigg|_{[0]*}\left(R_{[0]}^2R_{[2]}+R_{[1]}^2R_{[0]}\right)+\frac{1}{4!}\frac{d^4F_1^\mu}{dr^4}\bigg|_{[0]*}R_{[0]}^4+\frac{dF_2^\mu}{dr}\bigg|_{[0]*}R_{[1]}\right)$$
$$+\eta^{14/5}\left(\frac{dF_1^\mu}{dr}\bigg|_{[0]*}R_{[7]}+\frac{d^2F_1^\mu}{dr^2}\bigg|_{[0]*}\left(R_{[2]}R_{[3]}+R_{[1]}R_{[4]}+R_{[0]}R_{[5]}\right)\right.$$
$$+\frac{1}{6}\frac{d^3F_1^\mu}{dr^3}\bigg|_{[0]*}\left(R_{[1]}^3+3R_{[0]}^2R_{[3]}+6R_{[0]}R_{[1]}R_{[2]}\right)+\frac{1}{6}\frac{d^4F_1^\mu}{dr^4}\bigg|_{[0]*}R_{[0]}^3R_{[1]}\quad (103)$$
$$+\frac{dF_2^\mu}{dr}\bigg|_{[0]*}R_{[2]}+\frac{1}{2}\frac{d^2F_2^\mu}{dr^2}\bigg|_{[0]*}R_{[0]}^2\right)+O_s(\eta^3).$$

The same expansion can be made for $f_\mu$ or any tensorial quantity pulled back on the particle worldline. The terms $\sim\eta^2$ start to depend upon the second-order self-force $F_2^\mu$; the terms $\sim\eta^3$ start to depend upon $F_3^\mu$, ... We will denote the series expansion of the following self-force components as

$$f_{t,\phi}(\eta,s)=\eta\sum_{i=0}^\infty\eta^{i/5}f_{t,\phi[i]}(s),\qquad f^r(\eta,s)=\eta\sum_{i=0}^\infty\eta^{i/5}f_{[i]}^r(s).\qquad (104)$$

Comparing such an expansion of $f_\phi$ with the one in Eq. (103) and using Eq. (101) gives

$$f_{\phi[0]}(s)=-\kappa_*,\qquad f_{\phi[1]}(s)=0,\qquad f_{\phi[2]}(s)=\frac{dF_{\phi 1}}{dr}\bigg|_{[0]*}R_{[0]}(s),\qquad (105a)$$

$$\xi_{[0]}(s)=-\kappa_*s,\qquad \xi_{[1]}(s)=0,\qquad \xi_{[2]}(s)=\frac{dF_{\phi 1}}{dr}\bigg|_{[0]*}\int^s R_{[0]}(s')ds',\qquad (105b)$$

where

$$\kappa_*=-\left.F_{\phi 1}\right|_{[0]*}.\qquad (106)$$

It is clear that $\kappa_*>0$ since angular momentum loss drives the transition motion. Such expansions are compatible with the field equations (8), (9), (114).

During the transition motion we have the property

$$\frac{d\log r}{d\phi}=\frac{dr/dt}{r\Omega}=\frac{\eta^{3/5}}{r\Omega U}\frac{dR}{ds}=O_s(\eta^{3/5}),\qquad (107)$$

indicating a loss in quasi-circularity with respect to the inspiral, see Eq. (20), as expected when moving towards the final plunging motion.

### 4.1 Leading-order (0PL) transition

We now derive the solution to Eqs. (1), (2), (4), (5), (8), (9), (10), (12) and (15) at leading order in the transition-timescale expansion around the LSO. The condition

$$\frac{\partial^2 V^{\text{geo}}(e, \ell, r, a)}{\partial r^2} = 0, \tag{108}$$

together with (1), (2), (5) and (10) give at leading order in $\eta$ and at the LSO the same quantities $\ell_{[0]*} = \ell_{(0)*}$, $e_{[0]*} = e_{(0)*}$, $\Omega_{[0]*} = \Omega_{(0)*}$ as the adiabatic inspiral after identifying $r_{[0]*} = r_{(0)*}$.

Using the expansions (100) and (102) and comparing Eqs. (2) and (15), the deviation $\delta$ is given at leading order around the LSO as

$$\delta(\eta, s) = \eta^{4/5} \delta_{[0]}(s) + O_s(\eta), \qquad \delta_{[0]}(s) = \frac{\pi_*}{\Delta_{[0]*}^2} R_{[0]}^2(s) - \frac{A_*^{3/2} \Omega_{[0]*}}{\Delta_{[0]*}} \xi_{[0]}(s), \tag{109}$$

where $\pi_* \equiv \sigma a^3 - 6\sigma a r_{(0)*} + 2(3 + a^2) r_{(0)*}^{3/2} - 3\sigma a r_{(0)*}^2$. For the angular velocity, consistently with Eq. (2), we have

$$\Omega(\eta, s) = \Omega_{[0]*} + \eta^{2/5} \Omega_{[2]}(s) + O_s(\eta^{3/5}), \qquad \Omega_{[2]}(s) = -2 \frac{\beta_* \Omega_{[0]*}}{\gamma_*} R_{[0]}(s), \tag{110}$$

where the coefficients $\beta_*$ and $\gamma_*$ are defined in Eq. (111) below.

Given $V^{\text{geo}} = V^{\text{geo}}(e, \ell, r, a)$, we define the coefficients

$$\alpha_* \equiv \frac{1}{4} \left. \frac{\partial^3 V^{\text{geo}}}{\partial r^3} \right|_{[0]*}, \qquad \beta_* \equiv -\frac{1}{2} \left( \frac{\partial^2 V^{\text{geo}}}{\partial r \partial \ell} + \Omega \frac{\partial^2 V^{\text{geo}}}{\partial r \partial e} \right) \bigg|_{[0]*}, \qquad \gamma_* \equiv \left. \frac{\partial V^{\text{geo}}}{\partial \ell} \right|_{[0]*}. \tag{111}$$

Expanding Eqs. (5) and (10) by substituting Eqs. (99), (100), (102), (104) and (105b), the leading-order transition equations are then given by

$$\left( \frac{dR_{[0]}}{ds} \right)^2 = -\frac{2}{3} \alpha_* R_{[0]}^3 - 2\beta_* \kappa_* s R_{[0]} + \gamma_* Y_{[0]}, \tag{112a}$$

$$\frac{d^2 R_{[0]}}{ds^2} = -\alpha_* R_{[0]}^2 - \kappa_* \beta_* s, \tag{112b}$$

at order $\eta^{6/5}$ and $\eta^{4/5}$, respectively. These two equations are compatible if and only if

$$\frac{dY_{[0]}}{ds} = \frac{2\kappa_* \beta_*}{\gamma_*} R_{[0]}. \tag{113}$$

The orthogonality condition (9) then gives

$$f^r = (U g^{rr})|_* \Omega_{[0]*} \eta^{4/5} \left( \frac{dY_{[0]}}{ds} - 2 \frac{\kappa_* \beta_*}{\gamma_*} R_{[0]} \right) \left( \frac{dR_{[0]}}{ds} \right)^{-1} + O(\eta) = O(\eta), \tag{114}$$

which matches the expansion in Eq. (104). Being of order $\eta$, the radial self-force is subleading in Eq. (112b).

Introducing the rescaled variables

$$x_{[0]} \equiv \alpha_*^{3/5} (\beta_* \kappa_*)^{-2/5} R_{[0]}, \tag{115a}$$

$$y_{[0]} \equiv \alpha_*^{4/5} (\beta_* \kappa_*)^{-6/5} \gamma_* Y_{[0]}, \tag{115b}$$

$$t \equiv (\alpha_* \beta_* \kappa_*)^{1/5} s, \tag{115c}$$

we obtain the normalized leading-order transition equations [19–21]

$$\left(\frac{dx_{[0]}}{dt}\right)^2 = -\frac{2}{3}x_{[0]}^3 - 2x_{[0]}t + y_{[0]}, \tag{116a}$$

$$\frac{d^2x_{[0]}}{dt^2} = -x_{[0]}^2 - t, \tag{116b}$$

$$\frac{dy_{[0]}}{dt} = 2x_{[0]}. \tag{116c}$$

The solution $x_{[0]}$ is the Painlevé transcendent of the first kind and $y_{[0]}$ is twice its first integral [25].

As we will see below in Section 4.3, in order to match the inspiral, the relevant solutions to Eqs. (116) as $t \to -\infty$ are given by

$$x_{[0]} = \sqrt{-t} + \frac{1}{8}(-t)^{-2} + O(-t)^{-9/2}, \tag{117a}$$

$$y_{[0]} = -\frac{4}{3}(-t)^{3/2} + \frac{1}{4}(-t)^{-1} + O(-t)^{-7/2}. \tag{117b}$$

The Painlevé transcendent of the first kind is then uniquely defined from $t = -\infty$ to a finite $t$ that we denote as $t_{\text{break}}^{(+)} \approx 3.41$ where $x_{[0]} \to -\infty$ marking the breakdown of the transition-timescale expansion (100).

The transition equations are valid as leading equations for the actual motion for all $s$ such that $|\eta^{2/5}R_{[0]}(s)| \ll 1$, $|\eta^{4/5}\xi_{[0]}(s)| \ll 1$ and $|\eta^{6/5}Y_{[0]}(s)| \ll 1$. At early times, using Eqs. (105b), (115) and (117) to compute $\xi_{[0]}$, $R_{[0]}$ and $Y_{[0]}$ (see Eqs. (122) for the results), we deduce that the transition regime breaks down when $\tau_{\text{break}}^{(-)} - \tau_* \sim -\eta^{-1}$. At late times, $R_{[0]}$ diverges at $s \sim s_{\text{break}}^{(+)} = (\alpha_*\beta_*\kappa_*)^{-1/5}t_{\text{break}}^{(+)} \sim \eta^0$ or, equivalently, at

$$\tau_{\text{break}}^{(+)} - \tau_* \sim 3.41(\alpha_*\beta_*\kappa_*)^{-1/5}\eta^{-1/5}. \tag{118}$$

We can compute the asymptotic solution to Eq. (116b) as $\tau \to \tau_{\text{break}}^{(+)}$. At leading order, and using the relation (115a) to obtain of $R_{[0]}$, we get

$$R_{[0]} \sim -\frac{6}{\alpha_*\eta^{2/5}(\tau_{\text{break}}^{(+)} - \tau)^2}. \tag{119}$$

The validity condition $\eta^{2/5}R_{[0]} \ll 1$ of the transition-timescale expansion then translates into

$$\tau - \tau_* \ll \left(\frac{6}{\alpha_*}\right)^{1/2} + 3.41(\eta\,\alpha_*\beta_*\kappa_*)^{-1/5} \sim \eta^{-1/5}. \tag{120}$$

We summarize the range of validity of the transition motion as

$$-\eta^{-1} \ll \tau - \tau_* \ll \eta^{-1/5}. \tag{121}$$

### 4.1.1 Early-times 0PL transition solution

We compute the asymptotic solution for the dynamical quantities in the limit of early times $(s \to -\infty)$ where the transition will be shown to connect smoothly with the inspiral.

We obtain $R_{[0]}(s)$ and $Y_{[0]}(s)$ from Eqs. (115) and (117),

$$R_{[0]}(s) = \left(\frac{\kappa_* \beta_*}{\alpha_*}\right)^{1/2}(-s)^{1/2} + \frac{1}{8\alpha_*}(-s)^{-2} + O(-s)^{-9/2}, \tag{122a}$$

$$Y_{[0]}(s) = -\frac{4\beta_* \kappa_*}{\gamma_*}\left(\frac{\kappa_* \beta_*}{\alpha_*}\right)^{1/2}(-s)^{3/2} + \frac{\beta_* \kappa_*}{4\alpha_* \gamma_*}(-s)^{-1} + O(-s)^{-7/2}. \tag{122b}$$

Using this solution in Eq. (105b) we get $\xi_{[2]} = \lambda_*(-s)^{3/2} + O(s^{-1})$ where

$$\lambda_* \equiv -\frac{2}{3}\left(\frac{\beta_* \kappa_*}{\alpha_*}\right)^{1/2}\left.\frac{dF_{\phi 1}}{dr}\right|_{[0]*}. \tag{123}$$

Upon substituting this result together with Eqs. (102), (105b), (122) and (123) into Eqs. (100) and using the definition (99), the dependence on $\eta$ and $s$ recombines into a dependence on $\eta$ and $\tilde{\tau} = \eta\,\tau \to -\infty$ at leading order as

$$r(s) = \left(\frac{\kappa_* \beta_*}{\alpha_*}\right)^{1/2}(\tilde{\tau}_* - \tilde{\tau})^{1/2} + r_{[0]*} + O\left(\eta^2(\tilde{\tau}_* - \tilde{\tau})^{-2}\right) + O_s(\eta^{3/5}), \tag{124}$$

$$\ell(s) = \lambda_*(\tilde{\tau}_* - \tilde{\tau})^{3/2} + \kappa_*(\tilde{\tau}_* - \tilde{\tau}) + \ell_{[0]*} + O\left(\eta^2(\tilde{\tau}_* - \tilde{\tau})^{-1}\right) + O_s(\eta^{7/5}), \tag{125}$$

$$e(s) = \Omega_{[0]*}\left(-\frac{4\kappa_* \beta_*}{3\gamma_*}\left(\frac{\kappa_* \beta_*}{\alpha_*}\right)^{1/2} + \lambda_*\right)(\tilde{\tau}_* - \tilde{\tau})^{3/2} \tag{126}$$

$$+ \kappa_* \Omega_{[0]*}(\tilde{\tau}_* - \tilde{\tau}) + e_{[0]*} + O\left(\eta^2(\tilde{\tau}_* - \tilde{\tau})^{-1}\right) + O_s(\eta^{7/5}). \tag{127}$$

We consider the two quantities that encode the deviation from circularity, namely $\delta$ and $W = \Omega_{[0]*}^{-1}(e - e_{[0]*}) - (\ell - \ell_{[0]*})$. Using the solutions (122) with Eqs. (125) and (126) in Eqs. (109) and (49) we obtain their asymptotic early-time behaviour

$$\delta(s) = \frac{\pi_*}{4\alpha_* \Delta_{[0]*}^2}\left(\frac{\kappa_* \beta_*}{\alpha_*}\right)^{1/2}\eta^2(\tilde{\tau}_* - \tilde{\tau})^{-3/2} + O(\eta^4(\tilde{\tau}_* - \tilde{\tau})^{-4}) + O_s(\eta), \tag{128}$$

$$W(s) = -\frac{4\kappa_* \beta_*}{3\gamma_*}\left(\frac{\kappa_* \beta_*}{\alpha_*}\right)^{1/2}(\tilde{\tau}_* - \tilde{\tau})^{3/2} + O\left(\eta^2(\tilde{\tau}_* - \tilde{\tau})^{-1}\right) + O_s(\eta^{7/5}). \tag{129}$$

We notice that there is no zeroth-order term in the mass ratio in the solution for $\delta$. This is consistent with the adiabatic result in the inspiral $\delta_{(0)} = 0$.

For the angular velocity from Eqs. (110) and (122a) we obtain

$$\Omega(s) = \Omega_{[0]*} - 2\frac{\beta_* \Omega_{[0]*}}{\gamma_*}\left(\frac{\kappa_* \beta_*}{\alpha_*}\right)^{1/2}(\tilde{\tau}_* - \tilde{\tau})^{1/2} + O\left(\eta^2(\tilde{\tau}_* - \tilde{\tau})^{-2}\right) + O_s(\eta^{3/5}). \tag{130}$$

This behavior will be matched to the inspiral motion in Section 5.1.

## 4.2 Structure of the general subleading transition motion

Building upon what we have done for the 0PL transition motion, we present in the following the algorithm that we use to solve the transition motion at arbitrary high subleading order. We apply the following steps:

- As a consequence of Eqs. (100), $U$, $\Omega$ as well as $\delta$ also admit an expansion in powers of $\eta^{1/5}$. The expansion of $\delta$ takes the form

$$\delta(\eta, s) = \eta^{4/5}\sum_{i=0}^{\infty}\eta^{i/5}\delta_{[i]}(s), \tag{131}$$

since the leading order is $\eta^{4/5}$. Using the definitions (2), (15), (100) and the expansion (102) one can algebraically solve for each $\delta_{[i]}$ as a function of $R_{[j]}(s)$ and $\xi_{[j]}(s)$ with $j = 0, 1, \ldots, i$ as well as $Y_{[j]}(s)$ with $j = 0, 1, \ldots i-2$

$$\delta_{[i]}(s) = \delta_{[i]}\left[R_{[0,1,\ldots,i]}(s), \xi_{[0,1,\ldots,i]}(s), Y_{[0,1,\ldots,i-2]}(s)\right]. \tag{132}$$

The first six such terms can be found in Appendix B.

- We consider the self-force in the transition region. In the same manner as in Eq. (103), using the transition-timescale expansion (100) and (102) in Eq. (21) we obtain

$$f_{t,\phi}(\eta,s) = \eta \sum_{i=0}^{\infty} \eta^{i/5} f_{t,\phi[i]}(s), \qquad f_{t,\phi[i]}(s) = f_{t,\phi[i]}[R_{[0,1,\ldots,i-2]}(s)], \tag{133a}$$

$$f^r(\eta,s) = \eta \sum_{i=0}^{\infty} \eta^{i/5} f^r_{[i]}(s), \qquad f^r_{[i]}(s) = f^r_{[i]}[R_{[0,1,\ldots,i-2]}(s)]. \tag{133b}$$

These first coefficients of the expansions of the radial and azimuthal self-force are given in Eq. (103).

- Substituting the expansion of the azimuthal self-force (133a) into Eq. (101), we algebraically solve for $\xi_{[i]}(s)$ defined in Eq. (102),

$$\xi_{[i]}(s) = \xi_{[i]}[R_{[0,1,\ldots,i-2]}(s)]. \tag{134}$$

In particular $\xi_{[0]}(s) = -\kappa_* s$, $\xi_{[1]}(s) = 0$ and $\xi_{[2]}(s)$ is given by Eq. (105b). Terms up to $i = 6$ can be computed using the expansion (103).

- Substituting Eqs. (100), (102), (133b) and (134) into the remaining equations of motion, namely Eqs. (5), (9) and (10), we obtain

$$\frac{dR_{[0]}(s)}{ds}\frac{dR_{[i]}(s)}{ds} + \frac{1}{2}\sum_{j=1}^{i-1}\frac{dR_{[j]}(s)}{ds}\frac{dR_{[i-j]}(s)}{ds} \tag{135a}$$

$$+ \alpha_* R^2_{[0]}(s)R_{[i]}(s) + \beta_* \kappa_* s R_{[i]}(s) - \frac{\gamma_*}{2}Y_{[i]}(s) = S^{RY}_{[i]}(s), \tag{135b}$$

$$\frac{d^2 R_{[i]}(s)}{ds^2} + 2\alpha_* R_{[0]}(s)R_{[i]}(s) = S^R_{[i]}(s), \tag{135c}$$

$$\frac{dY_{[i]}(s)}{ds} - \frac{2\kappa_* \beta_*}{\gamma_*}R_{[i]}(s) = S^Y_{[i]}(s), \tag{135d}$$

where $S^{RY}_{[i]}(s)$, $S^R_{[i]}(s)$ and $S^Y_{[i]}(s)$, $i = 1, 2, \ldots$ are source terms that can be obtained algebraically. We list the first such terms in Appendix C. The operators on the left-hand side are just the linearization of the zeroth-order operators in Eqs. (112) and (113). It implies that at all orders in perturbations in $\eta$ the equations of motion are sourced linearized Painlevé transcendental equations of the first kind. The equations are consistent given that the sources obey

$$\frac{dS^{RY}_{[i]}(s)}{ds} = S^R_{[i]}(s)\frac{dR_{[0]}(s)}{ds} - \frac{\gamma_*}{2}S^Y_{[i]}(s) + \sum_{j=1}^{i-1}\frac{d^2 R_{[j]}(s)}{ds^2}\frac{dR_{[i-j]}(s)}{ds}. \tag{136}$$

We explicitly checked this condition up to $i = 6$ using the sources given in Appendix C.

- The point which requires the most attention is the solution to the equations of motion at order $i$ (135). We extend the definition (115) for $i = 0$ to any $i = 0, 1, 2, \ldots$ as

$$x_{[i]} \equiv \alpha_*^{3/5}(\beta_*\kappa_*)^{-2/5}R_{[i]}, \tag{137a}$$

$$y_{[i]} \equiv \alpha_*^{4/5}(\beta_*\kappa_*)^{-6/5}\gamma_* Y_{[i]}, \tag{137b}$$

as well as

$$s_{[i]}^{xy} \equiv \alpha_*^{4/5}(\beta_*\kappa_*)^{-6/5}S_{[i]}^{RY}, \tag{138a}$$

$$s_{[i]}^{x} \equiv \alpha_*^{1/5}(\beta_*\kappa_*)^{-4/5}S_{[i]}^{R}, \tag{138b}$$

$$s_{[i]}^{y} \equiv \alpha_*^{3/5}(\beta_*\kappa_*)^{-7/5}\gamma_* S_{[i]}^{Y}. \tag{138c}$$

Recalling the definition of $t$ (115c), the equations of motion at order $i$ (135) then take the normalized form

$$\frac{dx_{[0]}}{dt}\frac{dx_{[i]}}{dt} + \frac{1}{2}\sum_{j=1}^{i-1}\frac{dx_{[j]}}{dt}\frac{dx_{[i-j]}}{dt} + x_{[0]}^2 x_{[i]} + t x_{[i]} - \frac{1}{2}y_{[i]} = s_{[i]}^{xy}, \tag{139a}$$

$$\frac{d^2 x_{[i]}}{dt^2} + 2x_{[0]}x_{[i]} = s_{[i]}^{x}, \tag{139b}$$

$$\frac{dy_{[i]}}{dt} - 2x_{[i]} = s_{[i]}^{y}. \tag{139c}$$

We reduced the equations to sourced linearized Painlevé transcendental equations of the first kind. We will call these equations the *linearized PT1 equations* for short. In Section 4.3 we will first describe the two independent homogeneous solutions to Eqs. (139b). The inhomogeneous solution to (139b) will be then obtained from the method of variations of constants in Section 4.4.

- The motion in the transition region can be entirely parametrized by $x_{[i]}$, $i \geq 0$ (or, equivalently, $R_{[i]}$) which is obtained from the equation of motion (139b). All other quantities are algebraically determined. Note that $y_{[i]}$ obeys the differential equation (139c), but can be also solved for algebraically using Eq. (139a). Finally, once $x_{[i]}(t)$ and $y_{[i]}(t)$ are known, it will be straightforward to compute the solutions for all the quantities $(r, \ell, e, \delta, W, \ldots)$ that will be matched to the slow-time expansion of the inspiral.

## 4.3 Homogeneous solution to the linearized PT1 equations

In this section we describe the properties of the early-time asymptotic homogeneous solutions to Eq. (139b).

The non-linear Painlevé transcendental (PT1) equation (116b) which governs the transition at leading order admits a known explicit asymptotic solution as $t \to -\infty$ which depends upon two parameters $d$ and $\theta_0$. Using Mathematica, we found that the asymptotic solution takes the form

$$x_{[0]}(t; d, \theta_0) = \sqrt{-t}\left[1 + \sum_{n=1}^{\infty}\left(-\frac{t}{6^{1/5}}\right)^{-5n/8}\left(\sum_{k=0}^{\lfloor n/2 \rfloor} s_{n,k}\sin^{2k+n-2\lfloor\frac{n}{2}\rfloor}(\Phi)\right.\right.$$
$$\left.\left. + \sum_{k=0}^{\lfloor(n-3)/2\rfloor} c_{n,k}\sin^{2k+1-n+2\lfloor\frac{n}{2}\rfloor}(\Phi)\cos(\Phi)\right)\right], \tag{140}$$

or, equivalently, expanding as cases $n = 2i$ and $n = 2i + 1$,

$$
\begin{aligned}
x_{[0]}(t; d, \theta_0) = \sqrt{-t} \Bigg[ 1 + \sum_{i=1}^{\infty} \left( -\frac{t}{6^{1/5}} \right)^{-10i/8} & \Bigg( \sum_{k=0}^{i} s_{2i,k} \sin^{2k}(\Phi) \\
& + \sum_{k=0}^{\lfloor (2i-3)/2 \rfloor} c_{2i,k} \sin^{2k+1}(\Phi) \cos(\Phi) \Bigg) \\
+ \sum_{i=0}^{\infty} \left( -\frac{t}{6^{1/5}} \right)^{-5(2i+1)/8} & \Bigg( \sum_{k=0}^{\lfloor (2i+1)/2 \rfloor} s_{2i+1,k} \sin^{2k+1}(\Phi) \\
& + \sum_{k=0}^{i-1} c_{2i+1,k} \sin^{2k}(\Phi) \cos(\Phi) \Bigg) \Bigg],
\end{aligned}
\tag{141}
$$

where we defined

$$
\Phi(t; d, \theta_0) \equiv (24)^{1/4} \left( \frac{4}{5} \left( -\frac{t}{6^{1/5}} \right)^{5/4} - \frac{5}{8} d^2 \ln \left( -\frac{t}{6^{1/5}} \right) \right) + \theta_0,
\tag{142}
$$

and where $s_{n,k}$, $n = 1, 2, \ldots$, $k = 0, 1, \ldots \lfloor n/2 \rfloor$ and $c_{n,k}$, $n = 1, 2, \ldots$, $k = 0, 1, \ldots \lfloor (n-3)/2 \rfloor$ are real numbers which can be computed algebraically. The leading coefficients are given by $s_{1,0} = -\sqrt{6} d$, $s_{2,0} = -2d^2$, $s_{2,1} = d^2$, $c_{3,0} = \frac{5}{32} \left( \frac{3}{2} \right)^{1/4} d - \frac{47}{16} \left( \frac{3}{2} \right)^{3/4} d^5, \ldots$

Since the quasi-circular inspiral and transition have a monotonically descreasing radius, we are interested at leading order in the non-oscillating solution, which requires $d = 0$. We expect that oscillating solutions with $d \neq 0$ will be relevant for the inspiral and transition motion with eccentricity. For $d = 0$, the solution becomes

$$
x_{[0]}(t; d = 0, \theta_0) = \sqrt{-t} + \frac{1}{8}(-t)^{-2} - \frac{49}{128}(-t)^{-9/2} + \frac{1225}{256}(-t)^{-7} + O(-t)^{-19/2},
\tag{143}
$$

which originates from the coefficients $s_{4i,0}$, $i \geq 1$ that contain $d$-independent terms, which are non-vanishing when $d = 0$. The first such coefficients are $s_{4,0} = \frac{\sqrt{6} - 130 d^2}{48}$, $s_{8,0} = \frac{-169344 + 6413015 \sqrt{6} d^4 - 59848470 d^8 + 1582866 \sqrt{6} d^{12}}{2654208}$.

Twice the indefinite integral of $x_{[0]}(t; d = 0, \theta_0)$ is

$$
y_{[0]}(t; d = 0, \theta_0) = -\frac{4}{3}(-t)^{3/2} + \frac{1}{4}(-t)^{-1} - \frac{7}{32}(-t)^{-7/2} + \frac{1225}{768}(-t)^{-6} + O(-t)^{-17/2}.
\tag{144}
$$

The asymptotic homogeneous solution to Eqs. (139b) will be computed from $\partial x_{[0]}/\partial d$ and $\partial x_{[0]}/\partial \theta_0$,

$$
x_{\text{lin}}^1 = -6^{-5/8} \frac{\partial x_{[0]}}{\partial d} = \sum_{k=0}^{\infty} (-1)^k 2^{-13k/2} c_k^{\text{lin}} (-t)^{\frac{-1-10k}{8}} \sin \left( \frac{4\sqrt{2}}{5} (-t)^{5/4} + \theta_0 + \frac{\pi}{2} k \right),
\tag{145a}
$$

$$
x_{\text{lin}}^2 = -6^{-5/8} \frac{1}{d} \frac{\partial x_{[0]}}{\partial \theta_0} = \sum_{k=0}^{\infty} (-1)^k 2^{-13k/2} c_k^{\text{lin}} (-t)^{\frac{-1-10k}{8}} \cos \left( \frac{4\sqrt{2}}{5} (-t)^{5/4} + \theta_0 + \frac{\pi}{2} k \right).
\tag{145b}
$$

They depend upon the same coefficients $c_k^{\text{lin}}$, $k = 0, 1, \ldots$ The leading coefficients are given by $c_0^{\text{lin}} = 1$, $c_1^{\text{lin}} = 10$, $c_2^{\text{lin}} = 450$, $c_3^{\text{lin}} = \frac{1365316}{15}$, $c_4^{\text{lin}} = \frac{34654466}{3}$, $c_5^{\text{lin}} = 5174126188$ and the others are easily found. The Wronskian is given by

$$
\omega = x_{\text{lin}}^1 \frac{dx_{\text{lin}}^2}{dt} - \frac{dx_{\text{lin}}^1}{dt} x_{\text{lin}}^2 = \sqrt{2}.
\tag{146}
$$

Since the approach to the LSO of the quasi-circular inspiral is consistent with the absence of oscillations as developed in Section 3, we will set the homogeneous solution to Eqs. (139b) to 0 for all $i = 1, 2, \ldots$. These fixing of initial data are matching conditions between the late-time inspiral and the transition motion.

### 4.4 Inhomogeneous solution to the linearized PT1 equations

We will now give the general method to solve the inhomogeneous linearized PT1 equations (139) in the limit $t \to -\infty$ given the structure of the sources in the transition expansion.

In order to find a particular solution to Eq. (139b) at post-leading order $i$ we use the method of variation of constants: given the two homogeneous solutions $x_{\text{lin}}^{1,2}$ (145), the particular solution $x_{[i]}^p$ is computed from $x_{[i]}^p(t) = x_{(\text{inhom})}[s_{[i]}^x](t)$ where

$$x_{(\text{inhom})}[s_{[i]}^x](t) = p_{[i]}^1(t)x_{\text{lin}}^1(t) + p_{[i]}^2(t)x_{\text{lin}}^2(t), \tag{147}$$

and the coefficients $p_{[i]}^1(t)$ and $p_{[i]}^2(t)$ solve

$$p_{[i]}^1(t) = -\omega^{-1} \int^t dt' s_{[i]}^x(t')x_{\text{lin}}^2(t'), \qquad p_{[i]}^2(t) = \omega^{-1} \int^t dt' s_{[i]}^x(t')x_{\text{lin}}^1(t'). \tag{148}$$

After an explicit evaluation, the inhomogeneous PT1 solution for sources $(-t)^{-k/2}$ can be written as

$$x_{(\text{inhom})}[(-t)^{-k/2}](t) = \frac{1}{2}(-t)^{-(k+1)/2} \sum_{\ell=0}^{\infty} x_{\text{Painlevé}}^\ell(k)(-t)^{-5\ell/2}, \tag{149}$$

where the coefficients $x_{\text{Painlevé}}^\ell(k)$, $\ell \in \mathbb{N}$, can be computed from the asymptotic behavior of the Painlevé transcendent. The first coefficients are given in Table 1.

Table 1: Coefficients of the inhomogeneous solution to the Painlevé PT1 equations for a source $(-t)^{k/2}$.

| $x_{\text{Painlevé}}^0(k)$ | $x_{\text{Painlevé}}^1(k)$ | $x_{\text{Painlevé}}^2(k)$ | $x_{\text{Painlevé}}^3(k)$ | $\ldots$ |
|---|---|---|---|---|
| 1 | $\frac{(2+k)^2}{8}$ | $\frac{441+2k(9+k)(28+k(9+k))}{128}$ | $\frac{34300+k(14+k)(2977+k(14+k)(97+k(14+k)))}{512}$ | $\ldots$ |

Before deriving the general structure of the solutions $x_{[i]}$ to the inhomogeneous linearized PT1 equations, we first solve for $x_{[1]}$. Given the relation (138) and the source (236a), the differential equation (139b) for $i = 1$ becomes

$$\frac{d^2 x_{[1]}}{dt^2} + 2x_{[0]}x_{[1]} = s_{[1]}^x, \tag{150}$$

$$s_{[1]}^x \equiv \frac{\alpha_*^{1/5}}{(\beta_* \kappa_*)^{4/5}} f_{[0]}^r = \frac{\alpha_*}{(\alpha_* \beta_* \kappa_*)^{4/5}} \left. F_1^r \right|_{[0]*}. \tag{151}$$

The source term $s_{[1]}^x$ is a constant as it is proportional to the radial self-force evaluated at the ISCO (this means, in the notation below, $s_{[1]}^{x,0} = s_{[1]}^x$ and $s_{[1]}^{x,k} = 0$, $\forall k > 0$). Using the general result of Eq. (149) for $k = 0$ we get

$$x_{[1]}(t) = x_{(\text{inhom})}[s_{[1]}^x(-t)^0](t) = \frac{s_{[1]}^x}{2}(-t)^{-1/2} \sum_{\ell=0}^{\infty} x_{\text{Painlevé}}^\ell(0)(-t)^{-5\ell/2} \tag{152a}$$

$$\equiv (-t)^{-1/2} \sum_{\ell=0}^{\infty} x_{[1]}^\ell(-t)^{-5\ell/2}, \tag{152b}$$

$$x_{[1]}^\ell \equiv \frac{s_{[1]}^x}{2} x_{\text{Painlevé}}^\ell(0). \tag{152c}$$

We are now ready to study the generic $i \geq 1$ case.

The asymptotic $t \to -\infty$ behavior of $s^x_{[i]}(t)$ as given in Eqs. (236) for $i = 1, 2, 3, 4, 5, 6$ is consistent with

$$s^x_{[i]}(t) = (-t)^{c_i/2} \sum_{k=0}^{\infty} (-t)^{-5k/2} s^{x,k}_{[i]}, \tag{153}$$

where $s^{x,k}_{[i]}$ are coefficients and $c_i = (i+4)/2$ for $i \geq 2$ even and $c_i = (i-1)/2$ for $i \geq 1$ odd. We conjecture that this expansion holds for any $i \geq 7$ as well. These coefficients are the solution to the three term recurrence $c_1 = 0$, $c_2 = 3$, $c_3 = 1$, $c_n = c_{n-1} + c_{n-2} - c_{n-3}$ for $n \geq 3$. This defines by extension $c_0 = 2$.

Using Eqs. (149) and (153), the general solution is therefore

$$
\begin{aligned}
x_{[i]}(t) = x_{(\text{inhom})}[s^x_{[i]}](t) &= \frac{1}{2}(-t)^{(c_i-1)/2} \sum_{k=0}^{\infty} \sum_{\ell=0}^{\infty} s^{x,k}_{[i]} x^\ell_{\text{Painlevé}} (5k - c_i)(-t)^{-5(k+\ell)/2} \\
&= (-t)^{(c_i-1)/2} \sum_{k=0}^{\infty} \left( \frac{1}{2} \sum_{\ell=0}^{k} s^{x,k-\ell}_{[i]} x^\ell_{\text{Painlevé}} (5(k-\ell) - c_i) \right)(-t)^{-5k/2} \\
&\equiv (-t)^{(c_i-1)/2} \sum_{k=0}^{\infty} x^k_{[i]}(-t)^{-5k/2},
\end{aligned}
\tag{154}
$$

where we defined the coefficients

$$x^k_{[i]} \equiv \frac{1}{2} \sum_{\ell=0}^{k} s^{x,k-\ell}_{[i]} x^\ell_{\text{Painlevé}} (5(k-\ell) - c_i), \tag{155}$$

for all $k \in \mathbb{N}$, $i = 1, 2, \dots$ These coefficients can be extended to $i = 0$ defining $x^k_{[0]}$ as the coefficients in the solution (143): $x^0_{[0]} = 1$, $x^1_{[0]} = 1/8$, $x^2_{[0]} = -49/128$, ..., consistently with the definition $c_0 = 2$ obtained earlier.

With this result and using Eqs. (100a), (102), (137) and the substitution of time variables (115c), (99) and (17), we can now compute the asymptotic early-time solution for $r$. It is useful to distinguish the even and odd $i$ indices. For $i$ even we set $i = 2n$ while for $i$ odd we set $i = 2n + 1$. We can split the sum over $i$ into two sums over $n$ which both start from $n = 0$. We obtain

$$
\begin{aligned}
r(\tilde{\tau}) - r_{[0]*} &= \alpha_*^{-3/5}(\beta_*\kappa_*)^{2/5}\eta^{2/5} \sum_{i=0}^{\infty} \eta^{i/5} x_{[i]}(t) \\
&= \sum_{k=0}^{\infty} \sum_{n=0}^{\infty} \frac{(\alpha_*\beta_*\kappa_*)^{\frac{5+n-5k}{10}}}{\alpha_*} \eta^{2k}(\tilde{\tau}_* - \tilde{\tau})^{\frac{-5k+n+1}{2}} \left( x^k_{[2n]} + \frac{x^k_{[2n+1]}}{(\alpha_*\beta_*\kappa_*)^{1/5}} \eta(\tilde{\tau}_* - \tilde{\tau})^{-1} \right) \\
&= \sum_{k=0}^{\infty} \sum_{n=0}^{\infty} \eta^{2k}(\tilde{\tau}_* - \tilde{\tau})^{\frac{-5k+n+1}{2}} \left( \hat{x}^k_{[2n]} + \eta\, \hat{x}^k_{[2n+1]}(\tilde{\tau}_* - \tilde{\tau})^{-1} \right).
\end{aligned}
\tag{156}
$$

In the last step we defined, for convenience, the rescaled

$$\hat{x}^k_{[2n]} \equiv \frac{(\alpha_*\beta_*\kappa_*)^{\frac{5+n-5k}{10}}}{\alpha_*} x^k_{[2n]}, \qquad \hat{x}^k_{[2n+1]} \equiv \frac{(\alpha_*\beta_*\kappa_*)^{\frac{3+n-5k}{10}}}{\alpha_*} x^k_{[2n+1]}. \tag{157}$$

We now turn to the solution for $y_{[i]}$. According to the structure of the source terms for $i = 1, 2, 3, 4, 5, 6$ in Eqs. (237), the asymptotic $t \to -\infty$ behavior of $s^y_{[i]}(t)$ is consistent with

$$s^y_{[i]}(t) = (-t)^{(c_i-1)/2} \sum_{k=0}^{\infty} (-t)^{-5k/2} s^{y,k}_{[i]}. \tag{158}$$

We conjecture this holds for $i \geq 7$.

As $x_{[i]}$ is known, the equation (139a) is an algebraic equation for $y_{[i]}$. Given its complexity, we will more simply integrate Eq. (139c) using Eqs. (154) and (158). We get

$$y_{[i]}(t) = \bar{y}_{[i]} + \int \left( 2x_{[i]}(t') + s^y_{[i]}(t') \right) dt' = (-t)^{(c_i+1)/2} \sum_{k=0}^{\infty} y^k_{[i]} (-t)^{-5k/2}, \tag{159a}$$

$$y^k_{[i]} \equiv \bar{y}_{[i]} \, \delta_{k,(c_i+1)/5} - \sum_{k=0}^{\infty} \frac{2}{c_i + 1 - 5k} \left( 2x^k_{[i]} + s^{y,k}_{[i]} \right), \tag{159b}$$

where $\bar{y}_{[i]}$ is an integration constant that is uniquely fixed from the constraint (139a). Since all terms in Eq. (139a) admit the fall-off behavior (159), we deduce by consistency that $\bar{y}_{[i]} = 0$ for all $i$ except when $i = 4 + 5\mathbb{N}$ where we cannot conclude from this argument, and this coefficient might be non-zero. Consistently with the zeroth-order equation (116c) we define $s^{y,k}_{[0]} = 0$ for all $k \in \mathbb{N}$. The integrated form (159) is then also valid for $i = 0$.

Recalling equation (100c), we are now ready to compute $W$ (defined in Eq. (49)) as $W = \eta^{6/5} Y$. Using Eqs. (102) and (137b) we obtain

$$W = \frac{(\beta_* \kappa_*)^{6/5}}{\alpha_*^{4/5} \gamma_*} \sum_{i=0}^{\infty} \eta^{(6+i)/5} y_{[i]}. \tag{160}$$

We are interested in the asymptotic early-time behaviour of this quantity. We consider Eq. (159) and use Eqs. (154) and (158) together with the substitution of time variables (115c), (99) and (17). Again, it is useful to distinguish the even and odd $i$ indices. For $i$ even we set $i = 2n$ while for $i$ odd we set $i = 2n + 1$. and we split the sum over $i$ into two sums over $n$ which both start from $n = 0$. We obtain

$$W(\tilde{\tau}) = -\sum_{i=0}^{\infty} \sum_{k=0}^{\infty} 2 \frac{(\alpha_* \beta_* \kappa_*)^{\frac{c_i+13-5k}{10}}}{\alpha_*^2 \gamma_*} \frac{\left( 2x^k_{[i]} + s^{y,k}_{[i]} \right)}{c_i + 1 - 5k} \eta^{\frac{4+i-2c_i+10k}{5}} (\tilde{\tau}_* - \tilde{\tau})^{\frac{c_i+1-5k}{2}}$$

$$= \sum_{n=0}^{\infty} \sum_{k=0}^{\infty} \eta^{2k} (\tilde{\tau}_* - \tilde{\tau})^{\frac{n+3-5k}{2}} \left( u^k_{[2n]} + \eta \, u^k_{[2n+1]} (\tilde{\tau}_* - \tilde{\tau})^{-1} \right), \tag{161}$$

where we have defined

$$u^k_{[2n]} \equiv 2 \frac{(\alpha_* \beta_* \kappa_*)^{\frac{n+15-5k}{10}}}{\alpha_*^2 \gamma_*} \frac{\left( 2x^k_{[2n]} + s^{y,k}_{[2n]} \right)}{5k - n - 3}, \tag{162}$$

$$u^k_{[2n+1]} \equiv 2 \frac{(\alpha_* \beta_* \kappa_*)^{\frac{n+13-5k}{10}}}{\alpha_*^2 \gamma_*} \frac{\left( 2x^k_{[2n+1]} + s^{y,k}_{[2n+1]} \right)}{5k - n - 1}. \tag{163}$$

We obtain at leading order

$$W(\tilde{\tau}) = u^0_{[0]} (\tilde{\tau}_* - \tilde{\tau})^{3/2} + O(\tilde{\tau}_* - \tilde{\tau})^2 + O_{\tilde{\tau}}(\eta), \tag{164}$$

with $u^0_{[0]} = -\frac{4\beta_* \kappa_*}{3\gamma_*} \left( \frac{\kappa_* \beta_*}{\alpha_*} \right)^{1/2}$, which matches Eq. (129).

# 5 Inspiral-transition matching

The adiabatic inspiral has as range of validity $\tau < \tau_*$ where the bound arises because the expansion becomes singular at the LSO. The LSO is approached when $\tau_* - \tau \ll \eta^{-1}$ where $\eta^{-1}$ is the radiation reaction timescale. The transition solution is valid in the range (121) and approaches the inspiral at early proper times with respect to the transition timescale $\tau_* - \tau \gg \eta^{-1/5}$. The overlapping region between the inspiral and the transition solution is

$$-\eta^{-1} \ll \tau - \tau_* \ll -\eta^{-1/5}, \tag{165}$$

with $\tau < \tau_*$ which is indeed a subset of the region (121). We will now match the transition solution as $s \to -\infty$ with the inspiral solution as $\tau \to \tau_*$ in the overlapping region using the method of matched asymptotic expansions.

At each order $i \geq 0$, the $i$th post-leading transition equations of motion (116) and (139) require two initial data at $t \to -\infty$ which are determined by the late-time behavior of the quasi-circular inspiral. Since this late-time behavior is consistently described as a series in polynomial powers of the proper time difference with respect to the LSO, we already fixed in Section 4.3 that the two homogeneous oscillating solutions at each order $i > 0$ are identically zero. We shall now verify that the two asymptotic solutions as defined separately in Sections 3 and 4 are consistent with each other.

## 5.1 Leading-order match: 0PA-0PL

Making use of the relation (54), we list equivalent formulae for the coefficients of the inspiral and transition motions

$$\alpha_* = \frac{1}{r_{(0)*}^4}, \qquad \beta_* = \frac{2\sqrt{A_*}\Omega_{(0)*}}{r_{(0)*}^{5/2}}, \qquad \gamma_* = \frac{8\sigma A_*^{1/2}}{3r_{(0)*}^3}, \qquad \pi_* = \frac{r_{(0)*}^{5/2}\alpha_* A_* \Delta_{(0)*}}{2}. \tag{166}$$

In order to perform the match, we first recognize

$$r_{(0)*} = r_{[0]*}, \qquad e_{(0)*} = e_{[0]*}, \qquad \ell_{(0)*} = \ell_{[0]*}, \qquad \Omega_{(0)*} = \Omega_{[0]*}. \tag{167}$$

Let us recall and match the relevant leading-order solutions for the adiabatic inspiral and the leading-order transition motion. For the radial coordinate $r$ we have from the inspiral (Eqs. (16) and (55)) and the transition (Eq. (124)), respectively,

$$r(\tilde{\tau}) = r_{(0)*} + r_{(0),1}^* (\tilde{\tau}_* - \tilde{\tau})^{1/2} + O(\tilde{\tau}_* - \tilde{\tau}) + O_{\tilde{\tau}}(\eta), \tag{168}$$

$$r(\tilde{\tau}) = \left(\frac{\beta_* \kappa_*}{\alpha_*}\right)^{1/2} (\tilde{\tau}_* - \tilde{\tau})^{1/2} + r_{[0]*} + O(\eta^2 (\tilde{\tau}_* - \tilde{\tau})^{-2}) + O_s(\eta^{3/5}), \tag{169}$$

where the coefficient $r_{(0),1}^*$ is given in Eq. (64). The angular momentum in the inspiral (Eqs. (16) and (57a)) and the transition (Eq. (125)), respectively, is given by

$$\ell(\tilde{\tau}) = \ell_{(0)*} + \ell_{(0),2}^* (\tilde{\tau}_* - \tilde{\tau}) + \ell_{(0),3}^* (\tilde{\tau}_* - \tilde{\tau})^{3/2} + O(\tilde{\tau}_* - \tilde{\tau})^2 + O_{\tilde{\tau}}(\eta), \tag{170}$$

$$\ell(\tilde{\tau}) = \lambda_* (\tilde{\tau}_* - \tilde{\tau})^{3/2} + \kappa_* (\tilde{\tau}_* - \tilde{\tau}) + \ell_{[0]}^* + O(\eta^2 (\tilde{\tau}_* - \tilde{\tau})^{-1}) + O_s(\eta^{7/5}), \tag{171}$$

where the coefficients $\ell_{(0),2}^*$, $\ell_{(0),3}^*$, $\kappa_*$ and $\lambda_*$ are defined in Eqs. (66), (67), (106) and (123). Using the relations (166) we are able to match

$$r_{(0),1}^* = \left(\frac{\kappa_* \beta_*}{\alpha_*}\right)^{1/2}, \qquad \ell_{(0),2}^* = \kappa_*, \qquad \ell_{(0),3}^* = \lambda_*. \tag{172}$$

We now show for completeness that also the energy $e$, the angular velocity $\Omega$ and the functions $\delta$ and $W$ which can be constructed from the quantities above, are exactly matched in the overlapping region.

The solutions for the energy in the matching region are the following: for the inspiral and the transition we have, respectively,

$$e(\tilde{\tau}) = e_{(0)*} + \Omega_{(0)*}\ell^*_{(0),2}(\tilde{\tau}_* - \tilde{\tau}) + e^*_{(0),3}(\tilde{\tau}_* - \tilde{\tau})^{3/2} + O(\tilde{\tau}_* - \tilde{\tau})^2 + O_{\tilde{\tau}}(\eta), \tag{173}$$

$$e(\tilde{\tau}) = \Omega_{[0]*}\left(\lambda_* - \frac{4\kappa_*\beta_*}{3\gamma_*}\left(\frac{\kappa_*\beta_*}{\alpha_*}\right)^{1/2}\right)(\tilde{\tau}_* - \tilde{\tau})^{3/2} + \kappa_*\Omega_{[0]*}(\tilde{\tau}_* - \tilde{\tau}) \tag{174}$$

$$+ e^*_{[0]} + O\left(\eta^2(\tilde{\tau}_* - \tilde{\tau})^{-1}\right) + O_s\left(\eta^{7/5}\right), \tag{175}$$

where $e^*_{(0),3}$ is given in Eq. (60). The match is exact once we use the relations (172).

The angular velocity in the inspiral (Eqs. (57c) and (61))

$$\Omega(\tilde{\tau}) = \Omega_{(0)*} - \frac{3}{2}\sigma r^{1/2}_{(0)*}\Omega^2_{(0)*}r^*_{(0),1}(\tilde{\tau}_* - \tilde{\tau})^{1/2} + O(\tilde{\tau}_* - \tilde{\tau}) + O_{\tilde{\tau}}(\eta) \tag{176}$$

is matched by the angular velocity in the transition region (130)

$$\Omega(\tilde{\tau}) = \Omega_{[0]*} - 2\frac{\beta_*\Omega_{[0]*}}{\gamma_*}\left(\frac{\kappa_*\beta_*}{\alpha_*}\right)^{1/2}(\tilde{\tau}_* - \tilde{\tau})^{1/2} + O\left(\eta^2(\tilde{\tau}_* - \tilde{\tau})^{-2}\right) + O_s\left(\eta^{3/5}\right), \tag{177}$$

after using the relations (172) and (166).

The deviation from exact quasi-circularity $W$ in the inspiral (63)

$$W(\tilde{\tau}) = -\sigma r^{1/2}_{(0)*}\Omega_{(0)*}\ell^*_{(0),2}r^*_{(0),1}(\tilde{\tau}_* - \tilde{\tau})^{3/2} + O(\tilde{\tau}_* - \tilde{\tau})^2 + O_{\tilde{\tau}}(\eta) \tag{178}$$

matches with the result from the transition motion (129)

$$W(\tilde{\tau}) = -\frac{4\beta_*}{3\gamma_*}\kappa_*\left(\frac{\kappa_*\beta_*}{\alpha_*}\right)^{1/2}(\tilde{\tau}_* - \tilde{\tau})^{3/2} + O\left(\eta^2(\tilde{\tau}_* - \tilde{\tau})^{-1}\right) + O_s\left(\eta^{7/5}\right), \tag{179}$$

after using the relations (172) and (166). The first nonzero deviation $\delta$ from geodesic circular angular velocity only appears at subleading orders in the mass ratio and will be dealt with it in Section 5.4.

## 5.2 Structure of the subleading match

Since quasi-circular orbits can be parametrized entirely by the Boyer-Lindquist radius $r$ we first consider the match of this quantity. Using the slow-timescale expansion of the radius during the inspiral (18) and the asymptotic solution in the approach towards the LSO (94) we get

$$r(\tilde{\tau}) - r_{(0)*} = \sum_{i=0}^{\infty}\sum_{j=0}^{\infty}\eta^i r^*_{(i),j-q_{(i)}}(\tilde{\tau}_* - \tilde{\tau})^{(j-q_{(i)})/2}. \tag{180}$$

Instead, the transition motion gives the expansion around the LSO (156) from the expansion of the Painlevé transcendental equation,

$$r(\tilde{\tau}) - r_{[0]*} = \sum_{k=0}^{\infty}\sum_{n=0}^{\infty}\eta^{2k}(\tilde{\tau}_* - \tilde{\tau})^{\frac{-5k+n+1}{2}}\left(\hat{x}^k_{[2n]} + \eta\,\hat{x}^k_{[2n+1]}(\tilde{\tau}_* - \tilde{\tau})^{-1}\right). \tag{181}$$

In order to match the $\eta$ powers of these expressions, we recognize the $i$ index of Eq. (180) as either $i = 2k$ for the first term and $i = 2k + 1$ for the second term of the $k$ summation in Eq.

(181). Then, if we match $j$ of Eq. (180) with $n$ of Eq. (181), the powers of $\tilde{\tau}_* - \tilde{\tau}$ match given

$$q_{(2k)} = 5k - 1, \qquad q_{(2k+1)} = 5k + 1, \tag{182}$$

which is precisely the result (95). The matching of the coefficients is given by

$$r^*_{(2k),n-q_{(2k)}} = \hat{x}^k_{[2n]}, \qquad r^*_{(2k+1),n-q_{(2k+1)}} = \hat{x}^k_{[2n+1]}. \tag{183}$$

The structure of the match is visualized in Table 2.

As a further check, we now consider the deviation from exact quasi-circularity $W$. Combining the slow-timescale expansion (50) and the asymptotic solution of $W_{(i)}$ in the approach towards the LSO (97) we obtain

$$W(\tilde{\tau}) = \sum_{i=0}^{\infty} \sum_{j=0}^{\infty} \eta^i w^*_{(i),j-q''_{(i)}} (\tilde{\tau}_* - \tilde{\tau})^{(j-q''_{(i)})/2}. \tag{184}$$

On the other hand, the transition motion gives the expansion around the LSO (161),

$$W(\tilde{\tau}) = \sum_{k=0}^{\infty} \sum_{n=0}^{\infty} \eta^{2k} (\tilde{\tau}_* - \tilde{\tau})^{\frac{n+3-5k}{2}} \left( u^k_{[2n]} + \eta\, u^k_{[2n+1]} (\tilde{\tau}_* - \tilde{\tau})^{-1} \right). \tag{185}$$

In order to match these expressions, we recognize the $i$ index of Eq. (184) as either $i = 2k$ for the first term and $i = 2k + 1$ for the second term of the $k$ summation in Eq. (185). Then, if we match $j$ of Eq. (184) with $n$ of Eq. (185), the powers of $\tilde{\tau}_* - \tilde{\tau}$ match given

$$q''_{(2k)} = 5k - 3, \qquad q''_{(2k+1)} = 5k - 1, \tag{186}$$

which matches with the result (98). The matching conditions for the coefficients read

$$w^*_{(2k),n-q''_{(2k)}} = u^k_{[2n]}, \qquad w^*_{(2k+1),n-q''_{(2k+1)}} = u^k_{[2n+1]}. \tag{187}$$

The matching conditions (183) and (187) provide the precise match in the overlapping region (165) of the inspiral motion with transition-to-plunge motion.

## 5.3 Explicit match of the 1PA inspiral

The 1PA expansion corresponds to the index $i = 1$ in Eqs. (180) and (184) which corresponds to $k = 0$ in the transition (Eqs. (181) and (185)) from the match above. Given $j = 0, 1, 2, \ldots$ we need to consider $n = 0, 1, 2, \ldots$ since $n = j$ from the match. The relevant matching conditions for the coefficients are

$$r^*_{(1),-1} = \hat{x}^0_{[1]}, \qquad r^*_{(1),0} = \hat{x}^0_{[3]}, \qquad r^*_{(1),1} = \hat{x}^0_{[5]}, \qquad \ldots \tag{188}$$

$$w^*_{(1),1} = u^0_{[1]}, \qquad w^*_{(1),2} = u^0_{[3]}, \qquad w^*_{(1),3} = u^0_{[5]}, \qquad \ldots \tag{189}$$

We consider the first of the above matching conditions, namely $r^*_{(1),-1} = \hat{x}^0_{[1]}$. The inspiral coefficient $r^*_{(1),-1}$ is given in Eq. (90). For the inspiral, from Eqs. (157), (152) and (151) we can compute $\hat{x}^0_{[1]} = \frac{F^r_1|_{[0]*}}{2(\alpha_* \beta_* \kappa_*)^{1/2}}$. After using the results (166), (172) and noticing it implies

$$\frac{2 r^{3/2}_{(0)*} A^2_*}{r^*_{(0),1} \Omega_{(0)*} T^*_{15}} = \frac{1}{2(\alpha_* \beta_* \kappa_*)^{1/2}}, \tag{190}$$

Table 2: Visualization of the structure of the matching conditions derived in Eq. (183).

|  | 0PA | 1PA | 2PA | 3PA | ... |
|---|---|---|---|---|---|
| 0PL | $\hat{x}^0_{[0]} = r^*_{(0),1}$ | — | $\hat{x}^1_{[0]} = r^*_{(2),-4}$ | — | ... |
| 1PL | — | $\hat{x}^0_{[1]} = r^*_{(1),-1}$ | — | $\hat{x}^1_{[1]} = r^*_{(3),-6}$ | ... |
| 2PL | $\hat{x}^0_{[2]} = r^*_{(0),2}$ | — | $\hat{x}^1_{[2]} = r^*_{(2),-3}$ | — | ... |
| 3PL | — | $\hat{x}^0_{[3]} = r^*_{(1),0}$ | — | $\hat{x}^1_{[3]} = r^*_{(3),-5}$ | ... |
| 4PL | $\hat{x}^0_{[4]} = r^*_{(0),3}$ | — | $\hat{x}^1_{[4]} = r^*_{(2),-2}$ | — | ... |
| 5PL | — | $\hat{x}^0_{[5]} = r^*_{(1),1}$ | — | $\hat{x}^1_{[5]} = r^*_{(3),-4}$ | ... |
| 6PL | $\hat{x}^0_{[6]} = r^*_{(0),4}$ | — | $\hat{x}^1_{[6]} = r^*_{(2),-1}$ | — | ... |
| 7PL | — | $\hat{x}^0_{[7]} = r^*_{(1),2}$ | — | $\hat{x}^1_{[7]} = r^*_{(3),-3}$ | ... |
| ... | ... | ... | ... | ... | ... |

we recognize the equivalent expression to Eq. (90),

$$r^*_{(1),-1} = \frac{F^r_1\big|_{[0]*}}{2(\alpha_*\beta_*\kappa_*)^{1/2}} .$$
(191)

We then verify the condition $w^*_{(1),1} = u^0_{[1]}$. Substituting the results (64) and (90) into Eq. (88) we obtain $w^*_{(1),1} = 0$. We then compute $u^0_{[1]}$ from the definition (162) and using Eqs. (152)

$$u^0_{[1]} = -\frac{(\alpha_*\beta_*\kappa_*)^{13/10}}{\alpha^2_*\gamma_*}\left(s^x_{[1]} + s^{y,0}_{[1]}\right) .$$
(192)

From Eqs. (238) and (241) we have $s^x_{[1]} = -s^{y,0}_{[1]}$ and therefore also $u^0_{[1]} = 0$. This matching condition is then satisfied.

We proceed with the check of the condition $r^*_{(1),0} = \hat{x}^0_{[3]}$. We solve Eqs. (93) to obtain $r^*_{(1),0}$. Then, substituting Eqs. (64), (65), (67) and (90) gives the final answer

$$r^*_{(1),0} = \left(I_* - \frac{4r^{3/2}_{(0)*}A^2_*}{3\ell^*_{(0),2}\Omega_{(0)*}T^*_{15}}\frac{dF_{\phi 1}}{dr}\bigg|_{(0)*}\right)F^r_1\big|_{(0)*} + \frac{2r^{3/2}_{(0)*}A^2_*}{\Omega_{(0)*}T^*_{15}}\frac{dF^r_1}{dr}\bigg|_{(0)*} .$$
(193)

The coefficient $I_*$ is given in Appendix A. For the transition, using the definition (157) and Eqs. (155) and (240) we obtain

$$\hat{x}^0_{[3]} = -\left(\frac{\alpha_{4*}}{2\alpha^2_*} + \frac{d_{1*}}{3\alpha_*\gamma_*} + \frac{1}{3\alpha_*\kappa_*}\frac{dF_{\phi 1}}{dr}\bigg|_{(0)*}\right)F^r_1\big|_{[0]*} + \frac{1}{2\alpha_*}\frac{dF^r_1}{dr}\bigg|_{[0]*} .$$
(194)

After explicitly evaluating the coefficients we have

$$I_* = -\frac{\alpha_{4*}}{2\alpha^2_*} - \frac{d_{1*}}{3\alpha_*\gamma_*} , \qquad \frac{2r^{3/2}_{(0)*}A^2_*}{\Omega_{(0)*}T^*_{15}} = \frac{1}{2\alpha_*} .$$
(195)

The match is then exact.

Using Eqs. (64), (65), (67), (90) and (193) in the expression (89) for $w^*_{(1),2}$, we obtain

$$w^*_{(1),2} = L_* \left. F^r_1 \right|_{(0)*} \left. \frac{dF_{\phi\,1}}{dr} \right|_{(0)*} . \tag{196}$$

We now compute $u^0_{[3]}$ using the definition (162) and the results (194) and (243) and we obtain

$$u^0_{[3]} = \frac{2\beta_*}{\alpha_*\gamma_*} \left. F^r_1 \right|_{[0]*} \left. \frac{dF_{\phi\,1}}{dr} \right|_{[0]*} . \tag{197}$$

After explicitly evaluating $L_*$ we get

$$L_* = \frac{2\beta_*}{\alpha_*\gamma_*} , \tag{198}$$

and the condition $w^*_{(1),2} = u^0_{[3]}$ is therefore satisfied.

## 5.4 Explicit match of the deviation from the geodesic angular velocity for circular orbits

We consider the deviation $\delta$ (15) from a geodesic angular velocity for circular orbits. For the inspiral motion, the leading-order term in both the small mass ratio expansion and the LSO expansion is (from Eqs. (18), (82) and (87))

$$\eta\,\delta_{(1)}(\tilde{\tau}) + O_{\tilde{\tau}}(\eta^2) = \eta\,\delta^*_{(1),0}(\tilde{\tau} - \tilde{\tau})^0 + \eta\,O(\tilde{\tau}_* - \tilde{\tau})^{1/2} + O_{\tilde{\tau}}(\eta^2)$$
$$= \eta\frac{r^{5/2}_{(0)*}A_*}{2\Delta_{(0)*}} \left. F^r_1 \right|_{(0)*} + \eta\,O(\tilde{\tau}_* - \tilde{\tau})^{1/2} + O_{\tilde{\tau}}(\eta^2) . \tag{199}$$

In the transition region, this is matched by the leading term of $\delta_{[1]} = \frac{2\pi_*}{\Delta^2_{[0]*}} R_{[0]} R_{[1]}$. Using Eqs. (117), (128), (137), (151) and (152) we have

$$\eta^{4/5}\delta_{[0]}(s) + \eta\,\delta_{[1]}(s) + O_s(\eta^2) = \eta\frac{\pi_*}{\alpha_*\Delta^2_{(0)*}} \left. F^r_1 \right|_{[0]*} + \eta^{4/5}O\left(\frac{\eta^{6/5}}{(\tilde{\tau}_* - \tilde{\tau})^{3/2}}\right)$$
$$+ \eta\,O\left(\frac{\eta^2}{(\tilde{\tau}_* - \tilde{\tau})^{5/2}}\right) + O_s(\eta^2) . \tag{200}$$

Using Eq. (166), the two leading coefficients match exactly.

Going to 2PA order for the inspiral solution, substituting the solutions (37), (51), (52), (55), (70) and (78) into Eq. (43) we obtain

$$\eta^2\delta_{(2)} = \eta^2\delta^*_{(2),-3}(\tilde{\tau}_* - \tilde{\tau})^{-3/2} + O\left(\frac{\eta^2}{(\tilde{\tau}_* - \tilde{\tau})}\right) = -\frac{9r^7_{(0)*}T^*_5}{512A^3_*B_*}r^*_{(0),1}\frac{\eta^2}{(\tilde{\tau}_* - \tilde{\tau})^{3/2}} + O\left(\frac{\eta^2}{(\tilde{\tau}_* - \tilde{\tau})}\right) . \tag{201}$$

This leading-order coefficient matches the one in Eq. (128) after using Eqs. (166) and (172) and recognizing

$$-\frac{9r^7_{(0)*}T^*_5}{512A^3_*B_*} = \frac{\pi_*}{4\alpha_*\Delta^2_{[0]*}} . \tag{202}$$

This provides a non-trivial check of the matching. The deviation $\delta$ appears at leading order in the transition-timescale expansion through $\delta_{[0]}$. Its early time behaviour (128) matches terms in the 2PA and higher post-adiabatic inspiral, skipping the 0PA order. This is consistent with the adiabatic result $\delta_{(0)} = 0$.

# 6 Construction of the transition-to-plunge motion from 0PA and 1PA data

The aim of this section is to outline a practical scheme that provides the transition-to-plunge solution which asymptotically matches onto the quasi-circular inspiral that is only known up to adiabatic and first post-adiabatic order in the slow-timescale expansion.

The leading order $i = 0$ transition solution $x_{[0]}(t)$, $y_{[0]}(t)$ is the Ori-Thorne-Kesden solution, which is uniquely fixed by the requirement to have a monotonically decreasing radius. For $i \geq 1$, the functions $x_{[i]}(t)$ and $y_{[i]}(t)$ can be further deduced at all transition times for low $i$ orders from the knowledge of the sources $s^x_{[i]}$ and $s^y_{[i]}$. These are given by Eqs. (236) and (237) using (137) and (138). The sources up to order $i = 10$ depend at most on $f^\mu_{[9]}$. (This is easily seen from a scaling argument as $s^x_{[i]}$ scales as $\eta^{(4+i)/5}$ while $f^\mu_{[i]}$ scales as $\eta^{(5+i)/5}$). Given that the self-forces $f^\mu_{[i]}$ up to $i = 9$ only depend upon $F^\mu_1$ and $F^\mu_2$, see Eq. (103), they can be computed from the first and second-order self-force data. More precisely, one needs to obtain for $\mu = \phi, r$ (either covariant or contravariant indices) the 8 pure numbers at the ISCO:

$$F^\mu_1\big|_{[0]*}, \ \frac{dF^\mu_1}{dr}\bigg|_{[0]*}, \ \dots, \ \frac{d^4F^\mu_1}{dr^4}\bigg|_{[0]*}, \ F^\mu_2\big|_{[0]*}, \ \frac{dF^\mu_2}{dr}\bigg|_{[0]*}, \ \frac{d^2F^\mu_2}{dr^2}\bigg|_{[0]*}. \tag{203}$$

Given that these quantities are obtained in a 0PA-1PA inspiral scheme (for the second order self-force at least the orbit average self-force), one should be able to solve the transition equations exactly up to $i = 10$, though the details remain to be worked out. One could then use these expressions in Eqs. (102) and (137) to deduce the transition motion $R(\eta,s)$, $Y(\eta,s)$.

Moreover, the asymptotic matching provides further information on the transition-to-plunge motion if our assumption is correct that the non-linear sources in the transition motion are of the same order of magnitude or scale lower than the known sources, see Section 3.6.3. Let us start to assume that the solutions for $r_{(0)}(\tilde{\tau})$, $r_{(1)}(\tilde{\tau})$, $W_{(0)}(\tilde{\tau})$ and $W_{(1)}(\tilde{\tau})$ as defined in Eqs. (18) and (50) are known in the LSO limit, that is, we have computed the coefficients $r^*_{(0),1+i}$, $r^*_{(1),-1+i}$, $w^*_{(0),3+i}$ and $w^*_{(1),1+i}$ for $i \geq 0$ as defined in Eqs. (94) and (97). Such coefficients are explicitly given at lowest orders in Eqs. (55), (78), (63) and (83). From the matching conditions (183) and (187) and the definition (157) we obtain $x^0_{[i]}$ and $u^0_{[i]}$, $i \geq 0$. Using Eqs. (162) and (159) we can deduce, in turn, $s^{y,0}_{[i]}$ and then $y^0_{[i]}$, $i \geq 0$. We therefore obtained $x^0_{[i]}$, $y^0_{[i]}$ for all $i \geq 0$. From the definitions (154) and (159), the coefficients $x^0_{[i]}$ and $y^0_{[i]}$ provide the leading-order early-time behaviour of all functions $x_{[i]}(t)$ and $y_{[i]}(t)$ (even without completely knowing the sources of these equations to very high $i$ order due to our limited knowledge on the self-force!).

Finally, a smooth solution interpolating between the inspiral phase and the transition-to-plunge motion can be obtained from the asymptotically matched expansion scheme. The standard composite solution is obtained from the uniform method that consists in adding the inspiral and transition-to-plunge solution and subtracting the common matching values. Such an implementation remains to be worked out. The adiabatic, 1-post-adiabatic and 2-post-

adiabatic composition solutions for the radius read as, respectively,

$$
r^{\text{0PA/2PL}} = \left[r_{(0)}(\tilde{\tau})\right] + \left[r_{[0]*} + \eta^{2/5}\sum_{i=0}^{2}\eta^{i/5}R_{[i]}\left(\frac{\tau - \tau_*}{\eta^{i/5}}\right)\right]
$$
$$
- \left[r_{[0]*} + \sum_{i=1}^{2}r_{(0),i}^*(\tilde{\tau}_* - \tilde{\tau})^{i/2}\right], \tag{204}
$$

$$
r^{\text{1PA/7PL}} = \left[r_{(0)}(\tilde{\tau}) + \eta\, r_{(1)}(\tilde{\tau})\right] + \left[r_{[0]*} + \eta^{2/5}\sum_{i=0}^{7}\eta^{i/5}R_{[i]}\left(\frac{\tau - \tau_*}{\eta^{i/5}}\right)\right]
$$
$$
- \left[r_{[0]*} + \sum_{i=1}^{4}r_{(0),i}^*(\tilde{\tau}_* - \tilde{\tau})^{i/2} + \eta\sum_{i=-1}^{2}r_{(1),i}^*(\tilde{\tau}_* - \tilde{\tau})^{i/2}\right], \tag{205}
$$

$$
r^{\text{2PA/12PL}} = \left[r_{(0)}(\tilde{\tau}) + \eta\, r_{(1)}(\tilde{\tau}) + \eta^2 r_{(2)}(\tilde{\tau})\right] + \left[r_{[0]*} + \eta^{2/5}\sum_{i=0}^{12}\eta^{i/5}R_{[i]}\left(\frac{\tau - \tau_*}{\eta^{i/5}}\right)\right]
$$
$$
- \left[r_{[0]*} + \sum_{i=1}^{7}r_{(0),i}^*(\tilde{\tau}_* - \tilde{\tau})^{i/2} + \eta\sum_{i=-1}^{4}r_{(1),i}^*(\tilde{\tau}_* - \tilde{\tau})^{i/2}\right. \tag{206}
$$
$$
\left. + \eta^2\sum_{i=-4}^{2}r_{(2),i}^*(\tilde{\tau}_* - \tilde{\tau})^{i/2}\right].
$$

The quantities in the last bracket on the right-hand side of the above equations avoid double counting of terms. They also allow to cancel the divergences of the inspiral solution at the ISCO or, equivalently, the early-time divergences of the transition motion.

The 0PA/2PL composite expansion (204) leads to an error in the radius of the order of $O_{\tilde{\tau}}(\eta) + O_s(\eta)$. This composite expansion already improves the leading 0PA/0PL matching [19,21,35,36] which introduces errors of order $O_{\tilde{\tau}}(\eta) + O_s(\eta^{3/5})$. The 1PA/7PL composite expansion (205) leads to an error in the radius of the order of $O_{\tilde{\tau}}(\eta^2) + O_s(\eta^2)$. The 2PA/12PL composite expansion (206) would improve this error to $O_{\tilde{\tau}}(\eta^3) + O_s(\eta^3)$, but it would require third-order self-force data beyond the available second-order self-data (203). The numerical implementation of a waveform generation algorithm, which would allow for a precise accounting of the error made by such a scheme, is left for future work.

# 7  Conclusion

We have considered the quasi-circular inspiral of a point particle around a rotating black hole using a slow-timescale expansion of the dynamical variables. We have worked out the expansion to post-post-adiabatic order including all self-force effects, and sketched the structure of the equations to arbitrary high order in the mass ratio. We have computed the asymptotic solution to the equations of motion in the limit of reaching the last stable orbit, where the quasi-circular inspiral is matched with the transition-to-plunge motion.

In parallel, we have considered the transition-to-plunge motion across the last stable orbit, expanding the equations of motion using a transition-timescale expansion and including self-force corrections. We proved that the solution to arbitrary high order in the mass ratio is controlled by the sourced linearized Painlevé transcendental equation of the first kind. This consolidates and extends previous analyses for equal [20,27–34] and small mass ratios [19, 21,25,26,35,36]. We have computed the asymptotic early-time behaviour of the solution to the equations of motion from the asymptotic behaviour of the Painlevé transcendent.

We have shown the existence of an overlapping region where both the slow-timescale and the transition-timescale expansions are valid. Our main result is the explicit matching between these two expansions in the overlapping region using the method of matched asymptotic expansions. We explicitly performed the leading-order matching of the adiabatic and first post-adiabatic inspiral with the transition motion, and provided a general matching scheme to all perturbative orders in the mass ratio and in the asymptotic solutions. This self-consistent inspiral-transition motion potentially provides a new tool to further calibrate using self-force theory EOB waveforms [39–42] or other waveforms [43–45]. (These waveforms are currently calibrated in part using extreme mass-ratio waveforms calculated using methods from [23,46].)

The deviation from quasi-circularity appears in the inspiral starting from post-adiabatic order. In the transition, the deviation from the geodesic angular velocity for circular orbits needs to be taken into account starting at leading order in the expansion. However, we showed that this leading-order contribution in the transition matches to a departure from quasi-circularity during the inspiral at post-post-adiabatic order, not at adiabatic order [1].

Our treatment of the transition motion to arbitrary high perturbative orders in the mass ratio and the general matching scheme we propose provides a tool to numerically extend the quasi-circular inspiral beyond the last stable orbit. Our analysis was restricted so far to orbits without eccentricity and inclination and would need to be generalized to cover such orbits. The inclusion of finite size effects would require one further extension.

## Acknowledgments

We thank Adam Pound, Niels Warburton and Barry Wardell for their insights, encouragement and comments on the manuscript. We also thank the SciPost referee for pertinent comments. G.C. is Senior Research Associate of the F.R.S.-FNRS and acknowledges support from the FNRS research credit J.0036.20F, bilateral Czech convention PINT-Bilat-M/PGY R.M005.19 and the IISN convention 4.4503.15. L.K. acknowledges support from the ESA Prodex experiment arrangement 4000129178 for the LISA gravitational wave observatory Cosmic Vision L3.

## A  Expressions for the $n$PA auxiliary functions

All quantities defined in this paper have a parity property under the change of signs of both $\sigma$ and $a$. Parity-even quantities are functions of $\sigma a$. We define the following parity-even expressions in terms of $r_{(0)}(\tilde{\tau})$ and $\sigma a$:

$$
\begin{aligned}
T_1 = &-21\sigma a^7 + 115a^6 r_{(0)}^{1/2} - 168\sigma a^5 r_{(0)} - 168a^4 r_{(0)}^{3/2} - 24a^6 r_{(0)}^{3/2} + 740\sigma a^3 r_{(0)}^2 \\
&+ 55\sigma a^5 r_{(0)}^2 - 780a^2 r_{(0)}^{5/2} + 179a^4 r_{(0)}^{5/2} + 288\sigma a r_{(0)}^3 - 728\sigma a^3 r_{(0)}^3 + 856a^2 r_{(0)}^{7/2} \\
&- 44a^4 r_{(0)}^{7/2} - 324\sigma a r_{(0)}^4 + 149\sigma a^3 r_{(0)}^4 - 36r_{(0)}^{9/2} - 139a^2 r_{(0)}^{9/2} - 32\sigma a r_{(0)}^5 \\
&+ 96r_{(0)}^{11/2} - 16a^2 r_{(0)}^{11/2} + 41\sigma a r_{(0)}^6 - 43r_{(0)}^{13/2} + 4r_{(0)}^{15/2},
\end{aligned}
\tag{207}
$$

$$
T_2 = 7\sigma a^3 - 35a^2 r_{(0)}^{1/2} + 36\sigma a r_{(0)} + 2a^2 r_{(0)}^{3/2} - 3\sigma a r_{(0)}^2 - 9r_{(0)}^{5/2} + 2r_{(0)}^{7/2},
\tag{208}
$$

---

[1] The earlier claim [35] that the effect occurs at adiabatic order (which itself leads to a leading-order adiabatic self-force) has been corrected in the Erratum [36].

$$T_3 = -9a^4 + 32\sigma a^3 r_{(0)}^{1/2} - 30a^2 r_{(0)} - 6\sigma a^3 r_{(0)}^{3/2} + 5a^2 r_{(0)}^2 + 12\sigma a r_{(0)}^{5/2}$$
$$- 3a^2 r_{(0)}^3 + 10\sigma a r_{(0)}^{7/2} - 12r_{(0)}^4 + r_{(0)}^5, \tag{209}$$

$$T_4 = 3a^4 - 8\sigma a^3 r_{(0)}^{1/2} + 6a^2 r_{(0)} + 2\sigma a^3 r_{(0)}^{3/2} + 5a^2 r_{(0)}^2 - 12\sigma a r_{(0)}^{5/2}$$
$$+ a^2 r_{(0)}^3 + 2\sigma a r_{(0)}^{7/2} + r_{(0)}^5, \tag{210}$$

$$T_5 = -33\sigma a^7 + 189a^6 r_{(0)}^{1/2} - 328\sigma a^5 r_{(0)} - 62a^4 r_{(0)}^{3/2} - 30a^6 r_{(0)}^{3/2} + 840\sigma a^3 r_{(0)}^2$$
$$+ 75\sigma a^5 r_{(0)}^2 - 960a^2 r_{(0)}^{5/2} + 177a^4 r_{(0)}^{5/2} + 360\sigma a r_{(0)}^3 - 784\sigma a^3 r_{(0)}^3$$
$$+ 900a^2 r_{(0)}^{7/2} - 48a^4 r_{(0)}^{7/2} - 288\sigma a r_{(0)}^4 + 165\sigma a^3 r_{(0)}^4 - 72r_{(0)}^{9/2}$$
$$- 153a^2 r_{(0)}^{9/2} - 48\sigma a r_{(0)}^5 + 114r_{(0)}^{11/2} - 14a^2 r_{(0)}^{11/2} + 41\sigma a r_{(0)}^6 - 45r_{(0)}^{13/2} + 4r_{(0)}^{15/2}, \tag{211}$$

$$T_6 = 4\sigma a^3 - 16a^2 r_{(0)}^{1/2} + 15\sigma a r_{(0)} + 4a^2 r_{(0)}^{3/2} - 6\sigma a r_{(0)}^2 - 3r_{(0)}^{5/2} + 3\sigma a r_{(0)}^3 - 2r_{(0)}^{7/2} + r_{(0)}^{9/2}, \tag{212}$$

$$T_7 = 16a^4 - 27\sigma a^3 r_{(0)}^{1/2} - 24a^2 r_{(0)} + 42\sigma a r_{(0)}^{3/2} + 15\sigma a^3 r_{(0)}^{3/2} - a^2 r_{(0)}^2 - 33\sigma a r_{(0)}^{5/2}$$
$$+ 6r_{(0)}^3 + 5a^2 r_{(0)}^3 + 3\sigma a r_{(0)}^{7/2} - 3r_{(0)}^4 + r_{(0)}^5, \tag{213}$$

$$T_8 = 21a^4 - 85\sigma a^3 r_{(0)}^{1/2} + 114a^2 r_{(0)} - 54\sigma a r_{(0)}^{3/2} + 27\sigma a^3 r_{(0)}^{3/2} - 65a^2 r_{(0)}^2 + 45\sigma a r_{(0)}^{5/2}$$
$$+ 15a^2 r_{(0)}^3 - 29\sigma a r_{(0)}^{7/2} + 12r_{(0)}^4 - r_{(0)}^5, \tag{214}$$

$$T_9 = -33a^8 + 177\sigma a^7 r_{(0)}^{1/2} - 230a^6 r_{(0)} - 382\sigma a^5 r_{(0)}^{3/2} - 63\sigma a^7 r_{(0)}^{3/2} + 1364a^4 r_{(0)}^2$$
$$+ 258a^6 r_{(0)}^2 - 1392\sigma a^3 r_{(0)}^{5/2} - 131\sigma a^5 r_{(0)}^{5/2} + 504a^2 r_{(0)}^3 - 820a^4 r_{(0)}^3 - 30a^6 r_{(0)}^3$$
$$+ 1548\sigma a^3 r_{(0)}^{7/2} + 27\sigma a^5 r_{(0)}^{7/2} - 960a^2 r_{(0)}^4 + 332a^4 r_{(0)}^4 + 144\sigma a r_{(0)}^{9/2} - 893\sigma a^3 r_{(0)}^{9/2}$$
$$+ 794a^2 r_{(0)}^5 - 48a^4 r_{(0)}^5 - 174\sigma a r_{(0)}^{11/2} + 151\sigma a^3 r_{(0)}^{11/2} - 36r_{(0)}^6 - 114a^2 r_{(0)}^6$$
$$- 81\sigma a r_{(0)}^{13/2} + 96r_{(0)}^7 - 14a^2 r_{(0)}^7 + 45\sigma a r_{(0)}^{15/2} - 43r_{(0)}^8 + 4r_{(0)}^9, \tag{215}$$

$$T_{10} = -a^2 + \sigma a r_{(0)}^{1/2} + \sigma a r_{(0)}^{3/2} - 2r_{(0)}^2 + r_{(0)}^3, \tag{216}$$

$$T_{11} = 7a^2 - 9\sigma a r_{(0)}^{1/2} + 15\sigma a r_{(0)}^{3/2} - 18r_{(0)}^2 + 5r_{(0)}^3, \tag{217}$$

$$T_{12} = 2\sigma a - 5r_{(0)}^{1/2} + 3r_{(0)}^{3/2}, \tag{218}$$

$$T_{13} = 16\sigma a^3 - 27a^2 r_{(0)}^{1/2} - 24\sigma a r_{(0)} + 42r_{(0)}^{3/2} + 11a^2 r_{(0)}^{3/2} + 8\sigma a r_{(0)}^2 - 33r_{(0)}^{5/2} + 7r_{(0)}^{7/2}, \tag{219}$$

$$T_{14} = -54\sigma a^7 + 315a^6 r_{(0)}^{1/2} - 556\sigma a^5 r_{(0)} - 106a^4 r_{(0)}^{3/2} - 45a^6 r_{(0)}^{3/2} + 1468\sigma a^3 r_{(0)}^2$$
$$+ 90\sigma a^5 r_{(0)}^2 - 1704a^2 r_{(0)}^{5/2} + 389a^4 r_{(0)}^{5/2} + 648\sigma a r_{(0)}^3 - 1408\sigma a^3 r_{(0)}^3 + 1472a^2 r_{(0)}^{7/2}$$
$$- 81a^4 r_{(0)}^{7/2} - 324\sigma a r_{(0)}^4 + 266\sigma a^3 r_{(0)}^4 - 216r_{(0)}^{9/2} - 187a^2 r_{(0)}^{9/2} - 220\sigma a r_{(0)}^5$$
$$+ 282r_{(0)}^{11/2} - 27a^2 r_{(0)}^{11/2} + 90\sigma a r_{(0)}^6 - 101r_{(0)}^{13/2} + 9r_{(0)}^{15/2}, \tag{220}$$

$$T_{15} = 5\sigma a^5 - 20a^4 r_{(0)}^{1/2} + 4\sigma a^3 r_{(0)} + 54a^2 r_{(0)}^{3/2} + a^4 r_{(0)}^{3/2} - 48\sigma a r_{(0)}^2 + 6\sigma a^3 r_{(0)}^2$$
$$- 38a^2 r_{(0)}^{5/2} + 36\sigma a r_{(0)}^3 + 6r_{(0)}^{7/2} + 2a^2 r_{(0)}^{7/2} - 3\sigma a r_{(0)}^4 - 6r_{(0)}^{9/2} + r_{(0)}^{11/2},$$
(221)

$$T_{16} = r_{(0)}^{5/2}\left(7\sigma a^3 - 28a^2 r_{(0)}^{1/2} + 38\sigma a r_{(0)} - 18r_{(0)}^{3/2} + 5a^2 r_{(0)}^{3/2} - 13\sigma a r_{(0)}^2 + 10r_{(0)}^{5/2} - r_{(0)}^{7/2}\right),$$
(222)

$$T_{17} = 1683a^{14} - 19329\sigma a^{13} r_{(0)}^{1/2} + 88323a^{12} r_{(0)} - 177420\sigma a^{11} r_{(0)}^{3/2} + 3357\sigma a^{13} r_{(0)}^{3/2}$$
$$- 934a^{10} r_{(0)}^2 - 26640a^{12} r_{(0)}^2 + 779608\sigma a^9 r_{(0)}^{5/2} + 49050\sigma a^{11} r_{(0)}^{5/2} - 1624652a^8 r_{(0)}^3$$
$$+ 190122a^{10} r_{(0)}^3 + 1440a^{12} r_{(0)}^3 + 1061792\sigma a^7 r_{(0)}^{7/2} - 1078820\sigma a^9 r_{(0)}^{7/2}$$
$$+ 594\sigma a^{11} r_{(0)}^{7/2} + 1227656a^6 r_{(0)}^4 + 1974338a^8 r_{(0)}^4 - 77229a^{10} r_{(0)}^4$$
$$- 3160272\sigma a^5 r_{(0)}^{9/2} - 711632\sigma a^7 r_{(0)}^{9/2} + 348981\sigma a^9 r_{(0)}^{9/2} + 2889792a^4 r_{(0)}^5$$
$$- 3355152a^6 r_{(0)}^5 - 495075a^8 r_{(0)}^5 + 5280a^{10} r_{(0)}^5 - 1315008\sigma a^3 r_{(0)}^{11/2}$$
$$+ 6651344\sigma a^5 r_{(0)}^{11/2} - 569544\sigma a^7 r_{(0)}^{11/2} - 25197\sigma a^9 r_{(0)}^{11/2} + 248832a^2 r_{(0)}^6$$
$$- 5656920a^4 r_{(0)}^6 + 3063460a^6 r_{(0)}^6 - 18060a^8 r_{(0)}^6 + 2207520\sigma a^3 r_{(0)}^{13/2}$$
$$- 4600832\sigma a^5 r_{(0)}^{13/2} + 362668\sigma a^7 r_{(0)}^{13/2} - 138240a^2 r_{(0)}^7 + 3028792a^4 r_{(0)}^7$$
$$- 970132a^6 r_{(0)}^7 + 6768a^8 r_{(0)}^7 - 108864\sigma a r_{(0)}^{15/2} - 242112\sigma a^3 r_{(0)}^{15/2}$$
$$+ 1034696\sigma a^5 r_{(0)}^{15/2} - 40596\sigma a^7 r_{(0)}^{15/2} - 749160a^2 r_{(0)}^8 - 16700a^4 r_{(0)}^8$$
$$+ 76385a^6 r_{(0)}^8 + 276912\sigma a r_{(0)}^{17/2} - 1011952\sigma a^3 r_{(0)}^{17/2} + 26233\sigma a^5 r_{(0)}^{17/2}$$
$$+ 15552r_{(0)}^9 + 856560a^2 r_{(0)}^9 - 320707a^4 r\_278846a^2 r_{(0)}^{10} + 52680a^4 r_{(0)}^{10}$$
$$- 25752\sigma a r_{(0)}^{21/2} - 62278\sigma a^3 r_{(0)}^{21/2} + 63540r_{(0)}^{11} + 8426a^2 r_{(0)}^{11} - 192a^4 r_{(0)}^{11}$$
$$+ 48860\sigma a r_{(0)}^{23/2} - 702\sigma a^3 r_{(0)}^{23/2} - 33990r_{(0)}^{12} + 6585a^2 r_{(0)}^{12} - 13101\sigma a r_{(0)}^{25/2}$$
$$+ 8867r_{(0)}^{13} - 288a^2 r_{(0)}^{13} + 957\sigma a r_{(0)}^{27/2} - 1068r_{(0)}^{14} + 48r_{(0)}^{15},$$
(223)

$$T_{18} = 149a^{10} - 1209\sigma a^9 r_{(0)}^{1/2} + 2903a^8 r_{(0)} + 1454\sigma a^7 r_{(0)}^{3/2} + 77\sigma a^9 r_{(0)}^{3/2} - 15688a^6 r_{(0)}^2$$
$$+ 56a^8 r_{(0)}^2 + 17172\sigma a^5 r_{(0)}^{5/2} - 3536\sigma a^7 r_{(0)}^{5/2} + 10364a^4 r_{(0)}^3 + 12384a^6 r_{(0)}^3 + 8a^8 r_{(0)}^3$$
$$- 28248\sigma a^3 r_{(0)}^{7/2} - 8630\sigma a^5 r_{(0)}^{7/2} + 266\sigma a^7 r_{(0)}^{7/2} + 13248a^2 r_{(0)}^4 - 22780a^4 r_{(0)}^4$$
$$- 1214a^6 r_{(0)}^4 + 37560\sigma a^3 r_{(0)}^{9/2} - 1326\sigma a^5 r_{(0)}^{9/2} - 9672a^2 r_{(0)}^5 + 12538a^4 r_{(0)}^5$$
$$+ 32a^6 r_{(0)}^5 - 6120\sigma a r_{(0)}^{11/2} - 14670\sigma a^3 r_{(0)}^{11/2} + 224\sigma a^5 r_{(0)}^{11/2} - 4000a^2 r_{(0)}^6$$
$$- 1716a^4 r_{(0)}^6 + 9204\sigma a r_{(0)}^{13/2} + 1488\sigma a^3 r_{(0)}^{13/2} + 396r_{(0)}^7 + 4064a^2 r_{(0)}^7 + 48a^4 r_{(0)}^7$$
$$- 4458\sigma a r_{(0)}^{15/2} - 10\sigma a^3 r_{(0)}^{15/2} - 732r_{(0)}^8 - 759a^2 r_{(0)}^8 + 807\sigma a r_{(0)}^{17/2} + 431r_{(0)}^9$$
$$+ 32a^2 r_{(0)}^9 - 45\sigma a r_{(0)}^{19/2} - 100r_{(0)}^{10} + 8r_{(0)}^{11},$$
(224)

$$
\begin{aligned}
T_{19} =\; & 189a^{12} - 1809\sigma a^{11}r_{(0)}^{1/2} + 5933a^{10}r_{(0)} - 768\sigma a^{9}r_{(0)}^{3/2} + 837\sigma a^{11}r_{(0)}^{3/2} - 53598a^{8}r_{(0)}^{2} \\
& - 6795a^{10}r_{(0)}^{2} + 190264\sigma a^{7}r_{(0)}^{5/2} + 15637\sigma a^{9}r_{(0)}^{5/2} - 350676a^{6}r_{(0)}^{3} + 27231a^{8}r_{(0)}^{3} \\
& + 432a^{10}r_{(0)}^{3} + +397440\sigma a^{5}r_{(0)}^{7/2} - 237576\sigma a^{7}r_{(0)}^{7/2} - 801\sigma a^{9}r_{(0)}^{7/2} - 280728a^{4}r_{(0)}^{4} \\
& + 624284a^{6}r_{(0)}^{4} - 14478a^{8}r_{(0)}^{4} + 114480\sigma a^{3}r_{(0)}^{9/2} - 922632\sigma a^{5}r_{(0)}^{9/2} + 89022\sigma a^{7}r_{(0)}^{9/2} \\
& - 20736a^{2}r_{(0)}^{5} + 837324a^{4}r_{(0)}^{5} - 240234a^{6}r_{(0)}^{5} + 1224a^{8}r_{(0)}^{5} - 456624\sigma a^{3}r_{(0)}^{11/2} \\
& + 372280\sigma a^{5}r_{(0)}^{11/2} - 7046\sigma a^{7}r_{(0)}^{11/2} + 132624a^{2}r_{(0)}^{6} - 346368a^{4}r_{(0)}^{6} + 14142a^{6}r_{(0)}^{6} \\
& - 14256\sigma a r_{(0)}^{13/2} + 179784\sigma a^{3}r_{(0)}^{13/2} - 7086\sigma a^{5}r_{(0)}^{13/2} - 37404a^{2}r_{(0)}^{7} \\
& - 18126a^{4}r_{(0)}^{7} + 1088a^{6}r_{(0)}^{7} - 3024\sigma a r_{(0)}^{15/2} + 37704\sigma a^{3}r_{(0)}^{15/2} - 5370\sigma a^{5}r_{(0)}^{15/2} \\
& + 648r_{(0)}^{8} - 29340a^{2}r_{(0)}^{8} + 10605a^{4}r_{(0)}^{8} + 7560\sigma a r_{(0)}^{17/2} - 10477\sigma a^{3}r_{(0)}^{17/2} + 1188r_{(0)}^{9} \\
& + 3453a^{2}r_{(0)}^{9} + 240a^{4}r_{(0)}^{9} + 2952\sigma a r_{(0)}^{19/2} - 831\sigma a^{3}r_{(0)}^{19/2} - 2466r_{(0)}^{10} + 1613a^{2}r_{(0)}^{10} \\
& - 1959\sigma a r_{(0)}^{21/2} + 1071r_{(0)}^{11} - 48a^{2}r_{(0)}^{11} + 155\sigma a r_{(0)}^{23/2} - 156r_{(0)}^{12} + 8r_{(0)}^{13} ,
\end{aligned}
\tag{225}
$$

$$
\begin{aligned}
T_{20} =\; & -357a^{10} + 3899\sigma a^{9}r_{(0)}^{1/2} - 15939a^{8}r_{(0)} + 27674\sigma a^{7}r_{(0)}^{3/2} - 411\sigma a^{9}r_{(0)}^{3/2} \\
& - 3440a^{6}r_{(0)}^{2} + 3000a^{8}r_{(0)}^{2} - 65388\sigma a^{5}r_{(0)}^{5/2} - 4104\sigma a^{7}r_{(0)}^{5/2} + 109284a^{4}r_{(0)}^{3} \\
& - 19064a^{6}r_{(0)}^{3} - 96a^{8}r_{(0)}^{3} - 76392\sigma a^{3}r_{(0)}^{7/2} + 75822\sigma a^{5}r_{(0)}^{7/2} - 282\sigma a^{7}r_{(0)}^{7/2} \\
& + 20736a^{2}r_{(0)}^{4} - 105516a^{4}r_{(0)}^{4} + 5790a^{6}r_{(0)}^{4} + 57816\sigma a^{3}r_{(0)}^{9/2} - 18318\sigma a^{5}r_{(0)}^{9/2} \\
& + 1800a^{2}r_{(0)}^{5} + 18166a^{4}r_{(0)}^{5} - 272a^{6}r_{(0)}^{5} - 9720\sigma a r_{(0)}^{11/2} + 7782\sigma a^{3}r_{(0)}^{11/2} \\
& + 804\sigma a^{5}r_{(0)}^{11/2} - 24744a^{2}r_{(0)}^{6} + 1732a^{4}r_{(0)}^{6} + 10836\sigma a r_{(0)}^{13/2} - 8504\sigma a^{3}r_{(0)}^{13/2} \\
& + 1620r_{(0)}^{7} + 9720a^{2}r_{(0)}^{7} - 240a^{4}r_{(0)}^{7} - 1710\sigma a r_{(0)}^{15/2} + 842\sigma a^{3}r_{(0)}^{15/2} - 2556r_{(0)}^{8} \\
& - 633a^{2}r_{(0)}^{8} - 813\sigma a r_{(0)}^{17/2} + 1293r_{(0)}^{9} - 48a^{2}r_{(0)}^{9} + 167\sigma a r_{(0)}^{19/2} \\
& - 252r_{(0)}^{10} + 16r_{(0)}^{11} .
\end{aligned}
\tag{226}
$$

We define the following parity-even expressions defined at the LSO in terms of $r_{(0)*}$ and $\sigma a$:

$$
\begin{aligned}
E_{*} =\; & -77\sigma a^{5} + 423a^{4}r_{(0)*}^{1/2} - 861\sigma a^{3}r_{(0)*} + 774a^{2}r_{(0)*}^{3/2} - 136a^{4}r_{(0)*}^{3/2} - 270\sigma a r_{(0)*}^{2} \\
& + 540\sigma a^{3}r_{(0)*}^{2} - 693a^{2}r_{(0)*}^{5/2} + 315\sigma a r_{(0)*}^{3} - 85\sigma a^{3}r_{(0)*}^{3} + 196a^{2}r_{(0)*}^{7/2} \\
& - 129\sigma a r_{(0)*}^{4} - 35a^{2}r_{(0)*}^{9/2} + 55\sigma a r_{(0)*}^{5} - 18r_{(0)*}^{11/2} + r_{(0)*}^{13/2} ,
\end{aligned}
\tag{227}
$$

$$
F_{*} = -36\sigma a^{3} + 145a^{2}r_{(0)*}^{1/2} - 192\sigma a r_{(0)*} + 90r_{(0)*}^{3/2} - 45a^{2}r_{(0)*}^{3/2} + 100\sigma a r_{(0)*}^{2}
\tag{228}
$$

$$
- 69r_{(0)*}^{5/2} + 7r_{(0)*}^{7/2} ,
\tag{229}
$$

$$
G_{*} = \frac{3r_{(0)*}^{7/2}\left(16\sigma a - 21r_{(0)*}^{1/2} + 5r_{(0)*}^{3/2}\right)}{16A_{*}^{3/2}} ,
\tag{230}
$$

$$H_* = -\frac{r_{(0)*}^2}{32A_*^{7/2}}\left(-555a^4 + 3040\sigma a^3 r_{(0)*}^{1/2} - 6153a^2 r_{(0)*} + 5472\sigma a r_{(0)*}^{3/2} - 960\sigma a^3 r_{(0)*}^{3/2}\right.$$

$$-1890r_{(0)*}^2 + 3780a^2 r_{(0)*}^2 - 4704\sigma a r_{(0)*}^{5/2} + 2115r_{(0)*}^3 - 645a^2 r_{(0)*}^3 + 1248\sigma a r_{(0)*}^{7/2}$$

$$\left.-807r_{(0)*}^4 + 59r_{(0)*}^5\right),$$

$$(231)$$

$$I_* = \frac{r_{(0)*}^{7/2}}{12(r_{(0)*}^2 - 7r_{(0)*} + 10\sigma a r_{(0)*}^{1/2} - 4a^2)T_{15}^*}\left(-749\sigma a^5 + 3975a^4 r_{(0)*}^{1/2} - 8013\sigma a^3 r_{(0)*}\right.$$

$$+7290a^2 r_{(0)*}^{3/2} - 1912a^4 r_{(0)*}^{3/2} - 2538\sigma a r_{(0)*}^2 + 9468\sigma a^3 r_{(0)*}^2 - 17661a^2 r_{(0)*}^{5/2}$$

$$+14679\sigma a r_{(0)*}^3 - 1285\sigma a^3 r_{(0)*}^3 - 4536r_{(0)*}^{7/2} + 5224a^2 r_{(0)*}^{7/2} - 7269\sigma a r_{(0)*}^4$$

$$\left.+3420r_{(0)*}^{9/2} - 275a^2 r_{(0)*}^{9/2} + 955\sigma a r_{(0)*}^5 - 834r_{(0)*}^{11/2} + 61r_{(0)*}^{13/2}\right),$$

$$(232)$$

$$L_* = -\frac{r_{(0)*}^5 \Omega_{(0)*}}{72\Delta_{(0)*}(r_{(0)*}^2 - 7r_{(0)*} + 10\sigma a r_{(0)*}^{1/2} - 4a^2)^2 T_{15}^*}\left(7347\sigma a^9 - 59492a^8 r_{(0)*}^{1/2}\right.$$

$$+191450\sigma a^7 r_{(0)*} - 276400a^6 r_{(0)*}^{3/2} + 11643a^8 r_{(0)*}^{3/2} + 47812\sigma a^5 r_{(0)*}^2 - 86570\sigma a^7 r_{(0)*}^2$$

$$+430560a^4 r_{(0)*}^{5/2} + 229792a^6 r_{(0)*}^{5/2} - 648072\sigma a^3 r_{(0)*}^3 - 151910\sigma a^5 r_{(0)*}^3 + 4452\sigma a^7 r_{(0)*}^3$$

$$+405216a^2 r_{(0)*}^{7/2} - 498168a^4 r_{(0)*}^{7/2} - 27282a^6 r_{(0)*}^{7/2} - 98496\sigma a r_{(0)*}^4 + 1293600\sigma a^3 r_{(0)*}^4$$

$$+17328\sigma a^5 r_{(0)*}^4 - 1321920a^2 r_{(0)*}^{9/2} + 232332a^4 r_{(0)*}^{9/2} + 102a^6 r_{(0)*}^{9/2} + 646920\sigma a r_{(0)*}^5$$

$$-749706\sigma a^3 r_{(0)*}^5 - 124\sigma a^5 r_{(0)*}^5 - 123120r_{(0)*}^{11/2} + 1006968a^2 r_{(0)*}^{11/2} - 26300a^4 r_{(0)*}^{11/2}$$

$$-648036\sigma a r_{(0)*}^6 + 136714\sigma a^3 r_{(0)*}^6 + 163728r_{(0)*}^{13/2} - 271504a^2 r_{(0)*}^{13/2} - 142a^4 r_{(0)*}^{13/2}$$

$$+242454\sigma a r_{(0)*}^7 - 5188\sigma a^3 r_{(0)*}^7 - 81648r_{(0)*}^{15/2} + 25034a^2 r_{(0)*}^{15/2} - 38403\sigma a r_{(0)*}^8$$

$$\left.+19368r_{(0)*}^{17/2} - 462a^2 r_{(0)*}^{17/2} + 2268\sigma a r_{(0)*}^9 - 2247r_{(0)*}^{19/2} + 102r_{(0)*}^{21/2}\right).$$

$$(233)$$

# B   The deviations $\delta_{[i]}$

The functions $\delta_{[i]}$, $i = 1, 2, \ldots, 6$ read as follows:

$$\delta_{[1]} = \frac{2\pi_*}{\Delta_{[0]*}^2}R_{[0]}R_{[1]},$$
$$(234a)$$

$$\delta_{[2]} = \frac{\rho_*}{16r_{[0]*}^{3/2}\Delta_{[0]*}^3}R_{[0]}^3 + \frac{\pi_*}{\Delta_{[0]*}^2}\left(R_{[1]}^2 + 2R_{[0]}R_{[2]}\right) - \frac{2\Omega_{[0]*}A_*^{5/2}}{r_{[0]*}^2\Delta_{[0]*}^2}R_{[0]}\xi_{[0]}$$
$$(234b)$$

$$-\frac{\Omega_{[0]*}A_*^{3/2}}{\Delta_{[0]*}}\xi_{[2]} + \frac{\Omega_{[0]*}A_*^{1/2}B_*}{\Delta_{[0]*}}Y_{[0]},$$
$$(234c)$$

$$\delta_{[3]} = \frac{3\rho_*}{16 r_{[0]*}^{3/2} \Delta_{[0]*}^3} R_{[0]}^2 R_{[1]} + \frac{2\pi_*}{\Delta_{[0]*}^2} \left( R_{[0]} R_{[3]} + R_{[1]} R_{[2]} \right) - \frac{2\Omega_{[0]*} A_*^{5/2}}{r_{[0]*}^2 \Delta_{[0]*}^2} R_{[1]} \xi_{[0]}$$

$$- \frac{\Omega_{[0]*} A_*^{3/2}}{\Delta_{[0]*}} \xi_{[3]} + \frac{\Omega_{[0]*} A_*^{1/2} B_*}{\Delta_{[0]*}} Y_{[1]}, \tag{234d}$$

$$\delta_{[4]} = \frac{\lambda_*}{128 r_{[0]*}^{5/2} \Delta_{[0]*}^4} R_{[0]}^4 + \frac{3\rho_*}{16 r_{[0]*}^{3/2} \Delta_{[0]*}^3} \left( R_{[0]}^2 R_{[2]} + R_{[1]}^2 R_{[0]} \right)$$

$$+ \frac{\pi_*}{\Delta_{[0]*}^2} \left( R_{[2]}^2 + 2R_{[0]} R_{[4]} + 2R_{[1]} R_{[3]} \right) - \frac{\Omega_{[0]*} A_*^{1/2} \nu_*}{\Delta_{[0]*}^3} R_{[0]}^2 \xi_{[0]}$$

$$- \frac{2\Omega_{[0]*} A_*^{5/2}}{r_{[0]*}^2 \Delta_{[0]*}^2} \left( R_{[0]} \xi_{[2]} + R_{[2]} \xi_{[0]} \right) - \frac{\Omega_{[0]*} A_*^{3/2}}{\Delta_{[0]*}} \xi_{[4]} + \frac{\Omega_{[0]*}^2 A_*^2 C_*}{\Delta_{[0]*}^2} \xi_{[0]}^2$$

$$+ \frac{2\Omega_{[0]*} A_*^{3/2} B_*}{r_{[0]*}^2 \Delta_{[0]*}^2} R_{[0]} Y_{[0]} + \frac{\Omega_{[0]*} A_*^{1/2} B_*}{\Delta_{[0]*}} Y_{[2]}, \tag{234e}$$

$$\delta_{[5]} = \frac{\lambda_*}{32 r_{[0]*}^{5/2} \Delta_{[0]*}^4} R_{[0]}^3 R_{[1]} + \frac{\rho_*}{16 r_{[0]*}^{3/2} \Delta_{[0]*}^3} \left( R_{[1]}^3 + 3R_{[0]}^2 R_{[3]} + 6R_{[0]} R_{[1]} R_{[2]} \right)$$

$$+ \frac{2\pi_*}{\Delta_{[0]*}^2} \left( R_{[0]} R_{[5]} + R_{[1]} R_{[4]} + R_{[2]} R_{[3]} \right) - \frac{2\Omega_{[0]*} A_*^{1/2} \nu_*}{\Delta_{[0]*}^3} R_{[0]} R_{[1]} \xi_{[0]}$$

$$- \frac{2\Omega_{[0]*} A_*^{5/2}}{r_{[0]*}^2 \Delta_{[0]*}^2} \left( R_{[0]} \xi_{[3]} + R_{[1]} \xi_{[2]} + R_{[3]} \xi_{[0]} \right) - \frac{\Omega_{[0]*} A_*^{3/2}}{\Delta_{[0]*}} \xi_{[5]}$$

$$+ \frac{2\Omega_{[0]*} A_*^{3/2} B_*}{r_{[0]*}^2 \Delta_{[0]*}^2} \left( R_{[0]} Y_{[1]} + R_{[1]} Y_{[0]} \right) + \frac{\Omega_{[0]*} A_*^{1/2} B_*}{\Delta_{[0]*}} Y_{[3]}, \tag{234f}$$

$$\delta_{[6]} = \frac{\chi_*}{256 r_{[0]*}^{7/5} \Delta_{[0]*}^5} R_{[0]}^5 + \frac{\lambda_*}{64 r_{[0]*}^{5/2} \Delta_{[0]*}^4} \left( 2R_{[0]}^3 R_{[2]} + 3R_{[0]}^2 R_{[1]}^2 \right)$$

$$+ \frac{3\rho_*}{16 r_{[0]*}^{3/2} \Delta_{[0]*}^3} \left( R_{[0]}^2 R_{[4]} + R_{[1]}^2 R_{[2]} + R_{[2]}^2 R_{[0]} + 2R_{[0]} R_{[1]} R_{[3]} \right)$$

$$+ \frac{\pi_*}{\Delta_{[0]*}^2} \left( R_{[3]}^2 + 2R_{[0]} R_{[6]} + 2R_{[1]} R_{[5]} + 2R_{[2]} R_{[4]} \right) - \frac{4\Omega_{[0]*} A_*^{3/2} \mu_*}{r_{[0]*}^{5/2} \Delta_{[0]*}^4} R_{[0]}^3 \xi_{[0]}$$

$$- \frac{\Omega_{[0]*} A_*^{1/2} \nu_*}{\Delta_{[0]*}^3} \left( R_{[0]}^2 \xi_{[2]} + R_{[1]}^2 \xi_{[0]} + 2R_{[0]} R_{[2]} \xi_{[0]} \right)$$

$$- \frac{2\Omega_{[0]*} A_*^{5/2}}{r_{[0]*}^2 \Delta_{[0]*}^2} \left( R_{[0]} \xi_{[4]} + R_{[1]} \xi_{[3]} + R_{[2]} \xi_{[2]} + R_{[4]} \xi_{[0]} \right) + \frac{2\Omega_{[0]*}^2 A_*^3 \varphi_*}{r_{[0]*}^2 \Delta_{[0]*}^3} R_{[0]} \xi_{[0]}^2$$

$$- \frac{\Omega_{[0]*} A_*^{3/2}}{\Delta_{[0]*}} \xi_{[6]} + \frac{2\Omega_{[0]*}^2 A_*^2 C_*}{\Delta_{[0]*}^2} \xi_{[0]} \xi_{[2]} + \frac{\Omega_{[0]*} A_*^{1/2} B_* \psi_*}{r_{[0]*}^{3/2} \Delta_{[0]*}^3} R_{[0]}^2 Y_{[0]}$$

$$+ \frac{2\Omega_{[0]*} A_*^{3/2} B_*}{r_{[0]*}^2 \Delta_{[0]*}^2} \left( R_{[0]} Y_{[2]} + R_{[1]} Y_{[1]} + R_{[2]} Y_{[0]} \right) + \frac{\Omega_{[0]*} A_*^{1/2} B_*}{\Delta_{[0]*}} Y_{[4]} - \frac{\Omega_{[0]*}^2 A_* \phi_*}{\Delta_{[0]*}^2} \xi_{[0]} Y_{[0]}. \tag{234g}$$

All coefficients are listed in Appendix D.

# C  Source to the transition equations

The subleading transition equations are given by Eqs. (135a), (135c) and (135d). In the following we will denote $' = d/ds$. The source terms for low values of $i = 1, 2, 3, 4, 5, 6$ are

$$S^{RY}_{[1]} = 0\,, \tag{235a}$$

$$S^{RY}_{[2]} = -\frac{\alpha_{4*}}{4}R^4_{[0]} - \alpha_* R^2_{[1]}R_{[0]} + \frac{\beta_{2*}}{2}R^2_{[0]}\xi_{[0]} + \beta_* R_{[0]}\xi_{[2]} + \frac{q_{0*}}{2}\xi^2_{[0]} - d_{1*}R_{[0]}Y_{[0]}\,, \tag{235b}$$

$$\begin{aligned}
S^{RY}_{[3]} = &-\alpha_{4*}R^3_{[0]}R_{[1]} - \alpha_*\left(\frac{1}{3}R^3_{[1]} + 2R_{[0]}R_{[1]}R_{[2]}\right) + \beta_{2*}R_{[0]}R_{[1]}\xi_{[0]} \\
&+ \beta_*\left(R_{[0]}\xi_{[3]} + R_{[1]}\xi_{[2]}\right) - d_{1*}\left(R_{[0]}Y_{[1]} + R_{[1]}Y_{[0]}\right)\,,
\end{aligned} \tag{235c}$$

$$\begin{aligned}
S^{RY}_{[4]} = &-\frac{\alpha_{5*}}{5}R^5_{[0]} - \alpha_{4*}\left(R^3_{[0]}R_{[2]} + \frac{3}{2}R^2_{[0]}R^2_{[1]}\right) - \alpha_*\left(R^2_{[1]}R_{[2]} + R^2_{[2]}R_{[0]} + 2R_{[0]}R_{[1]}R_{[3]}\right) \\
&+ \frac{\beta_{3*}}{3}R^3_{[0]}\xi_{[0]} + \frac{\beta_{2*}}{2}\left(R^2_{[0]}\xi_{[2]} + R^2_{[1]}\xi_{[0]} + 2R_{[0]}R_{[2]}\xi_{[0]}\right) \\
&+ \beta_*\left(R_{[0]}\xi_{[4]} + R_{[1]}\xi_{[3]} + R_{[2]}\xi_{[2]}\right) + q_{1*}R_{[0]}\xi^2_{[0]} \\
&+ q_{0*}\xi_{[0]}\xi_{[2]} - \frac{d_{2*}}{2}R^2_{[0]}Y_{[0]} - d_{1*}\left(R_{[0]}Y_{[2]} + R_{[1]}Y_{[1]} + R_{[2]}Y_{[0]}\right) + \frac{u_{0*}}{2}\xi_{[0]}Y_{[0]}\,,
\end{aligned} \tag{235d}$$

$$\begin{aligned}
S^{RY}_{[5]} = &-\alpha_{5*}R^4_{[0]}R_{[1]} - \alpha_{4*}\left(R^3_{[0]}R_{[3]} + R^3_{[1]}R_{[0]} + 3R^2_{[0]}R_{[1]}R_{[2]}\right), \\
&-\alpha_*\left(R^2_{[1]}R_{[3]} + R^2_{[2]}R_{[1]} + 2R_{[0]}R_{[1]}R_{[4]} + 2R_{[0]}R_{[2]}R_{[3]}\right) + \beta_{3*}R^2_{[0]}R_{[1]}\xi_{[0]} \\
&+ \frac{\beta_{2*}}{2}\left(R^2_{[0]}\xi_{[3]} + 2R_{[0]}R_{[3]}\xi_{[0]} + 2R_{[0]}R_{[1]}\xi_{[2]} + 2R_{[1]}R_{[2]}\xi_{[0]}\right) \\
&+ \beta_*\left(R_{[0]}\xi_{[5]} + R_{[1]}\xi_{[4]} + R_{[2]}\xi_{[3]} + R_{[3]}\xi_{[2]}\right) + q_{1*}R_{[1]}\xi^2_{[0]} + q_{0*}\xi_{[0]}\xi_{[3]} \\
&- \frac{d_{2*}}{2}\left(R^2_{[0]}Y_{[1]} + 2R_{[0]}R_{[1]}Y_{[0]}\right) - d_{1*}\left(R_{[0]}Y_{[3]} + R_{[1]}Y_{[2]} + R_{[2]}Y_{[1]} + R_{[3]}Y_{[0]}\right) \\
&+ \frac{u_{0*}}{2}\xi_{[0]}Y_{[1]}\,,
\end{aligned} \tag{235e}$$

$$\begin{aligned}
S^{RY}_{[6]} = &-\frac{\alpha_{6*}}{6}R^6_{[0]} - \alpha_{5*}\left(R^4_{[0]}R_{[2]} + 2R^3_{[0]}R^2_{[1]}\right) \\
&-\alpha_{4*}\left(\frac{1}{4}R^4_{[1]} + R^3_{[0]}R_{[4]} + \frac{3}{2}R^2_{[0]}R^2_{[2]} + 3R^2_{[0]}R_{[1]}R_{[3]} + 3R^2_{[1]}R_{[0]}R_{[2]}\right) \\
&-\alpha_*\left(\frac{1}{3}R^3_{[2]} + R^2_{[1]}R_{[4]} + R^2_{[3]}R_{[0]} + 2R_{[0]}R_{[1]}R_{[5]} + 2R_{[0]}R_{[2]}R_{[4]} + 2R_{[1]}R_{[2]}R_{[3]}\right) \\
&+ \frac{\beta_{4*}}{4}R^4_{[0]}\xi_{[0]} + \frac{\beta_{3*}}{3}\left(R^3_{[0]}\xi_{[2]} + 3R^2_{[0]}R_{[2]}\xi_{[0]} + 3R^2_{[1]}R_{[0]}\xi_{[0]}\right) \\
&+ \frac{\beta_{2*}}{2}\left(R^2_{[0]}\xi_{[4]} + R^2_{[1]}\xi_{[2]} + R^2_{[2]}\xi_{[0]} + 2R_{[0]}R_{[4]}\xi_{[0]} + 2R_{[0]}R_{[1]}\xi_{[3]}\right. \\
&\left. + 2R_{[1]}R_{[3]}\xi_{[0]} + 2R_{[0]}R_{[2]}\xi_{[2]}\right) \\
&+ \beta_*\left(R_{[0]}\xi_{[6]} + R_{[1]}\xi_{[5]} + R_{[2]}\xi_{[4]} + R_{[3]}\xi_{[3]} + R_{[4]}\xi_{[2]}\right) + \frac{q_{2*}}{2}R^2_{[0]}\xi^2_{[0]} \\
&+ q_{1*}\left(R_{[2]}\xi^2_{[0]} + 2R_{[0]}\xi_{[0]}\xi_{[2]}\right) + \frac{q_{0*}}{2}\left(\xi^2_{[2]} + 2\xi_{[0]}\xi_{[4]}\right) - \frac{d_{3*}}{3}R^3_{[0]}Y_{[0]} \\
&- \frac{d_{2*}}{2}\left(R^2_{[0]}Y_{[2]} + R^2_{[1]}Y_{[0]} + 2R_{[0]}R_{[2]}Y_{[0]} + 2R_{[0]}R_{[1]}Y_{[1]}\right) + \\
&- d_{1*}\left(R_{[0]}Y_{[4]} + R_{[1]}Y_{[3]} + R_{[2]}Y_{[2]} + R_{[3]}Y_{[1]} + R_{[4]}Y_{[0]}\right) + u_{1*}R_{[0]}Y_{[0]}\xi_{[0]} \\
&+ \frac{u_{0*}}{2}\left(\xi_{[0]}Y_{[2]} + \xi_{[2]}Y_{[0]}\right) + \frac{v_*}{4}Y^2_{[0]}\,,
\end{aligned} \tag{235f}$$

$$S^R_{[1]} = f^r_{[0]},$$ (236a)

$$S^R_{[2]} = -\alpha_{4*}R^3_{[0]} - \alpha_* R^2_{[1]} + \beta_{2*}R_{[0]}\xi_{[0]} + \beta_*\xi_{[2]} - d_{1*}Y_{[0]},$$ (236b)

$$S^R_{[3]} = -3\alpha_{4*}R^2_{[0]}R_{[1]} - 2\alpha_*R_{[1]}R_{[2]} + \beta_{2*}R_{[1]}\xi_{[0]} + \beta_*\xi_{[3]} - d_{1*}Y_{[1]} + f^r_{[2]},$$ (236c)

$$S^R_{[4]} = -\alpha_{5*}R^4_{[0]} - 3\alpha_{4*}\left(R^2_{[0]}R_{[2]} + R^2_{[1]}R_{[0]}\right) - \alpha_*\left(R^2_{[2]} + 2R_{[1]}R_{[3]}\right)$$
$$+ \beta_{3*}R^2_{[0]}\xi_{[0]} + \beta_{2*}\left(R_{[0]}\xi_{[2]} + R_{[2]}\xi_{[0]}\right) + \beta_*\xi_{[4]} + q_{1*}\xi^2_{[0]}$$ (236d)
$$- d_{2*}R_{[0]}Y_{[0]} - d_{1*}Y_{[2]} + f^r_{[3]},$$

$$S^R_{[5]} = -4\alpha_{5*}R^3_{[0]}R_{[1]} - \alpha_{4*}\left(R^3_{[1]} + 3R^2_{[0]}R_{[3]} + 6R_{[0]}R_{[1]}R_{[2]}\right) - 2\alpha_*\left(R_{[1]}R_{[4]} + R_{[2]}R_{[3]}\right)$$
$$+ 2\beta_{3*}R_{[0]}R_{[1]}\xi_{[0]} + \beta_{2*}\left(R_{[0]}\xi_{[3]} + R_{[1]}\xi_{[2]} + R_{[3]}\xi_{[0]}\right) + \beta_*\xi_{[5]}$$
$$- d_{2*}\left(R_{[1]}Y_{[0]} + R_{[0]}Y_{[1]}\right) - d_{1*}Y_{[3]} + f^r_{[4]},$$ (236e)

$$S^R_{[6]} = -\alpha_{6*}R^5_{[0]} - 2\alpha_{5*}\left(2R^3_{[0]}R_{[2]} + 3R^2_{[0]}R^2_{[1]}\right)$$
$$- 3\alpha_{4*}\left(R^2_{[0]}R_{[4]} + R^2_{[1]}R_{[2]} + R^2_{[2]}R_{[0]} + 2R_{[0]}R_{[1]}R_{[3]}\right)$$
$$- \alpha_*\left(R^2_{[3]} + 2R_{[1]}R_{[5]} + 2R_{[2]}R_{[4]}\right) + \beta_{4*}R^3_{[0]}\xi_{[0]}$$
$$+ \beta_{3*}\left(R^2_{[0]}\xi_{[2]} + R^2_{[1]}\xi_{[0]} + 2R_{[0]}R_{[2]}\xi_{[0]}\right)$$ (236f)
$$+ \beta_{2*}\left(R_{[0]}\xi_{[4]} + R_{[1]}\xi_{[3]} + R_{[2]}\xi_{[2]} + R_{[4]}\xi_{[0]}\right) + \beta_*\xi_{[6]} + q_{2*}R_{[0]}\xi^2_{[0]}$$
$$+ 2q_{1*}\xi_{[0]}\xi_{[2]} - d_{3*}R^2_{[0]}Y_{[0]} - d_{2*}\left(R_{[0]}Y_{[2]} + R_{[1]}Y_{[1]} + R_{[2]}Y_{[0]}\right) - d_{1*}Y_{[4]}$$
$$+ u_{1*}\xi_{[0]}Y_{[0]} + f^r_{[5]},$$

$$S^Y_{[1]} = \frac{2}{\gamma_*}f^r_{[0]}R'_{[0]},$$ (237a)

$$S^Y_{[2]} = -\frac{2\beta_*}{\gamma_*}R_{[0]}\xi'_{(2)} - \frac{\beta_{2*}}{\gamma_*}R^2_{[0]}\xi'_{[0]} - \frac{2q_{0*}}{\gamma_*}\xi_{[0]}\xi'_{[0]} + \frac{2d_{1*}}{\gamma_*}R_{[0]}Y'_{[0]} + \frac{2}{\gamma_*}f^r_{[0]}R'_{[1]},$$ (237b)

$$S^Y_{[3]} = -\frac{2\beta_*}{\gamma_*}\left(R_{[0]}\xi'_{(3)} + R_{[1]}\xi'_{(2)}\right) - \frac{2\beta_{2*}}{\gamma_*}R_{[0]}R_{[1]}\xi'_{[0]} + \frac{2d_{1*}}{\gamma_*}\left(R_{[0]}Y'_{[1]} + R_{[1]}Y'_{[0]}\right)$$
$$+ \frac{2}{\gamma_*}\left(f^r_{[0]}R'_{[2]} + f^r_{[2]}R'_{[0]}\right),$$ (237c)

$$S^Y_{[4]} = -\frac{2\beta_{3*}}{3\gamma_*}R^3_{[0]}\xi'_{[0]} - \frac{\beta_{2*}}{\gamma_*}\left(R^2_{[0]}\xi'_{[2]} + R^2_{[1]}\xi'_{[0]} + 2R_{[0]}R_{[2]}\xi'_{[0]}\right)$$
$$- \frac{2\beta_*}{\gamma_*}\left(R_{[0]}\xi'_{[4]} + R_{[1]}\xi'_{[3]} + R_{[2]}\xi'_{[2]}\right) - \frac{4q_{1*}}{\gamma_*}R_{[0]}\xi_{[0]}\xi'_{[0]} - \frac{2q_{0*}}{\gamma_*}\left(\xi_{[2]}\xi'_{[0]} + \xi'_{[2]}\xi_{[0]}\right)$$
$$+ \frac{d_{2*}}{\gamma_*}R^2_{[0]}Y'_{[0]} + \frac{2d_{1*}}{\gamma_*}\left(R_{[0]}Y'_{[2]} + R_{[1]}Y'_{[1]} + R_{[2]}Y'_{[0]}\right)$$
$$- \frac{u_{0*}}{\gamma_*}\left(Y_{[0]}\xi'_{[0]} + Y'_{[0]}\xi_{[0]}\right) + \frac{2}{\gamma_*}\left(f^r_{[0]}R'_{[3]} + f^r_{[2]}R'_{[1]} + f^r_{[3]}R'_{[0]}\right),$$ (237d)

$$S^Y_{[5]} = -\frac{2\beta_{3*}}{\gamma_*} R^2_{[0]} R_{[1]} \xi'_{[0]} - \frac{\beta_{2*}}{\gamma_*} \left( R^2_{[0]} \xi'_{[3]} + 2R_{[0]} R_{[1]} \xi'_{[2]} + 2R_{[0]} R_{[3]} \xi'_{[0]} + 2R_{[1]} R_{[2]} \xi'_{[0]} \right)$$
$$- \frac{2\beta_*}{\gamma_*} \left( R_{[0]} \xi'_{[5]} + R_{[1]} \xi'_{[4]} + R_{[2]} \xi'_{[3]} + R_{[3]} \xi'_{[2]} \right) - \frac{4q_{1*}}{\gamma_*} R_{[1]} \xi_{[0]} \xi'_{[0]}$$
$$- \frac{2q_{0*}}{\gamma_*} \left( \xi_{[3]} \xi'_{[0]} + \xi'_{[3]} \xi_{[0]} \right) + \frac{d_{2*}}{\gamma_*} \left( R^2_{[0]} Y'_{[1]} + 2R_{[0]} R_{[1]} Y'_{[0]} \right)$$
$$+ \frac{2d_{1*}}{\gamma_*} \left( R_{[0]} Y'_{[3]} + R_{[1]} Y'_{[2]} + R_{[2]} Y'_{[1]} + R_{[3]} Y'_{[0]} \right) - \frac{u_{0*}}{\gamma_*} \left( Y_{[1]} \xi'_{[0]} + Y'_{[1]} \xi_{[0]} \right)$$
$$+ \frac{2}{\gamma_*} \left( f^r_{[0]} R'_{[4]} + f^r_{[2]} R'_{[2]} + f^r_{[3]} R'_{[1]} + f^r_{[4]} R'_{[0]} \right),$$

(237e)

$$S^Y_{[6]} = -\frac{\beta_{4*}}{2\gamma_*} R^4_{[0]} \xi'_{[0]} - \frac{2\beta_{3*}}{3\gamma_*} \left( R^3_{[0]} \xi'_{[2]} + 3R^2_{[0]} R_{[2]} \xi'_{[0]} + 3R^2_{[1]} R_{[0]} \xi'_{[0]} \right)$$
$$- \frac{\beta_{2*}}{\gamma_*} \left( R^2_{[0]} \xi'_{[4]} + R^2_{[1]} \xi'_{[2]} + R^2_{[2]} \xi'_{[0]} + 2R_{[0]} R_{[1]} \xi'_{[3]} \right.$$
$$\left. + 2R_{[0]} R_{[2]} \xi'_{[2]} + 2R_{[0]} R_{[4]} \xi'_{[0]} + 2R_{[1]} R_{[3]} \xi'_{[0]} \right)$$
$$- \frac{2\beta_*}{\gamma_*} \left( R_{[0]} \xi'_{[6]} + R_{[1]} \xi'_{[5]} + R_{[2]} \xi'_{[4]} + R_{[3]} \xi'_{[3]} + R_{[4]} \xi'_{[2]} \right) - \frac{2q_{2*}}{\gamma_*} R^2_{[0]} \xi_{[0]} \xi'_{[0]}$$
$$- \frac{4q_{1*}}{\gamma_*} \left( R_{[0]} \xi_{[0]} \xi'_{[2]} + R_{[0]} \xi_{[2]} \xi'_{[0]} + R_{[2]} \xi_{[0]} \xi'_{[0]} \right)$$
$$- \frac{2q_{0*}}{\gamma_*} \left( \xi_{[0]} \xi'_{[4]} + \xi_{[2]} \xi'_{[2]} + \xi_{[4]} \xi'_{[0]} \right) + \frac{2d_{3*}}{3\gamma_*} R^3_{[0]} Y'_{[0]}$$
$$+ \frac{d_{2*}}{\gamma_*} \left( R^2_{[0]} Y'_{[2]} + R^2_{[1]} Y'_{[0]} + 2R_{[0]} R_{[2]} Y'_{[0]} + 2R_{[0]} R_{[1]} Y'_{[1]} \right)$$
$$+ \frac{2d_{1*}}{\gamma_*} \left( R_{[0]} Y'_{[4]} + R_{[1]} Y'_{[3]} + R_{[2]} Y'_{[2]} + R_{[3]} Y'_{[1]} + R_{[4]} Y'_{[0]} \right) - \frac{v_*}{\gamma_*} Y_{[0]} Y'_{[0]}$$
$$- \frac{2u_{1*}}{\gamma_*} \left( R_{[0]} Y_{[0]} \xi'_{[0]} + R_{[0]} \xi_{[0]} Y'_{[0]} \right) - \frac{2u_{0*}}{\gamma_*} \left( \xi_{[0]} Y'_{[2]} + \xi_{[2]} Y'_{[0]} + Y_{[0]} \xi'_{[2]} + Y_{[2]} \xi'_{[0]} \right)$$
$$+ \frac{2}{\gamma_*} \left( f^r_{[0]} R'_{[5]} + f^r_{[2]} R'_{[3]} + f^r_{[3]} R'_{[2]} + f^r_{[4]} R'_{[1]} + f^r_{[5]} R'_{[0]} \right).$$

(237f)

All coefficients are listed in Appendix D.

One can use the sources above together with Eqs. (137), (138), (154) and (159) to compute the following coefficients

$$s^{x,0}_{[1]} = s^x_{[1]} = \frac{\alpha_*}{(\alpha_* \beta_* \kappa_*)^{4/5}} \left. F^r_1 \right|_{(0)*},$$

(238)

$$s^{x,0}_{[2]} = -\alpha_{4*} \frac{(\alpha_* \beta_* \kappa_*)^{2/5}}{\alpha^2_*} + \frac{\beta_{2*} \kappa_*}{(\alpha_* \beta_* \kappa_*)^{3/5}} - \frac{2}{3} \frac{\beta_*}{(\alpha_* \beta_* \kappa_*)^{3/5}} \left. \frac{dF_{\phi 1}}{dr} \right|_{(0)*} + \frac{4}{3} d_{1*} \frac{(\alpha_* \beta_* \kappa_*)^{2/5}}{\alpha_* \gamma_*},$$

(239)

$$s^{x,0}_{[3]} = \frac{2}{(\alpha_* \beta_* \kappa_*)^{2/5}} \left[ -\left( \frac{\alpha_{4*}}{2\alpha_*} + \frac{1}{3\kappa_*} \left. \frac{dF_{\phi 1}}{dr} \right|_{(0)*} + \frac{d_{1*}}{3\gamma_*} \right) \left. F^r_1 \right|_{(0)*} + \frac{1}{2} \left. \frac{dF^r_1}{dr} \right|_{(0)*} \right],$$

(240)

$$s^{y,0}_{[1]} = -\frac{\alpha_*}{(\alpha_* \beta_* \kappa_*)^{4/5}} \left. F^r_1 \right|_{(0)*},$$

(241)

$$s^{y,0}_{[2]} = -\frac{2}{\kappa_*} \left. \frac{dF_{\phi 1}}{dr} \right|_{(0)*} + \frac{\kappa_* \beta_{2*}}{(\alpha_* \beta_* \kappa_*)^{3/5}} - \frac{2\alpha^2_* \kappa^2_* q_{0*}}{(\alpha_* \beta_* \kappa_*)^{8/5}} + \frac{4d_{1*}(\alpha_* \beta_* \kappa_*)^{2/5}}{\alpha_* \gamma_*},$$

(242)

$$s_{[3]}^{y,0} = \frac{\beta_* \kappa_*}{(\alpha_* \beta_* \kappa_*)^{7/5}} \left[ -\frac{4}{3\kappa_*} \left. F_1^r \right|_{(0)*} \left. \frac{dF_{\phi 1}}{dr} \right|_{(0)*} + \left( \frac{2d_{1*}}{3\gamma_*} + \frac{\alpha_{4*}}{\alpha_*} \right) \left. F_1^r \right|_{(0)*} - \left. \frac{dF_1^r}{dr} \right|_{(0)*} \right]. \quad (243)$$

## D   Transition coefficients

The coefficients appearing in the leading transition motion are defined by

$$\alpha_* \equiv \frac{1}{4} \left. \frac{\partial^3 V^{\text{geo}}}{\partial r^3} \right|_{[0]*}, \qquad \beta_* \equiv -\frac{1}{2} \left( \frac{\partial^2 V^{\text{geo}}}{\partial r \partial \ell} + \Omega \frac{\partial^2 V^{\text{geo}}}{\partial r \partial e} \right)\Bigg|_{[0]*}, \qquad \gamma_* \equiv \left. \frac{\partial V^{\text{geo}}}{\partial \ell} \right|_{[0]*}, \quad (244)$$

$$\pi_* \equiv \sigma a^3 - 6\sigma a r_{(0)*} + 2(3+a^2) r_{(0)*}^{3/2} - 3\sigma a r_{(0)*}^2. \quad (245)$$

The first three coefficients are defined in Eq. (111) in the main text. Note that $\beta_*$ and $\gamma_*$ are parity-odd under spin reversal.

The coefficients appearing in the sources $S_{[i]}^{RY}$, $S_{[i]}^R$ and $S_{[i]}^Y$ of the equations of motion for $R_{[i]}$ and $Y_{[i]}$, $i = 0, \cdots, 6$ are as follows:

$$\alpha_{4*} = \frac{1}{12} \left. \frac{\partial^4 V^{\text{geo}}}{\partial r^4} \right|_{[0]*}, \quad (246)$$

$$\alpha_{5*} = \frac{1}{48} \left. \frac{\partial^5 V^{\text{geo}}}{\partial r^5} \right|_{[0]*}, \quad (247)$$

$$\alpha_{6*} = \frac{1}{240} \left. \frac{\partial^6 V^{\text{geo}}}{\partial r^6} \right|_{[0]*}, \quad (248)$$

$$\beta_{2*} = -\frac{1}{2} \left( \frac{\partial^3 V^{\text{geo}}}{\partial r^2 \partial \ell} + \Omega \frac{\partial^3 V^{\text{geo}}}{\partial r^2 \partial e} \right)\Bigg|_{[0]*}, \quad (249)$$

$$\beta_{3*} = -\frac{1}{4} \left( \frac{\partial^4 V^{\text{geo}}}{\partial r^3 \partial \ell} + \Omega \frac{\partial^4 V^{\text{geo}}}{\partial r^3 \partial e} \right)\Bigg|_{[0]*}, \quad (250)$$

$$\beta_{4*} = -\frac{1}{12} \left( \frac{\partial^5 V^{\text{geo}}}{\partial r^4 \partial \ell} + \Omega \frac{\partial^5 V^{\text{geo}}}{\partial r^4 \partial e} \right)\Bigg|_{[0]*}, \quad (251)$$

$$q_{0*} = \frac{1}{2} \left( 2\Omega^2 - \Omega^2 \frac{\partial^2 V^{\text{geo}}}{\partial e^2} - \frac{\partial^2 V^{\text{geo}}}{\partial \ell^2} - 2\Omega \frac{\partial^2 V^{\text{geo}}}{\partial e \partial \ell} \right)\Bigg|_{[0]*}, \quad (252)$$

$$q_{1*} = \frac{1}{4} \left( -\Omega^2 \frac{\partial^3 V^{\text{geo}}}{\partial r \partial e^2} - \frac{\partial^3 V^{\text{geo}}}{\partial r \partial \ell^2} - 2\Omega \frac{\partial^3 V^{\text{geo}}}{\partial r \partial e \partial \ell} \right)\Bigg|_{[0]*}, \quad (253)$$

$$q_{2*} = \frac{1}{2} \left( -\Omega^2 \frac{\partial^4 V^{\text{geo}}}{\partial r^2 \partial e^2} - \frac{\partial^4 V^{\text{geo}}}{\partial r^2 \partial \ell^2} - 2\Omega \frac{\partial^4 V^{\text{geo}}}{\partial r^2 \partial e \partial \ell} \right)\Bigg|_{[0]*}, \quad (254)$$

$$d_{1*} = \frac{1}{2} \Omega \left. \frac{\partial^2 V^{\text{geo}}}{\partial r \partial e} \right|_{[0]*}, \quad (255)$$

$$d_{2*} = \frac{1}{2} \Omega \left. \frac{\partial^3 V^{\text{geo}}}{\partial r^2 \partial e} \right|_{[0]*}, \quad (256)$$

$$d_{3*} = \frac{1}{4} \Omega \left. \frac{\partial^4 V^{\text{geo}}}{\partial r^3 \partial e} \right|_{[0]*}, \quad (257)$$

$$u_{0*} = \left( 2\Omega^2 - \Omega \frac{\partial^2 V}{\partial e \partial \ell} - \Omega^2 \frac{\partial^2 V}{\partial e^2} \right)\Bigg|_{[0]*}, \quad (258)$$

$$u_{1*} = -\frac{1}{2}\left(\Omega\frac{\partial^3 V}{\partial r \partial e \partial \ell} + \Omega^2\frac{\partial^3 V}{\partial r \partial e^2}\right)\bigg|_{[0]*}, \tag{259}$$

$$v_* = \left(2\Omega^2 - \Omega^2\frac{\partial^2 V^{\mathrm{geo}}}{\partial e^2}\right)\bigg|_{[0]*}. \tag{260}$$

The coefficients appearing in the expressions for $\delta_{[i]}$ are as follows:

$$\begin{aligned}
\rho_* =\ & a^6 + 32\sigma a^5 r_{[0]*}^{1/2} - 198a^4 r_{[0]*} + 480\sigma a^3 r_{[0]*}^{3/2} - 628a^2 r_{[0]*}^2 + 3a^4 r_{[0]*}^2 + 448\sigma a r_{[0]*}^{5/2} \\
& + 32\sigma a^3 r_{[0]*}^{5/2} - 136 r_{[0]*}^3 - 76a^2 r_{[0]*}^3 + 32\sigma a r_{[0]*}^{7/2} + 12 r_{[0]*}^4 + 3a^2 r_{[0]*}^4 - 6 r_{[0]*}^5 + r_{[0]*}^6,
\end{aligned} \tag{261}$$

$$\begin{aligned}
\lambda_* =\ & -3a^8 - 256\sigma a^7 r_{[0]*}^{1/2} + 2072a^6 r_{[0]*} - 6912\sigma a^5 r_{[0]*}^{3/2} + 12728a^4 r_{[0]*}^2 - 12a^6 r_{[0]*}^2 \\
& - 13824\sigma a^3 r_{[0]*}^{5/2} - 512\sigma a^5 r_{[0]*}^{5/2} + 8288a^2 r_{[0]*}^3 + 2632a^4 r_{[0]*}^3 - 2048\sigma a r_{[0]*}^{7/2} \\
& - 5120\sigma a^3 r_{[0]*}^{7/2} - 48 r_{[0]*}^4 + 5488a^2 r_{[0]*}^4 - 18a^4 r_{[0]*}^4 - 3584\sigma a r_{[0]*}^{9/2} - 256\sigma a^3 r_{[0]*}^{9/2} \\
& + 1120 r_{[0]*}^5 + 584a^2 r_{[0]*}^5 - 256\sigma a r_{[0]*}^{11/2} - 72 r_{[0]*}^6 - 12a^2 r_{[0]*}^6 + 24 r_{[0]*}^7 - 3 r_{[0]*}^8,
\end{aligned} \tag{262}$$

$$\begin{aligned}
v_* =\ & \left(2\sigma a - 3 r_{[0]*}^{1/2} + r_{[0]*}^{3/2}\right)\left(-4\sigma a^3 + 16a^2 r_{[0]*}^{1/2} - 24\sigma a r_{[0]*} + 12 r_{[0]*}^{3/2} + a^2 r_{[0]*}^{3/2}\right. \\
& \left. + 4\sigma a r_{[0]*}^2 - 6 r_{[0]*}^{5/2} + r_{[0]*}^{7/2}\right),
\end{aligned} \tag{263}$$

$$\begin{aligned}
\chi_* =\ & 3a^{10} + 512\sigma a^9 r_{[0]*}^{1/2} - 5150a^8 r_{[0]*} + 22016\sigma a^7 r_{[0]*}^{3/2} - 53128a^6 r_{[0]*}^2 + 15a^8 r_{[0]*}^2 \\
& + 78848\sigma a^5 r_{[0]*}^{5/2} + 1536\sigma a^7 r_{[0]*}^{5/2} - 71920a^4 r_{[0]*}^3 - 11384a^6 r_{[0]*}^3 + 36864\sigma a^3 r_{[0]*}^{7/2} \\
& + 34304\sigma a^5 r_{[0]*}^{7/2} - 7952a^2 r_{[0]*}^4 - 56984a^4 r_{[0]*}^4 + 30a^6 r_{[0]*}^4 + 57344\sigma a^3 r_{[0]*}^{9/2} \\
& + 1536\sigma a^5 r_{[0]*}^{9/2} - 96 r_{[0]*}^5 - 33248a^2 r_{[0]*}^5 - 7348a^4 r_{[0]*}^5 + 8192\sigma a r_{[0]*}^{11/2} \\
& + 12800\sigma a^3 r_{[0]*}^{11/2} + 240 r_{[0]*}^6 - 11928a^2 r_{[0]*}^6 + 30a^4 r_{[0]*}^6 + 7168\sigma a r_{[0]*}^{13/2} \\
& + 512\sigma a^3 r_{[0]*}^{13/2} - 2288 r_{[0]*}^7 - 1144a^2 r_{[0]*}^7 + 512\sigma a r_{[0]*}^{15/2} \\
& + 120 r_{[0]*}^8 + 15a^2 r_{[0]*}^8 - 30 r_{[0]*}^9 + 3 r_{[0]*}^{10},
\end{aligned} \tag{264}$$

$$\varphi_* = 2\sigma a - 2 r_{[0]*}^{1/2} + r_{[0]*}^{3/2}, \tag{265}$$

$$\mu_* = \left(\sigma a - r_{[0]*}^{1/2}\right)\left(a^4 - 6\sigma a^3 r_{[0]*}^{1/2} + 12a^2 r_{[0]*} - 10\sigma a r_{[0]*}^{3/2} + 2 r_{[0]*}^2 + a^2 r_{[0]*}^2\right), \tag{266}$$

$$\psi_* = -4\sigma a^3 + 16a^2 r_{[0]*}^{1/2} - 24\sigma a r_{[0]*} + 12 r_{[0]*}^{3/2} + a^2 r_{[0]*}^{3/2} + 4\sigma a r_{[0]*}^2 - 6 r_{[0]*}^{5/2} + r_{[0]*}^{7/2}, \tag{267}$$

$$\phi_* = \sigma a^3 - 8a^2 r_{[0]*}^{1/2} + 10\sigma a r_{[0]*} + a^2 r_{[0]*}^{3/2} - 3\sigma a r_{[0]*}^2 - 2 r_{[0]*}^{5/2} + r_{[0]*}^{7/2}. \tag{268}$$

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
