# Peer review of "Asymptotically matched quasi-circular inspiral and transition-to-plunge in the small mass ratio expansion"

_SciPost Physics, doi:SciPost Phys. 13, 043 (2022)_

## Round 1 · Referee Report · Anonymous (Referee 4) · 2022-2-21

Report

The present paper develops a method to compute the dynamics of an inspiraling compact binary beyond the innermost stable circular orbit in the small mass-ratio approximation (i.e. within the gravitational self-force formalism). This is important for constructing accurate and complete (inspiral-merger-ringdown) waveform models for gravitational wave astronomy. While this problem has been investigated in the past, the present work provides a systematic and rigorous treatment. Hence I think it fulfills the acceptance criteria for SciPost Physics, since it opens a new pathway in an existing research direction, with clear potential for multipronged follow-up work. I recommend to accept it.

  • validity: high
  • significance: good
  • originality: good
  • clarity: high
  • formatting: reasonable
  • grammar: good

Author:  Lorenzo Küchler  on 2022-03-04  [id 2268]

(in reply to Report 1 on 2022-02-21)

I would like to thank the referee for the careful reading and the positive assessment of our manuscript.

---

## Round 1 · Referee Report · Anonymous (Referee 3) · 2022-3-23

Report

This paper derives the asymptotic matching between a quasicircular, equatorial inspiral around a Kerr black hole and the plunge into the black hole including self-force effects, showing how to carry out this matching to high orders. This is a potentially important advance in modelling this portion of binary merger, though the paper leaves explicit calculations and the generalization to more generic orbits and finite size effects to later work. It thus satisfies expectation 3. It also satisfies all of the acceptance criteria except that one important bit of context (the accuracy of current calculations) is missing in the introduction, and there are some citations that are omitted or not the best ones. I give explicit suggestions in the requested changes. Once these and various other small issues I noticed are addressed, this will be suitable for publication in SciPost Physics.

The lengthy expressions obtained in the derivation are provided in Mathematica notebooks as well as typeset in the appendices of the paper. It is possible some of these expressions could just be given in Mathematica form, shortening the paper, but the expressions in the appendices are typeset well, so it is probably worth keeping them.

Requested changes

1- The abstract or at least the introduction should highlight that one only needs the second-order self-force to obtain quite high-order expressions for the transition in this framework, since this will clarify that the high-order expressions obtained here do not require significant developments in self-force calculations to obtain the third-order self-force in order to be useful.

2- The introduction should, to the extent possible without lengthy additional calculations, give some indication of the size of the errors that are being committed in current calculations of the matching to the plunge compared to the errors expected using the high-order scheme detailed here.

3- In the introduction, it seems preferable to cite recent reviews for NR, PN/PM, and EOB (e.g., dois:10.1146/annurev-astro-081913-040031, 10.12942/lrr-2014-2, and 10.1007/978-3-319-19416-5_7) rather than original papers in the second sentence. Also, the citation of observational results should presumably cite the latest GWTC-3 catalogue paper, arXiv:2111.03606, as well as the associated testing GR paper, arXiv:2112.06861, since the confrontation of models with observations is mentioned. If the GWTC-2 catalogue paper is still also cited, the associated testing GR paper should be cited, as well, as well as perhaps the GWTC-1 catalogue and testing GR papers, for completeness. (See https://www.ligo.caltech.edu/page/detection-companion-papers for the references.) I do not see a reason to cite the GW150914 paper here.

4- It is likely appropriate to cite a paper about EMRI science with LISA in the introduction, in addition to the general introduction to the mission in [11], e.g., doi:10.1103/PhysRevD.95.103012

5- The discussion in the final sentence of the first paragraph of the introduction should also mention the second-order results in arXiv:2112.12265 that appeared after this paper was submitted to the arXiv.

6- The opening text of Sec. II [through Eq. (2)] is taken almost verbatim from the authors’ previous work, Ref. [30], including the typo “Linquist” for “Lindquist.” This copying should at least be noted explicitly, though it would be better to rewrite the material.

7- The mention in the first paragraph of Sec. II that e = -p_t/m is made dimensionless using M is confusing, since it is already dimensionless.

8- The discussion above Eq. (30) is a bit confusing. It appears that what is meant is that D has a single root outside of the horizon and that root occurs at the location of the ISCO. Also, it is not clear why the roots of A, B, and C are not of interest here. Is it just that they all are inside the ISCO?

9- Below Eq. (31), it appears that the statement about even and odd quantities also involves switching the sign of a as well as the sign of \sigma, given the discussion in Appendix A. This should be made clear here, as I was initially confused.

10- In Sec. III C-D, is there a reason not to give T_1, …, T_20 names that reflect the expressions that they enter? The current notation suggests that they are all related, while they appear in very different expressions. They are not even labelled in the order they appear in the paper—T_15 through T_20 appear before T_8 through T_14.

11- It seems that the reason for using \kappa instead of \ell in Eq. (57a) is because \kappa appears in a similar manner to other Greek letters in Sec. IV. This probably deserves some comment.

12- Above Eq. (76), “inverse” -> “reciprocal” or “multiplicative inverse” so it’s clear that this does not refer to the functional inverse.

13- Above Eq. (92), it’s not clear to me how Eq. (82) is used in this calculation.

14- Below Eq. (98), it might be a good idea to recall that \delta_{(0)} = 0, to explain why there is a special case for q’_{(n)}.

15- In Sec. IV C, he discussion of the asymptotic solution is unclear, since the explicit examples of coefficients for the general case all vanish when d = 0, so it’s not clear how exactly one gets the expression in Eq. (142) from Eq. (141). Some more discussion is in order, e.g., mentioning which coefficients are nonzero for d = 0 and giving a few examples of these.

16- Why is it necessary to identify expressions in Eq. (199) instead of showing that they are equal, as in the previous matching calculations? Explicitly, what free parameters are being fixed by this identification?

17- In the conclusions, the discussion of using these results to calibrate EOB should cite the more recent EOB papers that perform this calibration, e.g., dois: 10.1103/PhysRevD.98.084028 and 10.1103/PhysRevD.102.024077 (for the higher modes, and perhaps also dois:10.1103/PhysRevD.95.044028 and 10.1103/PhysRevD.98.104052, which do the calibration for the dominant mode) and as well as the papers describing the extreme mass-ratio waveforms used for this calibration, dois:10.1103/PhysRevD.90.084025 and 10.1088/0264-9381/31/24/245004, likely instead of most of the older papers. You should probably also cite some of the non-EOB waveform papers that use extreme mass-ratio waveforms in the calibration, e.g., doi:10.1103/PhysRevD.102.064002 and arXiv:2012.11923. Also, “EOB” is not defined.

18- Also in the conclusions, the discussion of why the departure from quasicircularity in the inspiral is only at post-post-adiabatic order should be clarified, since \delta_{(1)} is nonzero (and this calculation is in the 1-post-adiabatic inspiral section).

* Minor issues:

19- In the third paragraph of the introduction, I find the use of the future tense when describing the contents of the paper distracting. I am used to the present tense in these sorts of guides to the contents of the paper, since the work is already completed.

20- There is an Eq. (104a) but no Eq. (104b).

21- Below Eq. (123), is there a reason not to list the equations being substituted in order of equation number? This also applies above Eqs. (197) and (198).

22- In Eqs. (144-5) and intervening expressions, as well as before Eq. (146), the “lin” subscript and superscript should always be set in Roman

23- Appendix B is only referred to in Appendices C and D, so perhaps it should go after them.

24- Ref. [11] should be P. Amaro-Seoane et al., not H. Audley et al.

  • validity: high
  • significance: ok
  • originality: good
  • clarity: ok
  • formatting: good
  • grammar: good

Author:  Lorenzo Küchler  on 2022-05-03  [id 2434]

(in reply to Report 2 on 2022-03-23)

We would like to thank the referee for at least partially reproducing our computations and results and for his/her pertinent comments. We addressed all questions and consequently improved the manuscript. The answers to the individual questions are listed below.

1-The abstract has been completed with 2 additional clarifications.

2-We mentioned the expected improvement in errors in the Introduction and justify these errors at the end of Section VI: we added (201)-(203) and commented on the error in the composite expansion.

3-We think that citing the founding papers of the topic is as appropriate as citing relevant reviews. We will therefore cite both. We agree with the other citation comments.

4-We agree and added the reference.

5-We agree and added the reference.

6.We adapted this convention section indeed totally shared with [35] as already stated and absolutely necessary for this paper as well. Thank you for pointing to the typo.

7-We corrected the phrasing.

8-We answered this question in the text after Eq. (30).

9-We rephrased after Eq. (31).

10-The T_i coefficients have been renamed in order of appearance. These coefficients are not specific to any expression: they are auxiliary functions to write more compact expressions for the quantities relevant to the inspiral.

11-Indeed, we changed the notation $\kappa$ to be coherent with previous notation $\ell$.

12-We will use "reciprocal".

13-Indeed, that reference should not be there. We removed it.

14-Thank you for pointing this out. We added the comment.

15-We added the first coefficients of the expansion that do not vanish when $d=0$.

16-This is indeed a check only. We rephrased.

17-We updated the references and defined "EOB" in the Introduction section.

18-We clarified the discussion in the conclusion and added a remark at the end of Section VD.

19-We now use present tense.

20-Fixed.

21-The referenced equations now appear in order of equation number.

22-Fixed.

23-We changed the order of appearance of the Appendices.

24-Fixed.

---

## Round 2 · Referee Report · Anonymous (Referee 2) · 2022-5-9

Report
Requested changes
1- Around Eq. (201), it would be good to explain why there is no \eta^{3/5} term at 0PA, even though there is an \eta^{4/5} term and there is an \eta^{3/5} term at 1PA. Also (and likely relatedly), it is necessary to reconcile the statement in the introduction that “the leading-order transition matching admits an error of the order of \eta^{3/5}” with the statement at the end of Sec. VI that “0PA composite expansion (201) leads to an error in the radius of the order of O_s(\eta) + O_\tilde{\tau}(\eta).” Of course, perhaps the statement in the introduction refers to a different expression, but if so, this needs to be made explicit.
2- Final sentence of abstract: “instrumental at” -> “instrumental for” (apologies for not noticing this previously).
3- I am fine with citing foundational papers along with the more recent reviews, in principle. However, just citing the Pretorius paper for the entire enterprise of numerical relativity (even just numerical relativity simulations of binary black holes) seems a bit odd, since even though it is the first BBH breakthrough paper (though commonly cited along with the two papers giving the moving punctures breakthrough), Pretorius’s code is not used to create the “accurate waveform models currently under confrontation with observations of gravitational waves produced from compact binary mergers” which instead are primarily created by the SXS collaboration with SpeC and secondarily by some of the moving punctures codes. (This also leaves aside numerical simulations with matter, which are also important for comparisons with observations but not so relevant to the current paper.) If you want to cite a single paper for numerical relativity along with the review, I would suggest instead the most recent SXS catalogue paper, doi:10.1088/1361-6382/ab34e2, since this describes the waveforms used to calibrate all binary black hole waveform models. If you want to cite the Pretorius paper, then it should be mentioned as purely being for the breakthrough (though I would argue that it is not the most useful citation here). I do not object to the PN/PM and EOB foundational citations, since the basic methods they introduce are still the ones used in current calculations, and the state-of-the-art calculations are still being carried out at least in part by these groups. Apologies for being picky about this, but I feel that it is important to be careful with these sorts of introductory citations.
4- In the final paragraph of the introduction, “expansions complements” -> “expansions complement”
5- Below Eq. (30), “in denominators as D” -> “in the same denominators that D does” or “in the denominators along with D” or something similar
6- Below Eqs. (31), “symmetry exist” -> “symmetry exists” and “when both” -> “when the signs of both”; also, the discussion of the parity property at the beginning of Appendix A should probably also mention the change of the sign of a, to prevent any confusion.
7- The lengthy numbers in s_{8,0} should be typeset with the same spacing between groups of numbers as for the c coefficients above Eq. (145) for consistency, or those coefficients should not have the spaces—I would be fine with either.
8- In Eqs. (201-3), putting the subscripts “0PA,” “1PA,” and “2PA” on the “r”s on the l.h.s. of the expressions would make things clearer.
9- In the conclusions, it’s probably worthwhile separating out the references for the calculations of EMRI waveforms ([21, 43]) from the papers that calibrate waveform models. Additionally, since you are citing the EOB papers that calibrate the dominant mode as well as those that calibrate the higher modes, it’s probably appropriate to cite the Phenom paper that calibrates the dominant mode of the frequency-domain model, doi:10.1103/PhysRevD.102.064001 (the calibration of the dominant mode of the time-domain Phenom model using the extreme mass-ratio waveforms is done in the paper that’s already cited). (I should have mentioned this explicitly in my previous report, as opposed to just giving the papers calibrating the higher modes with “e.g.”) Finally, it makes sense to cite the papers ordered by model, with the paper giving the dominant mode calibration first, i.e., in the order [41, 39, 42, 40] and 10.1103/PhysRevD.102.064001, [44, 45].
10- The sudden transition from the discussion of calibrating waveforms to “The deviation from quasi-circularity appears in the inspiral starting from post-adiabatic order.” is a bit jarring, so I suggest rewriting to let the reader know you’re changing subjects.
11- In the conclusions “without eccentricity nor inclination” -> “without either eccentricity or inclination” or something similar.
12- Bibliography: VIRGO -> Virgo, and it should be “LIGO Scientific, Virgo, and KAGRA Collaborations” (since they’re separate collaborations who author papers together) and sometimes the “D” in “Phys. Rev. D” gets put in the volume number (e.g., [3], presumably due to using old INSPIRE BibTeX entries), while in other cases it is correctly included as part of the journal name (e.g., [12]).
Author: Lorenzo Küchler on 2022-05-11 [id 2454]
(in reply to Report 1 on 2022-05-09)We thank the referee for her/his pertinent comments.
1- We thank the referee for finding this typo: we simply forgot to put the $R_{[1]}$ term in Eq. (201). For clarity, we now denote the composite expansion with both inspiral and transition approximations with the label nPA/mPL in Section VI. The scaling of the error in the introduction is correct and refers to the 0PA/0PL composite expansion, while the 0PA/2PL expansion (201) is precise up to $\eta$ corrections both at fixed $s$ and $\tilde\tau$. This is now explained at the end of Section VI.
2- Fixed, thank you.
3- We are not very knowledgeable of the numerical relativity literature and we thank you for your informative comments. We decided to cite the SXS catalog.
4- Fixed.
5- Fixed.
6- Fixed.
7- Fixed.
8- Fixed with the new notation nPA/mPL.
9- Fixed.
10- Fixed, we put this discussion in a new paragraph.
11- Fixed.
12- Fixed.

---

## Round 2 · Author Response

List of changes
1) We added absolute values above Eq. (118).
2) We added Table II (and a remark below Eq. (180)).
3) We added the reference to the published Erratum of the PRL.
4) We made the nomenclature consistent throughout the text: we use Painlevé "transcendent" as a noun referring to the solution to the Painlevé "transcendental" equation.
5) We changed Last Stable Circular Orbit to Last Stable Orbit (LSO).

---

## Round 2 · List of Changes

1) We added absolute values above Eq. (118).
2) We added Table II (and a remark below Eq. (180)).
3) We added the reference to the published Erratum of the PRL.
4) We made the nomenclature consistent throughout the text: we use Painlevé "transcendent" as a noun referring to the solution to the Painlevé "transcendental" equation.
5) We changed Last Stable Circular Orbit to Last Stable Orbit (LSO).

---

## Round 3 · Referee Report · Anonymous (Referee 2) · 2022-5-11

Report

The authors have addressed all my comments, so I am happy to recommend this for publication, with two small requested changes that can be made during the production stage.

Requested changes

1 - In Eqs. (201-3), the “PA” and “PL” parts of the superscripts should be set in Roman.

2- In the conclusions, “EOB waveforms [39–42] (derived using methods from [23, 43])” makes it sound like the EOB waveforms are derived using those methods. Something like “EOB waveforms [39–42] or other waveforms [44–46]. (These waveforms are currently are calibrated in part using extreme mass-ratio waveforms calculated using methods from [23, 43].)” would be clearer.

  • validity: high
  • significance: ok
  • originality: good
  • clarity: ok
  • formatting: good
  • grammar: good

Author:  Lorenzo Küchler  on 2022-05-18  [id 2491]

(in reply to Report 1 on 2022-05-11)

We would like to thank the referee for her/his comments which improved the quality of the paper and for the positive assessment of our manuscript.

---

## Round 3 · Author Response

We improved the manuscript addressing all of the referee's comments and made the requested changes.

---

## Round 4 · Author Response

We implement the final changes suggested by the referee.

---

## Editorial Decision

published